Hydrology and
Earth System
Sciences
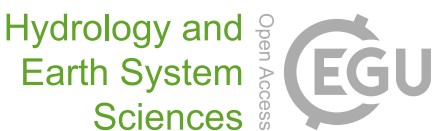

# Understanding dominant controls on streamflow spatial variability to set up a semi-distributed hydrological model: the case study of the Thur catchment

**Marco Dal Molin**[1,2,3]**, Mario Schirmer**[2,3]**, Massimiliano Zappa**[4]**, and Fabrizio Fenicia**[1]

[1]Department Systems Analysis, Integrated Assessment and Modelling, Eawag,
Swiss Federal Institute of Aquatic Science and Technology, 8600 Dübendorf, Switzerland
[2]The Centre of Hydrogeology and Geothermics (CHYN), University of Neuchâtel, 2000 Neuchâtel, Switzerland
[3]Department of Water Resources and Drinking Water, Eawag, Swiss Federal
Institute of Aquatic Science and Technology, 8600 Dübendorf, Switzerland
[4]Hydrological Forecast, Swiss Federal Research Institute WSL, 8903 Birmensdorf, Switzerland

**Correspondence:** Marco Dal Molin (marco.dalmolin@eawag.ch)

**Abstract.** This study documents the development of a semi-distributed hydrological model aimed at reflecting the dominant controls on observed streamflow spatial variability. The process is presented through the case study of the Thur catchment (Switzerland, 1702 km$^2$), an alpine and pre-alpine catchment where streamflow (measured at 10 sub-catchments) has different spatial characteristics in terms of amounts, seasonal patterns, and dominance of baseflow. In order to appraise the dominant controls on streamflow spatial variability and build a model that reflects them, we follow a two-stage approach. In a first stage, we identify the main climatic or landscape properties that control the spatial variability of streamflow signatures. This stage is based on correlation analysis, complemented by expert judgement to identify the most plausible cause–effect relationships. In a second stage, the results of the previous analysis are used to develop a set of model experiments aimed at determining an appropriate model representation of the Thur catchment. These experiments confirm that only a hydrological model that accounts for the heterogeneity of precipitation, snow-related processes, and landscape features such as geology produces hydrographs that have signatures similar to the observed ones. This model provides consistent results in space–time validation, which is promising for predictions in ungauged basins. The presented methodology for model building can be transferred to other case studies, since the data used in this work (meteorological variables, streamflow, morphology, and geology maps) are available in numerous regions around the globe.

## 1 Introduction

Semi-distributed rainfall–runoff models are widely applied in operation for applications such as flood forecasting (e.g. Ajami et al., 2004) or developing sustainable irrigation practices (e.g. McInerney et al., 2018). The main purpose of these models is to simulate streamflow at a limited number of fixed points along river channels (e.g. Boyle et al., 2001), and for this reason they are characterized by a coarser spatial resolution than fully distributed models, which allow a very detailed representation of the spatial variability of catchment processes. Compared to fully distributed models, they are characterized by lower data and computational requirements, which represents a valuable practical advantage in their operational use.

Similarly to the case of lumped models, the parameters of semi-distributed models are estimated via calibration. Therefore, it is important that the structure of these models is commensurate with the available data, including length, timescale, and spatial distribution (Wooldridge et al., 2001). However, semi-distributed models, even when used for sim-

ilar applications such as streamflow predictions, differ significantly in terms of their process representation as well as the number of calibration parameters. For example, Gao et al. (2014) assume topography to be a dominant control on hydrological processes, whereas the SWAT model (Arnold et al., 1998) emphasizes the role of soil properties. These differences raise the question of how to select an appropriate model structure for the data at hand, which requires understanding of the association between model parameters and the climatological and geomorphological characteristics of the catchment.

Understanding the relationship between climate, landscape, and catchment response is a common objective of many research areas in hydrology, including comparative hydrology (e.g. Falkenmark and Chapman, 1989), model regionalization (e.g. Parajka et al., 2005), catchment classification (e.g. Wagener et al., 2007), and prediction in ungauged basins (e.g. Hrachowitz et al., 2013). In the case of streamflow, the attempt to explain its spatial variability is typically accomplished either using statistical approaches, which are designed to regionalize selected characteristics of the hydrograph (streamflow signatures), or through hydrological models that account for relevant spatial information. In particular, statistical approaches such as regression analysis (e.g. Berger and Entekhabi, 2001; Bloomfield et al., 2009) and correlation analysis (e.g. Trancoso et al., 2017), or machine learning techniques like clustering (e.g. Sawicz et al., 2011; Toth, 2013; Kuentz et al., 2017), are used to group together catchments that present similar characteristics and to extrapolate the signatures where unknown. Such approaches have been useful for quantifying the hydrological variability and identifying its principal drivers. However, they are often not designed to discover causality links and can be affected by multicollinearity, which arises when multiple factors are correlated internally and with the target variable (Kroll and Song, 2013).

By incorporating spatial information about meteorological forcings and landscape characteristics, semi-distributed hydrological models have the ability to mimic the mechanisms that influence hydrograph spatial variability. However, identifying the relevant mechanisms is challenging. One possibility is to be as inclusive as possible in accounting for all the catchment properties that are, in principle, important in controlling catchment response. However, this approach leads to models that tend to be data demanding and contain many parameters. For example, Gurtz et al. (1999) considered several landscape characteristics (elevation, land use, etc.) in their application of a semi-distributed model to the Thur catchment (Switzerland), which resulted in a model with hundreds of hydrological response units (HRUs) that were defined a priori based on the complexity of the catchment. The other option is to try to identify the most relevant processes and neglect others in order to control model complexity. For example, Fenicia et al. (2016) compared various model hypotheses to determine an appropriate discretization of the catchment in HRUs and appropriate structures for different HRUs. Antonetti et al. (2016) used a map of dominant runoff processes following Scherrer and Naef (2003) for defining HRUs. However, these approaches require a good experimental understanding of the area, which is not always available.

Convincing model calibration–validation strategies are essential to provide confidence that the model ability to fit observations is a reflection of model realism and not a consequence of calibrating an overparameterized model (e.g. Andréassian et al., 2009). A common approach for the calibration of semi-distributed models is the so-called "sequential" approach, where subcatchments are calibrated sequentially from upstream to downstream (e.g. Verbunt et al., 2006; Feyen et al., 2008; Lerat et al., 2012; De Lavenne et al., 2016). Although this approach may provide good fits and therefore has its practical utility where data are available, it does not provide understanding of the causes of streamflow spatial variability and results in models that are not spatially transferable. Moreover, such models are prone to contain many parameters, as each subcatchment would be represented by its own set of parameters. Alternative calibration–validation approaches that enable model validation not only in time but also in space are conceptually preferable, particularly when the modelling is used for process understanding or prediction in ungauged locations (e.g. Wagener et al., 2004; Fenicia et al., 2016).

The objective of this study is to develop a semi-distributed hydrological model with the appropriate level of functional complexity to reproduce streamflow spatial variability in the Thur catchment. For this purpose, we use a two-stage approach, the first one dedicated to an in-depth analysis of the available data and the second one focused on hydrological modelling.

Our specific objectives are to (1) explore the spatial variability present in the Swiss Thur catchment regarding landscape characteristics, meteorological forcing, and streamflow signatures; (2) identify the main climate and landscape controls that explain the variability of the hydrological response; (3) based on this analysis, build a set of model experiments aimed to test the relative importance of dominant processes and their effect on the hydrograph; and (4) appraise model assumptions against competing alternatives using a stringent validation strategy.

The paper is organized as follows: Sect. 2 presents the study area and gives information about data availability; Sect. 3 illustrates the methodology; Sect. 4 shows the results; Sect. 5 analyses the results and puts them in perspective, showing what other similar studies have found; Sect. 6, finally, summarizes the main conclusions.

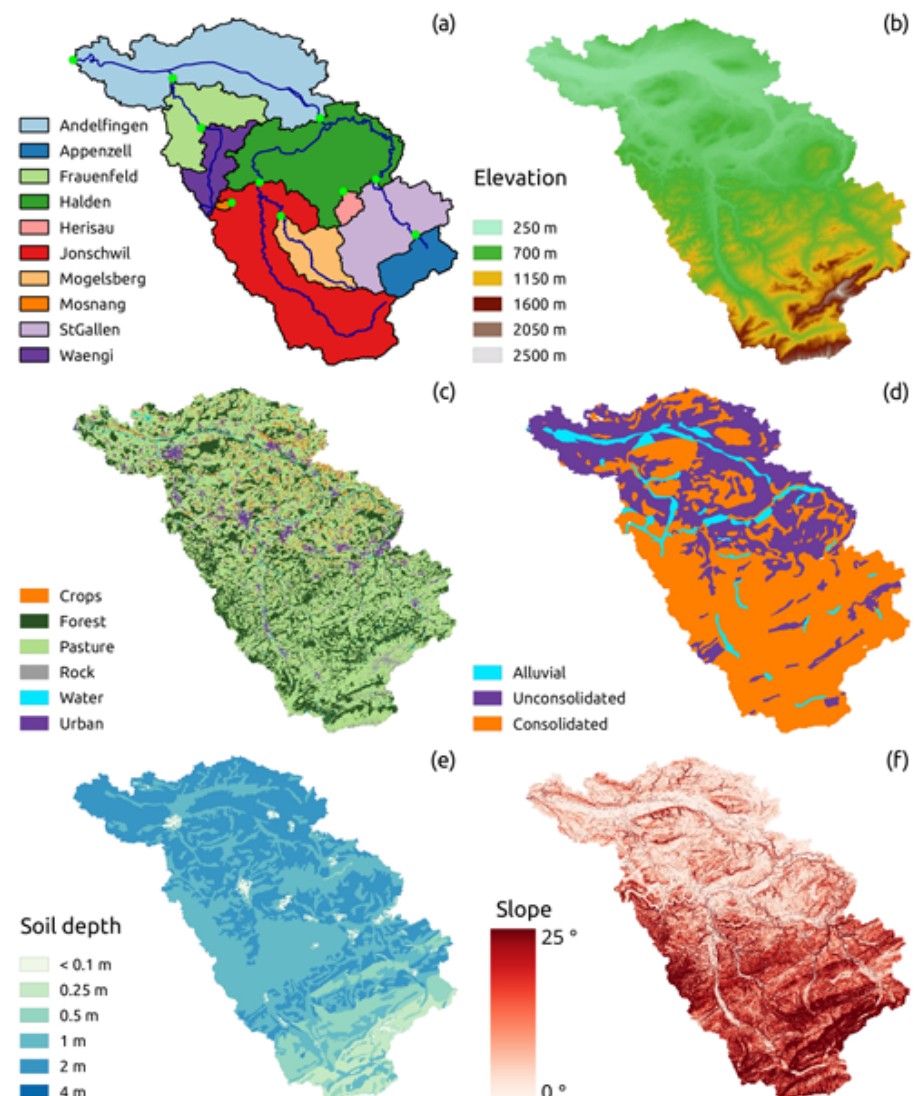

**Figure 1.** TS1 Landscape characteristics of the Thur catchment: **(a)** subdivision into subcatchments, river network, and gauging stations; **(b)** elevation map; **(c)** land-use map; **(d)** simplified geology map; **(e)** soil depth map; **(f)** slope map (derived from the elevation map).

## 2 Study area

This study is carried out in the Thur catchment (Fig. 1), located in the north-east of Switzerland, south-west of Lake Constance. With a total length of 127 km and a catchment area of 1702 km², the Thur is the longest Swiss river, without any natural or artificial reservoir along its course. The Thur River is very dynamic, with streamflow values that can change by 2 CE1 orders of magnitude within a few hours (Schirmer et al., 2014).

The Thur catchment has been the subject of several studies in the past: Gurtz et al. (1999) performed the first modelling study on the entire catchment using a semi-distributed hydrological model; Abbaspour et al. (2007) modelled hydrology and water quality using the SWAT model; Fundel et al. (2013) and Jorg-Hess et al. (2015) focused on low flows and droughts; Jasper et al. (2004) investigated the impact of climate change on the natural water budget. Other modelling studies also include Melsen et al. (2014, 2016), who investigated parameter estimation in data-limited scenarios and parameter transferability across spatial and temporal scales, and Brunner et al. (2019), who studied the spatial dependence of floods. The Thur also includes a small-sized experimental subcatchment (Rietholzbach, called Mosnang in this paper after the name of the gauging station) that was the subject of many field studies at the interface between process understanding and hydrological modelling (e.g. Menzel, 1996; Gurtz et al., 2003; Seneviratne et al., 2012; von Freyberg et al., 2014, 2015).

**Table 1.** Identification of the gauging stations and description of the river network.

|  | Index | Code* | Upstream catchments |
|---|---|---|---|
| Andelfingen | 1 | 2044 | 2–10 |
| Appenzell | 2 | 2112 | – |
| Frauenfeld | 3 | 2386 | 10 |
| Halden | 4 | 2181 | 2, 3, 5–10 |
| Herisau | 5 | 2305 | – |
| Jonschwil | 6 | 2303 | 7, 8 |
| Mogelsberg | 7 | 2374 | – |
| Mosnang | 8 | 2414 | – |
| St. Gallen | 9 | 2468 | 2 |
| Wängi | 10 | 2126 | – |

* Code of the gauging station, as defined by the Federal Office for the Environment, FOEN.

The topography of the catchment is presented in Fig. 1b; the elevation ranges between 356 m a.s.l. at the outlet and 2502 m a.s.l. at Mount Säntis. The majority of the catchment lies below 1000 m a.s.l. (75 %) and only 0.6 % is above 2000 m a.s.l. (Gurtz et al., 1999). Based on topography (Fig. 1b), the catchment can be visually subdivided into two distinct regions: the northern part, with low elevation and dominated by hills and flat land, and the southern part, which presents a mountainous landscape.

The land use (Fig. 1c) is dominated by pasture and sparsely vegetated soil (60 %) and forest (25 %); urbanized and cultivated areas are located mainly in the north and cover 7 % and 4 % of the catchment respectively.

Most of the catchment is underlain by conglomerates, marl incrustations, and sandstone (Gurtz et al., 1999). For the purpose of this study, the geological formations have been divided into three classes (Fig. 1d): "consolidated", covering mainly the mountainous part of the catchment, "unconsolidated", located in the north, and "alluvial", located in the proximity of the river network, mainly in the plateau area; the latter formation constitutes the main source of groundwater in the region (Schirmer et al., 2014). The soil depth (Fig. 1e) is shallower in the mountainous part of the catchment and deeper in the northern part.

Based on the availability of gauging stations (Table 1), the catchment was divided into 10 subcatchments (Fig. 1a), with a total drained area that ranges between 3.2 km² (Mosnang) and 1702 km² (Andelfingen). Streamflow time series are obtained from the Federal Office for the Environment (FOEN) CE2, and the data are available from 1974 to 2017 but are used only from 1981 to 2005 to match the precipitation, temperature, and potential evapotranspiration (PET) time series. In the considered range, the streamflow data are relatively continuous, with two gaps, one in St. Gallen, from 31 December 1981 to 1 January 1983, and the other one in Herisau, from 31 December 1982 to 9 May 1983.

The raw maps (topography, land use, geology, and soil) are obtained from the Federal Office of Topography (swisstopo). The meteorological data are obtained from the Federal Office of Meteorology and Climatology (MeteoSwiss) CE3. Precipitation and temperature are interpolated, as done in Melsen et al. (2016), with the WINMET pre-processing tool (Viviroli et al., 2009) using inverse distance weight (IDW) and detrended IDW respectively; while the first method considers only the horizontal variability (related to the distance from the meteorological stations), the second adds a vertical component to the variability related to the elevation (Garen and Marks, 2001). PET data are then obtained, as done in Gurtz et al. (1999), starting from meteorological and land-use data, using the Penman–Monteith equation (Monteith, 1975), implemented as part of the PREVAH hydrological model (Viviroli et al., 2009). All these values are calculated at pixel (100 m) scale and then averaged over the subcatchments. All the time series are used at daily resolution in the subsequent analyses, aggregating the available hourly data. This choice was influenced on the one hand by the need to limit the computational demand for the model experiments, for which a coarser temporal resolution is preferable, and on the other hand by the need to represent relevant hydrograph dynamics, for which finer temporal resolution is desirable (e.g. Kavetski et al., 2011). A daily data resolution, although it may obscure subdaily process dynamics, appeared to be a good compromise, and it is a typical choice in distributed model applications at such spatial scales (e.g. Kirchner et al., 2004).

## 3  Methods

The methodology follows a two-stage approach. The first stage aims at determining the climatic and landscape controls on streamflow signatures. The second stage uses this understanding to configure the structure of a semi-distributed model, whose functional suitability is tested through a set of model experiments. Section 3.1 describes the first stage of the analysis, that is, the identification of influence factors on the spatial variability of streamflow signatures. Section 3.2 describes the general structure of the semi-distributed model and the model evaluation approach. The design of the model experiments, which is dependent on the outcomes of the first stage of analyses, is presented directly in the results (Sect. 4.2.2).

### 3.1  Identification of influence factors on the spatial variability of streamflow signatures

The purpose of the analysis presented in this section is to understand the influence of climatic conditions and landscape characteristics on streamflow. Climatic conditions are represented by precipitation, potential evaporation, and temperature time series. Landscape characteristics are represented by maps of topography, land use, geology, and soil.

Climatic conditions, landscape characteristics, and streamflow are represented through a set of statistics (listed in Table 2). In the following, statistics calculated based on streamflow data will be called streamflow "signatures", as is often done in the catchment classification literature (e.g. Sivapalan, 2006). We will refer to climatic and landscape "indices" for statistics calculated based on climatic data and landscape characteristics. A broad list of signatures and indices is presented in Sect. 3.1.1; Sect. 3.1.2 presents the approach for reducing such a list to a set of meaningful variables; Sect. 3.1.3 illustrates the approach for determining the indices that mostly control streamflow signatures; Sect. 3.1.4 describes how the signature analysis is used to set up the model experiments.

### 3.1.1 Catchment indices for representing streamflow, climate, and landscape

Streamflow signatures ($\zeta$) and climatic indices ($\psi$) were obtained using streamflow, precipitation, PET, and temperature time series. The values were calculated using 24 years of data, between 1 September 1981 and 31 August 2005; we considered 1 September to be the beginning of the hydrological year. The periods with gaps in the data (refer to Sect. 2 for details) were discarded from the analysis of the specific subcatchment. Landscape indices were obtained using the maps described in Sect. 2.

Addor et al. (2017) recently compiled a comprehensive list of streamflow signatures and climatic indices for characterizing catchment behaviour (see Table 3 in Addor et al., 2017). Here, we adopted their selection: while being originally introduced for a study about large-sample hydrology, we believe that the indices proposed are also able to capture several different aspects of the time series and are therefore also suitable for this regional study. The streamflow signatures considered here are described hereafter, followed by an explanation of their rationale:

- average daily streamflow $\zeta_Q$ TS2 $= \overline{q}$, where $\boldsymbol{q}$ is the streamflow time series and the overbar represents the average over the observation period;

- runoff ratio $\zeta_{RR} = \frac{\overline{q}}{\overline{p}}$, where $\boldsymbol{p}$ is the precipitation time series;

- streamflow elasticity ($\zeta_{EL}$) defined as

$$\zeta_{EL} = \mathrm{med}\left(\left(\frac{\Delta \overline{q}}{\overline{q}}\right) \bigg/ \left(\frac{\Delta \overline{p}}{\overline{p}}\right)\right), \qquad (1)$$

where $\Delta \overline{q}$ and $\Delta \overline{p}$ represent the streamflow and precipitation difference between two consecutive years and med is the median function;

- slope of the flow duration curve ($\zeta_{FDC}$) defined as the slope between the log-transformed 33rd and 66th streamflow percentiles;

- baseflow index $\zeta_{BFI} = \frac{\overline{q^{(b)}}}{\overline{q}}$, where $\boldsymbol{q}^{(b)}$ represents the baseflow and was calculated using a low-pass filter as illustrated in Ladson et al. (2013) with the equations

$$q_t^{(f)} = \left(0, \vartheta_b q_{t-1}^{(f)} + \frac{1 + \vartheta_b}{2}(q_t - q_{t-1})\right), \qquad (2)$$

$$q_t^{(b)} = q_t - q_t^{(f)}, \qquad (3)$$

with $q_t^{(f)}$ representing the quickflow. The settings of the filter were taken according to the findings of Ladson et al. (2013) and, in particular, three filter passes were applied (forward, backward, and forward), the parameter $\vartheta_b$ was chosen to be equal to 0.925, and a reflection of 30 time steps at the beginning and at the end of the time series was used;

- mean half streamflow date ($\zeta_{HFD}$) (Court, 1962), defined as the number of days needed in order to have a cumulated streamflow that reaches 50 % of the total annual streamflow;

- 5th and 95th percentiles of the streamflow ($\zeta_{Q5}$ and $\zeta_{Q95}$ respectively);

- frequency ($\zeta_{HQF}$) and mean duration ($\zeta_{HQD}$) of high-flow events: they are defined as the days when the streamflow is bigger than 9 CE5 times the median daily streamflow;

- frequency ($\zeta_{LQF}$) and mean duration ($\zeta_{LQD}$) of low-flow events: they are defined as the days when the streamflow is smaller than 0.2 times the mean daily streamflow.

The frequency of days with zero streamflow, present in Addor et al. (2017), was not considered in this study because there are no ephemeral subcatchments in the study area.

This group of streamflow signatures is capable of capturing various characteristics of the hydrograph: $\zeta_Q$ measures the overall water flows, $\zeta_{RR}$ represents the proportion of precipitation that becomes streamflow, $\zeta_{EL}$ measures the sensitivity of the streamflow to precipitation variations, with a value greater than 1 indicating an elastic subcatchment (i.e. sensitive to change in precipitation) (Sawicz et al., 2011), $\zeta_{FDC}$ measures the variability of the hydrograph, with a steeper flow duration curve indicating a more variable streamflow, $\zeta_{BFI}$ measures the magnitude of the baseflow component of the hydrograph and can be considered a proxy for the relative amount of groundwater flow in the hydrograph, $\zeta_{HFD}$ measures the streamflow seasonality, $\zeta_{Q5}$, $\zeta_{LQF}$, and $\zeta_{LQD}$ measure low-flow dynamics, and $\zeta_{Q95}$, $\zeta_{HQF}$, and $\zeta_{HQD}$ measure high-flow dynamics.

Climatology was represented through the following indices (see Addor et al., 2017, Table 2):

- average daily precipitation $\psi_P = \overline{p}$;

**Table 2.** List of streamflow signatures, climatic indices, and subcatchment characteristics considered in the study.

| Symbol | Name |
| --- | --- |
| **Streamflow signatures** | |
| $\zeta_Q$ (mm d$^{-1}$) CE4 | Average daily streamflow |
| $\zeta_{RR}$ (−) | Runoff ratio |
| $\zeta_{EL}$ (−) | Streamflow elasticity |
| $\zeta_{FDC}$ (−) | Slope of the flow duration curve |
| $\zeta_{BFI}$ (−) | Baseflow index |
| $\zeta_{HDF}$ (day of the year) | Mean half streamflow date |
| $\zeta_{Q5}$ (mm d$^{-1}$) | 5th percentile of the streamflow |
| $\zeta_{Q95}$ (mm d$^{-1}$) | 95th percentile of the streamflow |
| $\zeta_{HQF}$ (d yr$^{-1}$) | Frequency of high-flow events |
| $\zeta_{HQD}$ (d) | Mean duration of high-flow events |
| $\zeta_{LQF}$ (d yr$^{-1}$) | Frequency of low-flow events |
| $\zeta_{LQD}$ (d) | Mean duration of low-flow events |
| **Climatic indices** | |
| $\psi_P$ (mm d$^{-1}$) | Average daily precipitation |
| $\psi_{PET}$ (mm d$^{-1}$) | Average daily potential evapotranspiration |
| $\psi_{AI}$ (−) | Aridity index |
| $\psi_{FS}$ (−) | Fraction of snow |
| $\psi_{HPF}$ (d yr$^{-1}$) | Frequency of high-precipitation events |
| $\psi_{HPD}$ (d) | Mean duration of high-precipitation events |
| $\psi_{HDS}$ (−) | Season with most high-precipitation events |
| $\psi_{LPF}$ (d yr$^{-1}$) | Frequency of low-precipitation events |
| $\psi_{LPD}$ (d) | Mean duration of low-precipitation events |
| $\psi_{LPS}$ (−) | Season with most low-precipitation events |
| **Subcatchment characteristics** | |
| $\xi_A$ (km$^2$) | Subcatchment area |
| $\xi_{TE}$ (m) | Average elevation |
| $\xi_{TS_m}$ (°) | Average slope |
| $\xi_{TS_s}$ (%) | Fraction of the subcatchment with steep areas |
| $\xi_{TA_s}$ (%) | Fraction of the subcatchment facing south |
| $\xi_{TA_n}$ (%) | Fraction of the subcatchment facing north |
| $\xi_{TA_{ew}}$ (%) | Fraction of the subcatchment facing east or west |
| $\xi_{SM}$ (m) | Average soil depth |
| $\xi_{SD}$ (%) | Fraction of the subcatchment with deep soil |
| $\xi_{LF}$ (%) | Fraction of the subcatchment with forest land use |
| $\xi_{LC}$ (%) | Fraction of the subcatchment with crop land use |
| $\xi_{LU}$ (%) | Fraction of the subcatchment with urbanized land use |
| $\xi_{LP}$ (%) | Fraction of the subcatchment with pasture land use |
| $\xi_{GA}$ (%) | Fraction of the subcatchment with alluvial geology |
| $\xi_{GC}$ (%) | Fraction of the subcatchment with consolidated geology |
| $\xi_{GU}$ (%) | Fraction of the subcatchment with unconsolidated geology |

– average daily PET $\psi_{PET} = \overline{e_{pot}}$, where $e_{pot}$ is the potential evapotranspiration time series;

– aridity index $\psi_{AI} = \frac{\overline{e_{pot}}}{\overline{p}}$;

– fraction of snow ($\psi_{FS}$), defined as the volumetric fraction of precipitation falling as snow (i.e. on days colder than 0 °C);

– frequency ($\psi_{HPF}$) and mean duration ($\psi_{HPD}$) of high-precipitation events: they are defined as days when the precipitation is more than 5 times the mean daily precipitation;

– season ($\psi_{HPS}$) with most high-precipitation events (defined as above);

**Table 3.** Values of the streamflow signatures. The names of the subcatchments are abbreviated using the first three letters and the symbols are reported in Table 2. The last column contains the coefficient of variation of each signature.

| | | | | | | Subcatchment | | | | | |
|---|---|---|---|---|---|---|---|---|---|---|---|
| | And | App | Fra | Hal | Her | Jon | Mog | Mos | StG | Wän | CV |
| $\zeta_Q$ | 2.46 | 4.14 | 1.64 | 3.08 | 2.95 | 3.71 | 3.21 | 2.91 | 3.43 | 2.03 | 0.25 |
| $\zeta_{RR}$ | 0.63 | 0.80 | 0.49 | 0.70 | 0.71 | 0.80 | 0.70 | 0.72 | 0.71 | 0.56 | 0.14 |
| $\zeta_{EL}$ | 1.35 | 1.22 | 1.68 | 1.24 | 1.17 | 1.35 | 0.97 | 1.37 | 0.99 | 1.54 | 0.17 |
| $\zeta_{FDC}$ | 2.12 | 2.41 | 2.11 | 2.30 | 2.08 | 2.49 | 2.76 | 2.78 | 2.47 | 2.02 | 0.12 |
| $\zeta_{BFI}$ | 0.55 | 0.50 | 0.56 | 0.52 | 0.50 | 0.50 | 0.45 | 0.42 | 0.48 | 0.57 | 0.10 |
| $\zeta_{HDF}$ | 194.21 | 220.63 | 170.38 | 202.00 | 193.87 | 205.38 | 196.96 | 168.33 | 209.36 | 173.17 | 0.09 |
| $\zeta_{Q5}$ | 0.50 | 0.70 | 0.35 | 0.57 | 0.74 | 0.54 | 0.44 | 0.28 | 0.60 | 0.49 | 0.27 |
| $\zeta_{Q95}$ | 6.96 | 12.85 | 4.83 | 9.23 | 9.17 | 11.19 | 10.57 | 10.46 | 11.00 | 5.98 | 0.28 |
| $\zeta_{HQF}$ | 2.21 | 5.17 | 3.50 | 3.67 | 6.34 | 4.46 | 6.54 | 12.96 | 5.87 | 2.96 | 0.57 |
| $\zeta_{HQD}$ | 1.39 | 1.25 | 1.45 | 1.35 | 1.40 | 1.39 | 1.37 | 1.58 | 1.35 | 1.29 | 0.06 |
| $\zeta_{LQF}$ | 17.50 | 31.92 | 12.92 | 24.21 | 2.62 | 37.21 | 49.42 | 66.92 | 28.35 | 7.25 | 0.71 |
| $\zeta_{LQD}$ | 6.67 | 6.18 | 3.69 | 6.53 | 2.00 | 7.44 | 6.38 | 7.11 | 4.53 | 4.35 | 0.32 |

– frequency ($\psi_{LPF}$) and mean duration ($\psi_{LPD}$) of dry days: they are defined as days when the precipitation is lower than $1\,\mathrm{mm\,d^{-1}}$;

– season ($\psi_{LPS}$) with most dry days (defined as above).

The seasonality of precipitation used in Addor et al. (2017) was not considered in this study as it relied on fitting a sinusoidal function to the precipitation values, which in our case did not produce reliable results. Nevertheless, these climatological indices are able to comprehensively represent the climatic conditions of the subcatchment, with $\psi_P$ representing average water input, $\psi_{PET}$ representing average evaporative demand, $\psi_{AI}$ measuring the dryness of the climate, $\psi_{FS}$ measuring the relative importance of snow, $\psi_{HPF}$, $\psi_{HPD}$, and $\psi_{HPS}$ measuring the importance of intense precipitation events, and $\psi_{LPF}$, $\psi_{LPD}$, and $\psi_{LPS}$ measuring the importance of dry days.

The landscape characteristics were divided into four categories: topography, land use, soil, and geology. In order to quantify the characteristics of each category, a set of indices ($\xi$) was defined. It is important to notice that all the areas calculated in this analysis were normalized by the respective subcatchment area ($\xi_A$) in order to get comparable values between subcatchments of different sizes.

Topography was represented with the following indices, calculated based on the digital elevation model:

– average elevation ($\xi_{TE}$);

– average slope ($\xi_{TS_m}$);

– fraction of the subcatchment with steep areas ($\xi_{TS_s}$), with slope larger than $10°$;

– aspect, i.e. fraction of the subcatchment facing north ($\xi_{TA_n}$), south ($\xi_{TA_s}$), or east and west ($\xi_{TA_{ew}}$).

Land use was represented with the following characteristics, obtained by reclassifying the land-use map into four categories (from 22 original classes):

– fraction of the subcatchment with crop land use ($\xi_{LC}$);

– fraction of the subcatchment with pasture land use ($\xi_{LP}$);

– fraction of the subcatchment with forest land use ($\xi_{LF}$);

– fraction of the subcatchment with urbanized land use ($\xi_{LU}$).

Soil type was represented with the following indices, derived by the soil map:

– fraction of the subcatchment with deep soil (soil depth greater than $2\,\mathrm{m}$) ($\xi_{SD}$);

– average soil depth ($\xi_{SM}$).

Geology was represented by the following indices, obtained by reclassifying the original map into three categories (from 22 original classes):

– fraction of the subcatchment with alluvial geology ($\xi_{GA}$);

– fraction of the subcatchment with consolidated geology ($\xi_{GC}$);

– fraction of the subcatchment with unconsolidated geology ($\xi_{GU}$).

The reclassification of the land use and of the geology maps consisted in aggregating specific classes into general classes (e.g. combining different types of forests into a unique forest class) with the objective of reducing their number, in order to facilitate subsequent analyses.

The consideration of topography, land use, soil, and geology for defining landscape indices was based on their potential influence on hydrological processes and, in turn, on the shape of the hydrograph. These landscape characteristics can all play an important role in controlling hydrological processes: land use can, for example, influence the infiltration of water in the substrate; soil thickness can affect the partitioning between water storage and runoff; vegetation is typically assumed to affect evaporation, and geology can affect groundwater dynamics. Indeed, these characteristics are used by many semi-distributed hydrological models, for example for determining parameter values or for dividing the catchment into areas with a homogenous hydrological response (e.g. Gurtz et al., 1999).

### 3.1.2 Selection of meaningful streamflow signatures, climatic indices, and catchment indices

The sets of statistics presented in Sect. 3.1.1 were designed to be comprehensive. However, they may also be redundant, for example by containing metrics that express similar characteristics of the underlying data. In order to facilitate subsequent correlation analyses between the various sets of statistics, it is important to reduce each set to a short list of meaningful variables. The reduction of each set of streamflow signatures, climatic indices, and landscape indices was achieved through the following steps.

- All the statistics that did not show sufficient variability between the subcatchments were eliminated. We were in fact interested in identifying causes of spatial variability in the streamflow dynamics of the subcatchments of the Thur. Therefore, statistics that had a low variability were not of interest in this analysis. The variability was assessed using the coefficient of variation (defined by the ratio between the standard deviation and the average) and statistics with a value lower than 5 % were discarded.

- All the catchment indices (e.g. a certain type of land use) that account for a limited part of any subcatchment were discarded. This point was motivated by the expectation that landscape characteristics covering a very small fraction of the subcatchment should not have a strong influence on the streamflow signatures considered. Here, landscape indices accounting for less than 5 % of any subcatchment area were discarded.

- Within each set of streamflow signatures, climatic indices, and catchment indices, we retained only relatively independent metrics, if these are believed to represent the same underlying features of the time series. This step was motivated by the need to remove redundant information within each set. The selection of independent metrics was aided by Spearman's rank score between each pair of metrics, which represents (also nonlinear) correlation between variables. Pairs of metrics

with high absolute values of Spearman's rank score are potentially redundant. In eliminating potentially redundant variables, we adopted the following criteria.

- Among highly correlated metrics, we preferred those depending on single variables (e.g. only precipitation or only streamflow) to those containing multiple variables (e.g. combining precipitation and streamflow or evaporation, such as the runoff ratio or the aridity index), as this may be a problem when looking for correlations between metrics.

- With respect to landscape indices, in many cases the high correlation is due to the fact that they are complementary (the areal fractions sum up to unity). In such cases, we kept one index per class (e.g. a single index for geology).

- A high correlation between metrics does not always mean that the metrics represent the same information. Therefore, the final selection of relevant metrics within each set was guided by expert judgement.

Based on this process, we compiled a reduced list of signatures, climatic indices, and landscape indices, which was used in subsequent analyses.

### 3.1.3 Identification of climate and landscape controls on streamflow and consequences for model development

This analysis aimed to identify climatic and landscape indices that mostly control streamflow signatures. In order to identify causality links between indices ($\psi$ and $\xi$) and signatures ($\zeta$), we proceed as follows:

- we calculated the correlation between indices and signatures using Spearman's rank score and identified pairs of variables with high correlation;

- we scrutinized pairs of variables with high correlations using expert judgement to decide whether a causality link between variables is justified.

The outcome of this process will be used to inform the semi-distributed model setup. The expert judgement is a critical step in the elicitation of causality from correlation (e.g. Antonetti and Zappa, 2018), and it is clearly subjective, being dependent on personal experience and subject matter knowledge. Although personal and subjective, expert decisions are based on an attempt to interpret data rather than being a priori defined, which is typically the case in the application of semi-distributed hydrological models.

### 3.1.4 Semi-distributed model setup and model experiments

We assumed a generic structure for a semi-distributed hydrological model, described in Sect. 3.2.1, where some model

structure characteristics are defined a priori and others are to be defined. In order to motivate the open decisions, we proceeded as follows:

– we used the identified causality links to interpret the dominant processes influencing signature spatial variability;

– we designed model experiments aimed to confirm the hypothesized climatic and landscape controls on streamflow spatial variability.

The overall objective of the model experiments is to prove that only models that incorporate the correct dependencies are able to correctly predict regional streamflow variability. In order to test this assumption, the model experiments will include cases where the assumed dependencies are not incorporated. Omitting an assumed dependency leads to a structurally simpler model, which may raise doubt that potential differences in model performance might be due to differences in model complexity. For this reason, the model experiments will include cases where alternative dependencies are incorporated, which do not reduce model complexity. In order to keep the study and presentation tractable, the model experiments will be limited to a few cases, illustrated in Sect. 4.2.1, which we judge relevant for this specific application.

### 3.2 General structure of the semi-distributed hydrological model and model evaluation approach

This section describes the approach for building and testing a semi-distributed hydrological model designed to represent the observed streamflow and particularly the observed spatial variability of streamflow signatures. The general model structure is explained in Sect. 3.2.1, the error model and the calibration procedure are described in Sects. 3.2.2 and 3.2.3, and the metrics utilized to assess the performance are shown in Sect. 3.2.4.

#### 3.2.1 General structure of the hydrological model

We describe here the general model structure; the definition of specific model experiments, which depends on the results of the signature analysis done in the first step, will be described in Sect. 4.2.2.

The model uses a two-layer decomposition of the catchment.

1. Subcatchments are defined by the presence of the gauging stations; this subdivision is due to the necessity of having locations in the model where the streamflow is both observed and simulated and, therefore, it is possible to calibrate and evaluate the parameters of the hydrological model.

2. HRUs are defined based on catchment characteristics (e.g. topography, geology, or vegetation); they represent

parts of the catchment that are supposed to have a similar hydrological response to the meteorological forcing. Each HRU is characterized by its own parameterization. Different definitions of HRUs are tested, as described in Sect. 4.2.2.

Each HRU has a unique parameterization. However, depending on how the inputs are discretized, the same HRU can have different states in different parts of the catchment. Therefore, the same HRU needs its own model representation whenever the spatial variability of states needs to be considered. For example, if the inputs are discretized per subcatchment, the same HRU needs a separate model representation in each subcatchment where it is present. For more details about our model implementation of HRUs, refer to Fig. 4 of Fenicia et al. (2016).

In order to limit the levels of decisions of the semi-distributed models, some of the aspects of the distributed models are fixed a priori, and others are left open. In particular,

– the structure chosen to represent the various HRUs is kept fixed. That is, differences between HRUs will be reflected only through the parameter values.

– The definition of HRUs is left open. In particular, we do not a priori specify which approach is used to discretize the landscape.

– The spatial discretization of the model inputs is left open. Hence, we do not decide in advance which spatial discretization of the inputs is most appropriate.

Only the fixed decision about the HRU model structure is described here, whereas the open decisions are described in the results section (Sect. 4.2.2). The spatial organization of the model structure is represented in Fig. 6, with the equations listed in Appendix A. The structure includes a snow reservoir (WR), with inputs distributed per subcatchments. Snowmelt and rainfall are input to an unsaturated reservoir (UR), which determines the portion of precipitation that produces runoff. This flux is split through a fast reservoir (FR), designed to represent the peaks of the hydrograph, proceeded by a lag function to offset the hydrograph, and a slow reservoir (SR), designed to represent baseflow. This structure was chosen to be parsimonious while general enough to reproduce typical hydrograph behaviour; it was tested in previous applications (e.g. van Esse et al., 2013; Fenicia et al., 2014, 2016), demonstrating its suitability for reproducing a wide range of catchment responses. It also resembles popular conceptual hydrological models such as HBV (Lindstrom et al., 1997) and HyMod (Wagener et al., 2001), which have been shown to have wide applicability. The model was built using the SUPERFLEX modelling framework (Fenicia et al., 2011).

### 3.2.2 Error model

As commonly done in hydrological modelling (e.g. McInerney et al., 2017), we here account for uncertainties by considering a probabilistic model of the observations $\boldsymbol{Q}(\boldsymbol{\theta}, \boldsymbol{x})$ where $\boldsymbol{\theta}$ is the vector of parameters and $\boldsymbol{x}$ the model input, which is composed of a deterministic hydrological model $\boldsymbol{h}(\boldsymbol{\theta}_{\mathrm{h}}, \boldsymbol{x})$ (illustrated in Sect. 3.2.1) and a random residual error term $\boldsymbol{E}(\boldsymbol{\theta}_{\mathrm{E}})$ that accounts for all data and model uncertainties ($\boldsymbol{\theta}_{\mathrm{h}}$ and $\boldsymbol{\theta}_{\mathrm{E}}$ represent the hydrological and error parameters):

$$z[\boldsymbol{Q}(\boldsymbol{\theta}, \boldsymbol{x}); \lambda] = z[\boldsymbol{h}(\boldsymbol{\theta}_{\mathrm{h}}, \boldsymbol{x}); \lambda] + E(\boldsymbol{\theta}_{\mathrm{E}}), \tag{4}$$

where $z[\boldsymbol{y}; \lambda]$ represents the Box–Cox transformation (Box and Cox, 1964) with parameter $\lambda$, which is used to account for heteroscedasticity (stabilize the variance). For $\lambda \neq 0$,

$$z[y_t; \lambda] = \frac{y_t^{\lambda} - 1}{\lambda}. \tag{5}$$

The residual error term is assumed to follow a Gaussian distribution with zero mean and variance $\sigma^2$:

$$E_t \sim N\left(0; \sigma^2\right). \tag{6}$$

The error model has, therefore, two parameters ($\lambda$ and $\sigma^2$); the first one was fixed to 0.5 (McInerney et al., 2017) and the second one was inferred.

This choice of error model (Gaussian noise applied to the Box–Cox transformation of the streamflow) allows for an explicit definition of the likelihood function (McInerney et al., 2017)

$$p\left(\boldsymbol{q}_{\mathrm{obs}} | \boldsymbol{\theta}_{\mathrm{h}}, \boldsymbol{\theta}_{\mathrm{E}}, x\right) = \prod_{t=1}^{T} z'\left(q_{\mathrm{obs},t} | \boldsymbol{\theta}_{\mathrm{E}}\right) f_N\left(E_t | 0; \sigma^2\right), \tag{7}$$

where T represents the length of the time series, $f_N$ is the Gaussian probability density function (PDF) and $z'(\boldsymbol{q}_{\mathrm{obs}} | \boldsymbol{\theta}_{\mathrm{E}})$ is the derivative of $z(\boldsymbol{q}_{\mathrm{obs}}, \boldsymbol{\theta}_{\mathrm{E}})$ with respect to $\boldsymbol{q}$ evaluated at the observed data $\boldsymbol{q}_{\mathrm{obs}}$. Specifying Eq. (5) for the case where $z(\boldsymbol{q}_{\mathrm{obs}}; \boldsymbol{\theta}_{\mathrm{E}})$ is defined by Eq. (5), the expression of the likelihood function becomes

$$p\left(\boldsymbol{q}_{\mathrm{obs}} | \boldsymbol{\theta}_{\mathrm{h}}, \boldsymbol{\theta}_{\mathrm{E}}, \boldsymbol{x}\right) = \prod_{t=1}^{T} q_{\mathrm{obs},t}^{(\lambda - 1)} f_N\left(E_t | 0; \sigma^2\right). \tag{8}$$

Equation (8) represents the likelihood function that is then used, together with a uniform prior distribution, to calibrate the parameters of the model as described in Sect. 3.2.3.

### 3.2.3 Calibration

Parameter calibration is performed with the objective of maximizing their posterior density. According to the Bayes equation, the posterior distribution of model parameters is expressed as the product between the prior distribution and the likelihood function; since a uniform prior is used for the parameters, this is equivalent to maximizing the likelihood function in the defined parameter space; the optimization procedure is performed with a multi-start quasi-Newton method (Kavetski et al., 2007) with 20 independent searches. We empirically established that with models of our complexity (about 10 parameters), 20 independent searches provide good confidence that a global optimum will be found.

The evaluation of the model ability to reproduce streamflow is carried out in space–time validation (see also Fenicia et al., 2016). For this purpose, the time domain is divided into two periods of 12 years each (from 1 September 1981 to 1 September 1993 and from 1 September 1993 to 1 September 2005) and the subcatchments are split into two groups (A and B), according to a spatial alternation (subcatchment in group A flows into a subcatchment in group B that flows into one in group A and so on); the subcatchments belonging to group A are Andelfingen, Herisau, Jonschwil, St. Gallen, and Wängi and the ones in group B are Appenzell, Frauenfeld, Halden, Mogelsberg, and Mosnang. This method implies a division of the space–time domain into four quadrants, such that the model can be calibrated in one quadrant and validated in the other three. For space–time validation, the model is calibrated using each group of subcatchment and period and validated using the other group of subcatchment and period. That is, the model calibrated using group A and period 1 was validated using group B and period 2, and so on for the other three combinations of subcatchments and groups. The model output in the four space–time validation periods is then combined to calculate model performance using various indicators (see Sect. 3.2.4). Results are presented for space–time validation, which represents the most challenging test of model performance.

### 3.2.4 Performance assessment

Model performance is assessed using the following metrics.

1. Time series metrics, which evaluate the ability to reproduce streamflow time series. The metrics used for this assessment are the following.

   – Normalized log likelihood **CE6** ($F_{\mathrm{LL}}$), that is, the logarithm of Eq. (8) normalized by the number of time steps present in the time series. This metric corresponds to the objective function used for model optimization. It can be observed that, since $\lambda$ is fixed at 0.5 in the Box–Cox transformation, model calibration is equivalent to maximizing the Nash–Sutcliffe efficiency ($F_{\mathrm{NS}}$) calculated with the square root of the streamflow. $F_{\mathrm{LL}}$ is not bounded, but a higher value means a better match between two time series since, in this case, the absolute value of the residual is smaller and, thus, their PDF higher.

– Nash–Sutcliffe efficiency:

$$F_{\mathrm{NS}}\left(\boldsymbol{q}_{\mathrm{obs}}, \boldsymbol{q}_{\mathrm{sim}}\right) = 1 - \frac{\sum\limits_{t=1}^{T}\left(q_{\mathrm{sim},t} - q_{\mathrm{obs},t}\right)^2}{\sum\limits_{t=1}^{T}\left(q_{\mathrm{obs},t} - \overline{\boldsymbol{q}_{\mathrm{obs}}}\right)^2}, \quad (9)$$

which is often used in hydrological applications and provides a sense of the general quality of the simulations. $F_{\mathrm{NS}}$ is bounded between $-\infty$ and 1, with 1 meaning a perfect match.

2. Signature metrics, which determine the ability to reproduce the streamflow signatures ($\zeta$) selected using the procedure illustrated in Sect. 3.1.2. The agreement between simulated and observed signatures is assessed using two metrics: Spearman's rank correlation ($r$) and the normalized root mean square error:

$$F_{\mathrm{RMSE}} = \frac{\sqrt{\dfrac{\sum\limits_{t=1}^{T}\left(q_{\mathrm{sim},t} - q_{\mathrm{obs},t}\right)^2}{T}}}{\dfrac{\sum\limits_{t=1}^{T} q_{\mathrm{obs},t}}{T}}. \quad (10)$$

TS3 While $r$ assesses how well the simulated signatures can be described using a monotonic function, $F_{\mathrm{RMSE}}$ imposes a more stringent requirement, as it assesses how well the simulated and observed signatures line up on the diagonal line.

The use of multiple metrics for assessing model performance enables a comprehensive assessment of various characteristics of the simulations. Time series metrics are designed to appraise the general quality of the model fit. Signatures, instead, are designed to highlight selected characteristics of the data at the expense of others.

# 4 Results and interpretation

## 4.1 Influence factors on the spatial variability of streamflow signatures

This section illustrates the results of the correlation analysis complemented by expert judgement aimed to identify influence factors that control the spatial variability of streamflow signatures; Sect. 4.1.1 presents the results of the selection of meaningful statistics; Sect. 4.1.2 identifies climate and landscape indices controlling streamflow signatures and presents consequences for model development.

### 4.1.1 Selection of meaningful streamflow signatures, climatic indices, and catchment indices

The streamflow signatures defined in Sect. 3.1.1 were calculated for each subcatchment and the values are shown in Table 3 together with the coefficient of variation. All the signatures have a coefficient of variability bigger than the threshold value of 5 %, with the most variable signature being $\zeta_{\mathrm{LQF}}$ (71 %) and the least variable $\zeta_{\mathrm{HQD}}$ (6 %). Therefore, none of these signatures was discarded.

Figure 2 shows the correlations between the streamflow signatures: the lower triangle contains Spearman's rank correlation and the upper triangle the $p$ value CE7 associated with the correlations. Based on correlations and on its interpretation, a subset of $\zeta$ can be defined as follows.

– $\zeta_Q$, $\zeta_{\mathrm{RR}}$ and $\zeta_{\mathrm{EL}}$ are strongly correlated ($r > 0.72$). We retained $\zeta_Q$ and discarded $\zeta_{\mathrm{RR}}$ and $\zeta_{\mathrm{EL}}$ because both contain climatic information (precipitation) in their definition.

– $\zeta_{\mathrm{BFI}}$ and $\zeta_{\mathrm{FDC}}$ are strongly correlated ($r = -0.77$). We decided to retain $\zeta_{\mathrm{BFI}}$ as it is of easier interpretation (it is a proxy for the importance of groundwater flow, which is a potentially important process for the subsequent model development).

– $\zeta_{\mathrm{HFD}}$ was kept because it measures the seasonality of the streamflow. Note that $\zeta_{\mathrm{HFD}}$ is strongly correlated with $\zeta_Q$ ($r = 0.88$). However, they reflect different properties of the hydrograph. In particular, $\zeta_{\mathrm{HFD}}$ can be an useful indicator of the effect of snow-related processes.

– $\zeta_{Q5}$ and $\zeta_{\mathrm{HQD}}$ were retained because they have low correlation ($r < 0.71$) with the other selected signatures and because the first represents low flows and the second high flows;

– $\zeta_{Q95}$, $\zeta_{\mathrm{HQF}}$, $\zeta_{\mathrm{LQD}}$, and $\zeta_{\mathrm{LQF}}$ were discarded because they all show correlations with the selected signatures.

In summary, the original set of streamflow signatures was reduced to a set of five meaningful signatures, which will be used in the subsequent analyses: average daily streamflow ($\zeta_Q$), baseflow index ($\zeta_{\mathrm{BFI}}$), half streamflow period ($\zeta_{\mathrm{HFD}}$), 5th percentiles of the streamflow ($\zeta_{Q5}$), and duration of high-flow events ($\zeta_{\mathrm{HQD}}$).

In terms of climatic indices, Table 4 shows their values together with the coefficient of variation. It can be seen that there are some indices that show very little or no variation at all and, therefore, they could already be excluded from the subsequent correlation analysis; they are $\psi_{\mathrm{HPD}}$ (1 %), $\psi_{\mathrm{HPS}}$ (0 %), $\psi_{\mathrm{LPF}}$ (4 %), $\psi_{\mathrm{LPD}}$ (3 %), and $\psi_{\mathrm{LPS}}$ (0 %).

Figure 3 shows the correlation between the remaining indices. It can be observed that they all have strong internal correlation ($r > 0.71$). For this reason it was decided to retain only $\psi_{\mathrm{P}}$ and $\psi_{\mathrm{FS}}$, as they have lower correlation. The former represents an important term of the water budget, and the latter captures snow dynamics.

Table 5 shows the values of the catchment characteristics considered in this study. All of them have a coefficient of

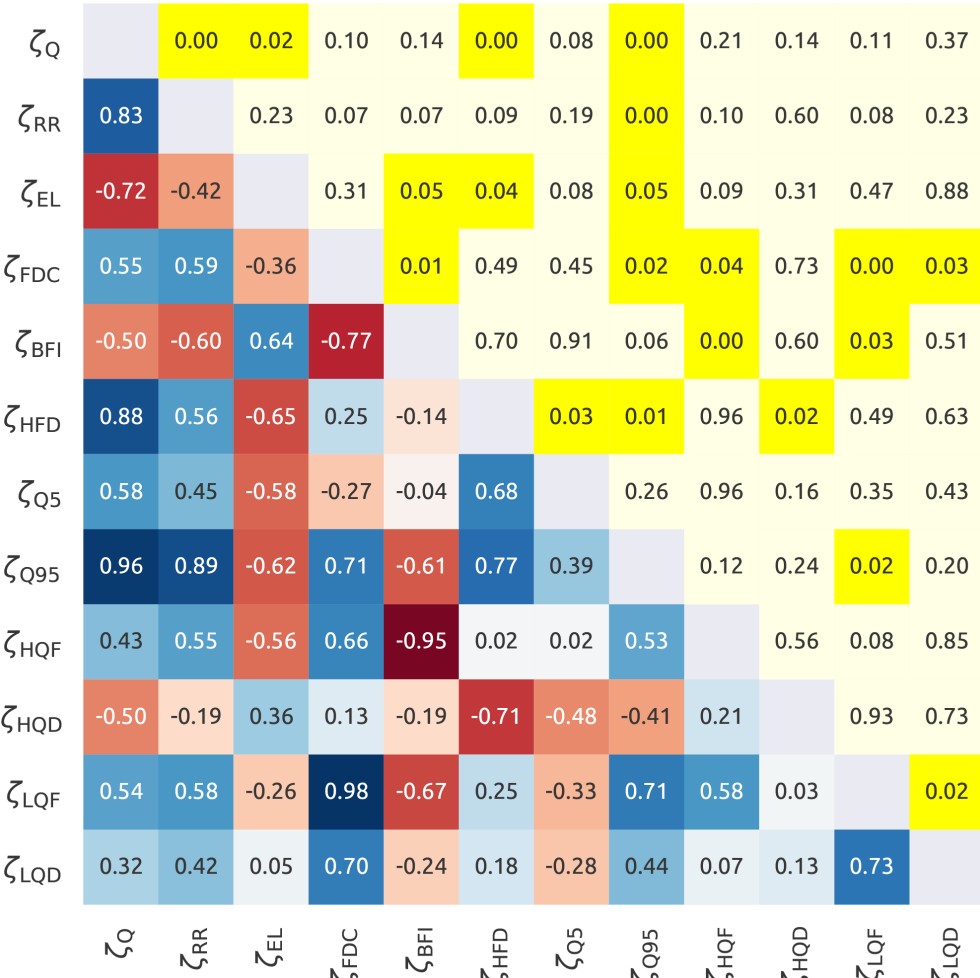

**Figure 2.** Internal correlation between the streamflow signatures. The lower triangle shows Spearman's rank score with the red colour that indicates negative correlations and the blue that indicates positive correlations. The upper triangle reports the corresponding $p$ values, where yellow colour indicates a statistically significant correlation ($p$ value $< 0.05$). The symbols used in the figure are reported in Table 2.

**Table 4.** Values of the climatic indices. The names of the subcatchments are abbreviated using the first three letters and the symbols are reported in Table 2. The last column contains the coefficient of variation of each index.

| | | | | | Subcatchment | | | | | | |
|---|---|---|---|---|---|---|---|---|---|---|---|
| | And | App | Fra | Hal | Her | Jon | Mog | Mos | StG | Wän | CV |
| $\psi_P$ | 3.91 | 5.15 | 3.36 | 4.38 | 4.13 | 4.64 | 4.57 | 4.04 | 4.80 | 3.62 | 0.13 |
| $\psi_{PET}$ | 1.60 | 1.37 | 1.70 | 1.55 | 1.61 | 1.54 | 1.57 | 1.69 | 1.49 | 1.71 | 0.07 |
| $\psi_{AI}$ | 0.41 | 0.27 | 0.50 | 0.35 | 0.39 | 0.33 | 0.34 | 0.42 | 0.31 | 0.47 | 0.19 |
| $\psi_{FS}$ | 0.04 | 0.21 | 0.04 | 0.05 | 0.09 | 0.15 | 0.13 | 0.09 | 0.13 | 0.05 | 0.57 |
| $\psi_{HPF}$ | 15.21 | 14.38 | 17.67 | 14.58 | 15.82 | 14.54 | 14.58 | 16.13 | 14.31 | 17.50 | 0.08 |
| $\psi_{HPD}$ | 1.20 | 1.17 | 1.17 | 1.18 | 1.22 | 1.20 | 1.19 | 1.22 | 1.17 | 1.19 | 0.01 |
| $\psi_{HDS}$ | Summer | Summer | Summer | Summer | Summer | Summer | Summer | Summer | Summer | Summer | 0.00 |
| $\psi_{LPF}$ | 201.67 | 195.79 | 216.83 | 198.54 | 205.04 | 197.21 | 198.92 | 205.75 | 197.69 | 213.17 | 0.04 |
| $\psi_{LPD}$ | 3.57 | 3.50 | 3.83 | 3.50 | 3.63 | 3.51 | 3.51 | 3.66 | 3.51 | 3.76 | 0.03 |
| $\psi_{LPS}$ | Autumn | Autumn | Autumn | Autumn | Autumn | Autumn | Autumn | Autumn | Autumn | Autumn | 0.00 |

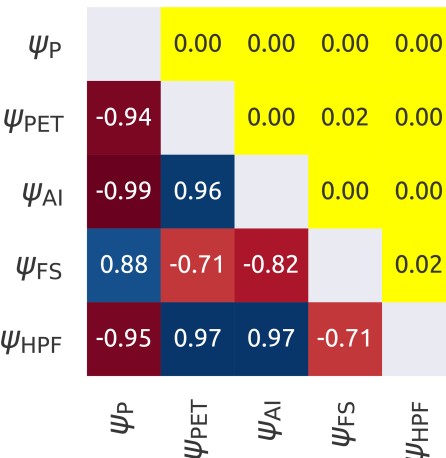

**Figure 3.** Internal correlation between the climatic indices. The lower triangle shows Spearman's rank score with the red colour that indicates negative correlations and the blue that indicates positive correlations. The upper triangle reports the corresponding $p$ values, where yellow colour indicates a statistically significant correlation ($p$ value $< 0.05$). The symbols used in the figure are reported in Table 2.

variation larger than the minimum threshold of 5 %. Therefore, none of them was excluded based on this criterion. The second criterion for the pre-exclusion of the catchments characteristics, consisting in removing $\xi$ occupying less than 5 % of the subcatchments, led to the suppression of $\xi_{LC}$ (which occupies 4 % of the subcatchment).

Figure 4 shows the correlations between catchment characteristics; in many cases the high correlation is due to the fact that many indices are complementary (e.g. different types of geology). The following $\xi$ were selected (one index per class):

- $\xi_A$ because it is low correlated with the other features;

- $\xi_{TE}$ and $\xi_{TA_s}$ in representation of the topography;

- $\xi_{LF}$ for the land use;

- $\xi_{SD}$ representing the soil characteristics;

- $\xi_{GC}$ for the geology.

In summary, the original set of catchment indices was reduced to a set of six indices.

### 4.1.2 Selection of controlling factors on streamflow signatures

Figure 5 reports the results of Spearman's correlation between climatic indices plus catchment characteristics and streamflow signatures. Panel (a) contains Spearman's rank coefficients and panel (b) shows $p$ values associated with them.

The following results can be noted.

- The three statistics average precipitation ($\psi_P$), fraction of snow ($\psi_{FS}$), and average elevation ($\xi_{TE}$) correlate strongly with average streamflow ($\zeta_Q$) and seasonality ($\zeta_{HFD}$) ($r > 0.64$ and $p$ value $< 0.05$). This correlation can be interpreted as follows: subcatchments with high elevation ($\xi_{TE}$) tend to have higher precipitation ($\psi_P$) due to orographic effects, which leads to higher streamflow ($\zeta_Q$). They also tend to have more snow ($\psi_{FS}$) due to lower temperatures, which influences the seasonality ($\zeta_{HFD}$).

- There are then some catchment characteristics that have no correlation ($r < 0.45$) with the streamflow signatures (catchment area ($\xi_A$) and land use ($\xi_{LF}$)) or limited correlation (aspect ($\xi_{TA_s}$) and deep soil ($\xi_{SD}$), with $r < 0.64$.

- The consolidated geology ($\xi_{GC}$) presents a strong correlation ($r = -0.87$) only with the baseflow index ($\zeta_{BFI}$); that is not captured by the other indices.

- The streamflow signatures of low and high flows ($\zeta_{Q5}$ and $\zeta_{HQD}$) cannot be explained by any index, with little correlation only with $\psi_P$ and $\xi_{TE}$ ($r < 0.60$) that is not sufficient to reach a $p$ value lower than 0.05.

These results are the premise for designing meaningful model experiments.

### 4.2 Hypotheses for model building

This section interprets the results found in Sect. 4.1.2 and formulates some hypotheses regarding the hydrological functioning of the catchment (Sect. 4.2.1). Section 4.2.2, then, presents the model alternatives designed for testing those hypotheses.

### 4.2.1 Hypotheses on catchment functioning

The results of the correlation analysis can be interpreted to formulate the following hypotheses regarding the drivers of streamflow variability.

1. The precipitation is the first driver of the differences in the water balance of the subcatchments. The effect of topographic variability manifests itself primarily as an influence on precipitation (amount and type). Accounting for variability of precipitation therefore implicitly reflects such effect of topography on the hydrograph, since some inputs were interpolated taking into account the effect of the elevation (Sect. 2). Other phenomena potentially altering the water balance (e.g. regional groundwater flow) do not have a significant role and should not be considered.

2. Snow-related processes (e.g. amount of snow, timing of snowmelt) control differences in streamflow seasonality

**Table 5.** Values of the subcatchment characteristics. The names of the subcatchments are abbreviated using the first three letters and the symbols are reported in Table 2. The last two columns contain the coefficient of variation and the maximum value of each signature.

| | Subcatchment | | | | | | | | | | CV | MAX |
|---|---|---|---|---|---|---|---|---|---|---|---|---|
| | And | App | Fra | Hal | Her | Jon | Mog | Mos | StG | Wän | | |
| $\xi_A$ | 1701 | 74.46 | 213.34 | 1085 | 16.72 | 493.0 | 88.11 | 3.19 | 261.1 | 78.96 | 1.40 | 1701 |
| $\xi_{TE}$ | 768 | 1250 | 591 | 908 | 831 | 1020 | 954 | 797 | 1039 | 650 | 0.22 | 1250 |
| $\xi_{TS_m}$ | 13.32 | 25.23 | 9.70 | 16.87 | 15.44 | 20.66 | 19.77 | 15.68 | 19.72 | 12.49 | 0.27 | 25.23 |
| $\xi_{TS_s}$ | 0.47 | 0.81 | 0.33 | 0.62 | 0.69 | 0.77 | 0.79 | 0.71 | 0.73 | 0.45 | 0.26 | 0.81 |
| $\xi_{TA_s}$ | 0.25 | 0.22 | 0.23 | 0.23 | 0.21 | 0.23 | 0.24 | 0.40 | 0.24 | 0.21 | 0.23 | 0.40 |
| $\xi_{TA_n}$ | 0.32 | 0.35 | 0.33 | 0.32 | 0.33 | 0.32 | 0.31 | 0.24 | 0.33 | 0.32 | 0.09 | 0.35 |
| $\xi_{TA_{ew}}$ | 0.43 | 0.43 | 0.44 | 0.44 | 0.46 | 0.44 | 0.45 | 0.36 | 0.43 | 0.47 | 0.07 | 0.47 |
| $\xi_{SM}$ | 1.30 | 0.56 | 1.48 | 1.10 | 1.32 | 0.93 | 1.17 | 1.00 | 1.03 | 1.35 | 0.23 | 1.48 |
| $\xi_{SD}$ | 0.40 | 0.04 | 0.49 | 0.25 | 0.41 | 0.13 | 0.28 | 0.00 | 0.26 | 0.36 | 0.63 | 0.49 |
| $\xi_{LF}$ | 0.26 | 0.25 | 0.28 | 0.27 | 0.21 | 0.31 | 0.34 | 0.18 | 0.27 | 0.30 | 0.17 | 0.34 |
| $\xi_{LC}$ | 0.04 | 0.00 | 0.04 | 0.03 | 0.03 | 0.01 | 0.01 | 0.01 | 0.01 | 0.04 | 0.79 | 0.04 |
| $\xi_{LU}$ | 0.08 | 0.03 | 0.10 | 0.06 | 0.15 | 0.04 | 0.03 | 0.03 | 0.05 | 0.10 | 0.63 | 0.15 |
| $\xi_{LP}$ | 0.60 | 0.59 | 0.57 | 0.61 | 0.61 | 0.61 | 0.62 | 0.77 | 0.63 | 0.55 | 0.09 | 0.77 |
| $\xi_{GA}$ | 0.06 | 0.01 | 0.09 | 0.03 | 0.00 | 0.02 | 0.02 | 0.00 | 0.01 | 0.11 | 1.05 | 0.11 |
| $\xi_{GC}$ | 0.59 | 0.92 | 0.54 | 0.73 | 0.88 | 0.90 | 0.92 | 1.00 | 0.88 | 0.63 | 0.20 | 1.00 |
| $\xi_{GU}$ | 0.35 | 0.07 | 0.36 | 0.23 | 0.12 | 0.07 | 0.06 | 0.00 | 0.10 | 0.26 | 0.79 | 0.36 |

between subcatchments. Hence, the model needs to account for snow-related processes and their spatial variability.

3. Geology exerts an important control on the partitioning between quickflow and baseflow. Hence, the model should distinguish the different response behaviours of distinct geological areas.

4. The other catchment characteristics (e.g. soil, vegetation) show little or no correlation CE8 with the streamflow signatures, and therefore they should not be considered if the idea is to keep the model as simple as possible.

The streamflow signatures $\zeta_{Q5}$ and $\zeta_{HQD}$, which have been selected as part of the analysis shown in Sect. 4.1.1, do not manifest a strong correlation with any of the indices ($r$ is always less than 0.60), meaning that the identification of their potential controls is not obvious with the chosen approach. Hence, we have not been able to build model hypotheses that specifically target those signatures. As a result, we expect that the chosen models will not excel and will perform similarly in reproducing these signatures. The model comparisons used to test the four hypotheses listed above are described in Sect. 4.2.2.

### 4.2.2 Modelling experiments for testing the hypotheses

Using the model structure described in Sect. 3.2.1, four model configurations were compared by varying the number and the definition of the HRUs, and changing the structure of the HRUs (Fig. 6). The objective of the experiments was to test the hypotheses 1–4 in Sect. 4.2.1 using semi-distributed hydrological models.

For all the models, the meteorological inputs (precipitation, PET, temperature) are aggregated at the subcatchment scale. Based on the first hypothesis, we assume that this discretization is sufficient to capture the regional difference in water balance between subcatchments. This hypothesis is tested with model M0, with uniform parameters in the catchment (i.e. a single HRU) and distributed precipitation input. This model does not consider snow processes. We expect that this model will be able to reproduce differences in streamflow averages between subcatchments.

The second hypothesis (snow controls seasonality) is tested with model M1. Relative to M0, M1 accounts for snow processes, represented by a simple degree-day snow module (see Kavetski and Kuczera, 2007), with inputs (temperature) distributed per subcatchment. We expect that this model will be able to reproduce differences in streamflow seasonality between subcatchments.

The third hypothesis (geology controls baseflow) is tested with model M2. Relative to M1, M2 considers two HRUs, defined based on geology type. One HRU contains the areas with consolidated geology, while the other HRU contains the rest of the catchment (unconsolidated and alluvial geology together). We expect that M2 will be able to reproduce differences in the baseflow index between subcatchments.

The fourth hypothesis (other catchment characteristics should not be considered if the idea is to keep the model as simple as possible) is exemplified by model M3. M3 is analogous to M2 in terms of complexity, but the HRUs are based on catchment characteristics that did not show correlation with the streamflow signatures. Among those charac-

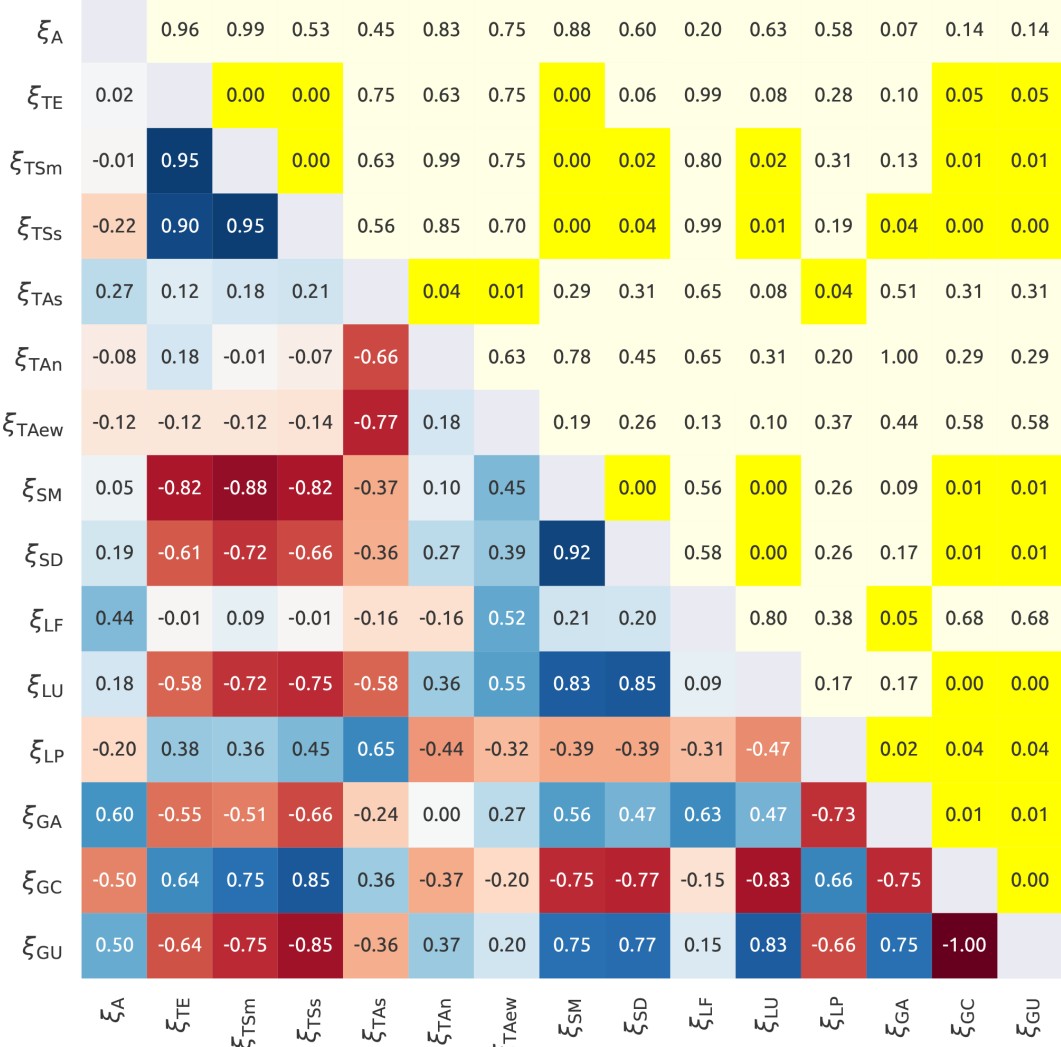

**Figure 4.** Internal correlation between the catchment characteristics. The lower triangle shows Spearman's rank score with the red colour that indicates negative correlations and the blue that indicates positive correlations. The upper triangle reports the corresponding $p$ values, where yellow colour indicates a statistically significant correlation ($p$ value $< 0.05$). The symbols used in the figure are reported in Table 2.

teristics, we have selected land use and considered an HRU based on forest and crops and the second one that occupies the rest of the catchment. This model is as complex as M2 (therefore it is more complex than M1); hence it has the same dimensions of flexibility to fit the data. However, since the structure of this model does not incorporate the cause–effect relationships derived from the signature analysis, we expect that its predictive performance will be poorer than M2.

The total number of the calibrated parameters depends on the number of HRUs and on the structure used to represent them: it was 8 for M0, 9 in M1, and 13 in M2 and M3, of which 5 parameters are common to all the HRUs (Fig. 6 and Table A1); these parameters are $C_e$ that governs the evapotranspiration, $t_{rise}^{OL}$ and $t_{rise}^{IL}$ that control the routing in the river network, $k_{WR}$ that regulates the outflow of the snow reser-

voir, and $S_{max}^{UR}$ that determines the behaviour of the unsaturated reservoir.

### 4.3 Modelling results

The models presented in Sect. 4.2.2 are evaluated in terms of hydrograph metrics (Sect. 4.3.1) and signature metrics (Sect. 4.3.2).

### 4.3.1 Model performance in terms of hydrograph metrics

Figure 7a shows the values of the likelihood function (corresponding to the calibration objective function) for the four models in calibration and validation. It can be observed that M0 is, by far, the worst model, with the lowest value

(a)

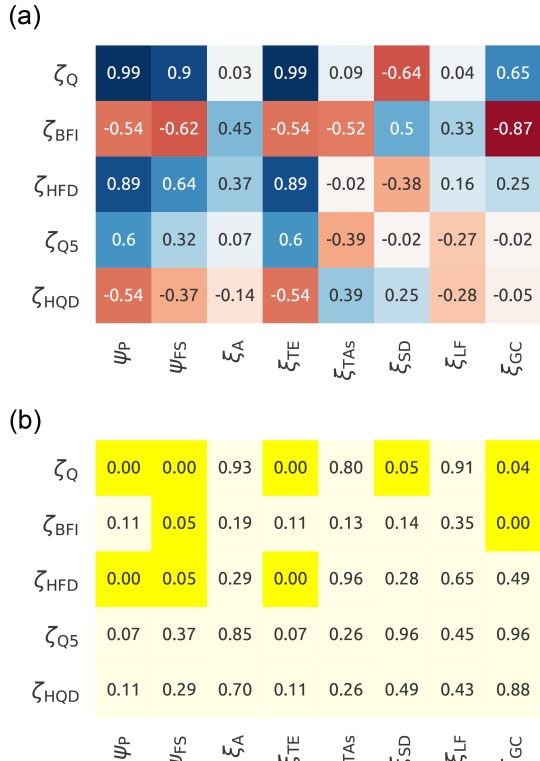

(b)

**Figure 5.** Correlation between the selected streamflow signatures (rows) and the selected climatic indices and catchment characteristics (columns). Panel **(a)** shows Spearman's rank score with the red colour that indicates negative correlations and the blue that indicates positive correlations. Panel **(b)** reports the corresponding $p$ values, where yellow colour indicates a statistically significant correlation ($p$ value $< 0.05$). The symbols used in the figure are reported in Table 2.

of the likelihood function. Regarding the other three models, it can be seen that, during calibration, M1, which has the lowest number of calibration parameters, has the lowest performance, whereas M2 and M3 have higher and similar likelihood values. This behaviour persists in time validation, with M2 and M3 that outperform M1. In space and space–time validation, however, M3 has the lowest likelihood value of the three, whereas M1 and M2 limit their decrease in performance, ranking respectively second and first in terms of optimal likelihood value.

The likelihood function represents an aggregate metric of model performance; in order to get a sense of appreciation of model fit on individual subcatchments, Fig. 7b reports the values of Nash–Sutcliffe efficiency in space–time validation for each of the subcatchments. On average, M2 has the best performance of all models ($F_{NS} = 0.79$), followed by M1 ($F_{NS} = 0.78$), M3 ($F_{NS} = 0.77$), and M0 ($F_{NS} = 0.68$). M3 and M0 have the highest variability of performance, with $F_{NS}$ values between 0.58 and 0.86 and between 0.59 and 0.81. M1 and M2 have similar spread of $F_{NS}$ values,

ranging from 0.69 to 0.85 for M1 and from 0.73 to 0.87 for M2. Therefore, M1 and M2 have a more stable performance across subcatchments than M3. M3 obtains a significantly worse performance than the other three models on Mosnang, where it reaches a $F_{NS}$ value of 0.58 (M0, M1, and M2 have values of 0.62, 0.69, and 0.73 respectively).

It can also be observed that M2 is generally better than M1, with $F_{NS}$ values that are higher or approximately equal except for subcatchments Andelfingen and Halden, where the $F_{NS}$ is slightly worse (however still higher than 0.80). M3 is clearly better than M1 in Andelfingen, Frauenfeld, and Wängi, and clearly worse in Herisau and Mosnang. In particular, in Mosnang (the smallest basin), M3 reaches the worst performance of all the models on all the subcatchments.

Regarding M0, it is interesting to observe that it has the worst performance (among all the subcatchments) in Appenzell, which is the subcatchment that is most affected by snow ($\psi_{FS} = 0.21$), while it reaches a performance similar to M1 in Frauenfeld and Wängi, which are two subcatchments with almost no snow.

### 4.3.2 Model performance in terms of signature metrics

Figure 8 compares the ability of M0 and M1 to capture the signatures representing average streamflow ($\zeta_Q$) and seasonality ($\zeta_{HFD}$). The analysis is presented for space–time validation and, for $\zeta_{HFD}$, focuses only on the four subcatchments that are most affected by the snow ($\psi_{FS} > 0.10$), to emphasize the differences between the results of the two models. Each colour represents a different subcatchment and each dot a year; the red dashed line has a 45° slope and represents where the dots should align in case of perfect simulation results. The normalized root mean square error and Spearman's rank score are also reported. It is important to stress that the models have not been calibrated using any of the signatures as an objective function, which therefore represent independent evaluation metrics.

It can be observed that M0 represents $\zeta_Q$ equally well as M1, with almost no difference between the two models ($r$ is 0.95 in both cases, whereas $F_{RMSE}$ is 0.11 for M0 and 0.10 for M1). Focusing on the ability to capture $\zeta_{HFD}$, it can be seen that the points corresponding to M0 all lie in the upper-left part of the plot, meaning that this model underestimates the signature values. With respect to M1, instead, the points are more aligned around the diagonal. This difference in performance is captured by the values of $F_{RMSE}$ (0.13 for M0 and 0.07 for M1) and of $r$ (0.66 for M0 and 0.85 for M1).

Figure 9 compares the observed and simulated signatures for the other three models (M1, M2, and M3). All of them are equally good in representing $\zeta_Q$ ($F_{RMSE}$ is 0.10, 0.10, and 0.11, and $r$ is 0.95, 0.96, and 0.95 for M1, M2, and M3 respectively) and $\zeta_{HFD}$ ($F_{RMSE}$ is 0.07, 0.07, and 0.05 and $r$ is 0.85, 0.84, and 0.87 for M1, M2, and M3 respectively). In all cases the cloud of points appears to be aligned to the di-

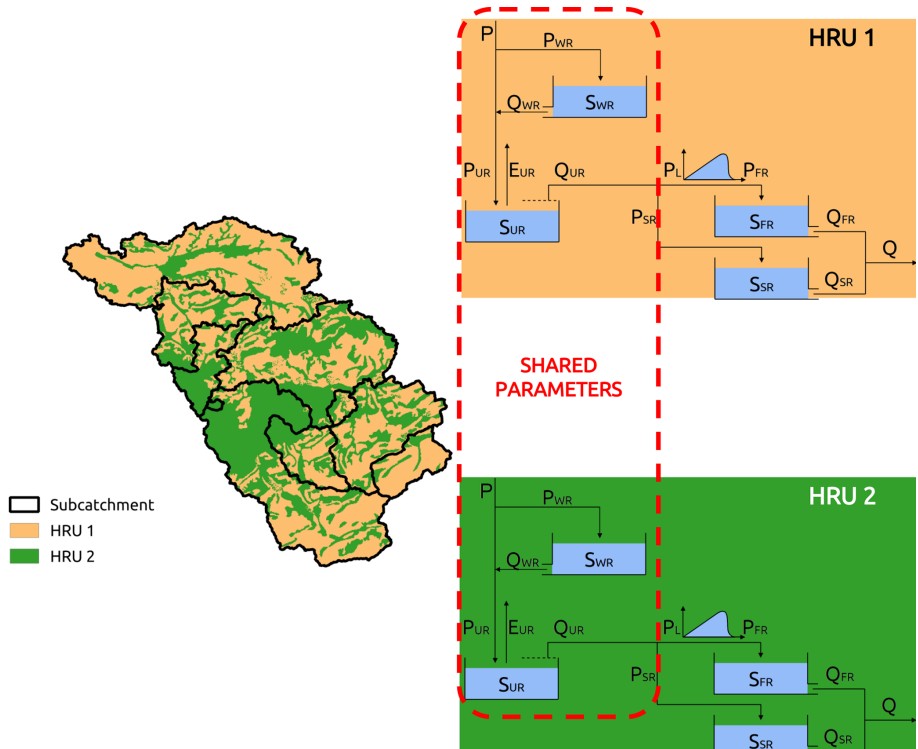

**Figure 6.** Spatial organization of the model structure: the catchment is divided into subcatchments (black lines), based on the location of the gauging stations, and HRUs (background colour), based on the catchment characteristics. All the HRUs have the same structure, but each HRU has its own parameterization except for some shared parameters. In the case of a single-HRU model (i.e. M0 and M1), the model maintains the subdivision into subcatchments but loses the subdivision into multiple HRUs.

agonal, meaning that the three models are able to capture the values of the signatures each year. Moreover, there is no sensible difference in the various models in representing those signatures.

The performance of all the models decreases for $\zeta_{Q5}$ where the models have a similar performance, with $F_{\mathrm{RMSE}}$ equal to 0.32, 0.28, and 0.33, and $r$ equal to 0.62, 0.66, and 0.61 for M1, M2, and M3 respectively. The points are still aligned along the diagonal but are quite dispersed, especially if compared with $\zeta_Q$ and $\zeta_{\mathrm{HFD}}$, meaning that the models capture the general tendency but have deficiencies in capturing the inter-annual variability.

In terms of $\zeta_{\mathrm{BFI}}$, M2 performs clearly better than the other models. It is the only model that is able to represent this signature, with $F_{\mathrm{RMSE}} = 0.07$, $r = 0.83$, and the points that align compactly with the diagonal. The other two models have a lower performance ($F_{\mathrm{RMSE}}$ equal to 0.11 and 0.10, and $r$ equal to 0.31 and 0.52 for M1 and M3 respectively), with points that are quite dispersed and align almost vertically, implying that the simulated values have a range of variability that is definitely smaller than the observed data.

Figure 10 shows the comparison between observed and simulated $\zeta_{\mathrm{HQD}}$; since this signature requires a long time window to be computed, it is not calculated year by year (as

done with the other signatures) but as an aggregated value over the 24 years. In terms of performance, M2 still remains the best among the three models, with $F_{\mathrm{RMSE}}$ of 0.09 and $r$ of 0.69; in second place comes M1, which outperforms M2 in terms of $r$ (0.77) but has a higher $F_{\mathrm{RMSE}}$ (0.19), meaning that M1 has the points that are more aligned but on a line that is farther from the diagonal compared to M2; M3, finally, has a bad performance, with high $F_{\mathrm{RMSE}}$ (0.18) and low $r$ (0.48). All the models tend to slightly overestimate the duration of high-flow events with most of the points that lie on the right-hand side of the diagonal.

## 4.4 Hypotheses testing

The results of the hydrological model experiments appear to support our general hypothesis that only models that account for the influence factors that affect the streamflow signatures are able to reproduce streamflow spatial variability (see Sect. 4.2.1). This provides confidence that those models are a realistic representation of dominant processes in the catchment.

The implications of the modelling results with respect to the evaluation of the four hypotheses are explained as follows.

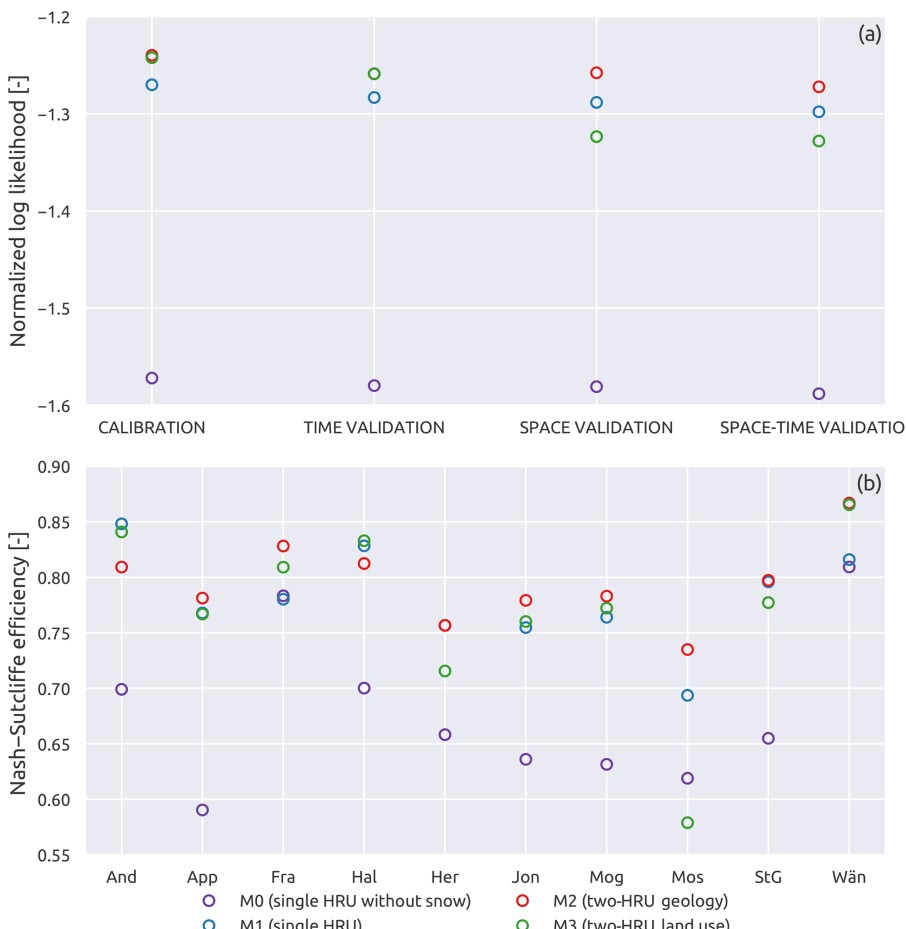

**Figure 7.** Normalized log likelihood **(a)** and Nash–Sutcliffe efficiency **(b)** for the three model configurations. **(a)** reports the variation between calibration and validation of the average of the 10 subcatchments; **(b)** shows the variation between subcatchments during space–time validation.

1. *Hypothesis 1: precipitation is the first driver of differences in the water balance.* The good performance of model M0 in the representation of the mean annual streamflow ($\zeta_Q$) suggests that accounting for the spatial heterogeneity of the precipitation alone is sufficient to achieve a good representation of the annual water balance. More complex models, with more HRUs, processes, and parameters, while resulting in an overall improvement of time series metrics, do not result in any improvement in simulating the water balance signature $\zeta_Q$.

2. *Hypothesis 2: snow-related processes control differences in streamflow seasonality.* The improvement in the representation of the streamflow seasonality $\zeta_{HFD}$ by M1 can be largely attributed to the (spatially variable) effect of snow accumulation and melting. More complex models (M2 and M3) do not demonstrate an improvement in this signature, indicating that the structural differences between these models do not have an influence on this signature.

3. *Hypothesis 3: geology controls the partitioning between quickflow and baseflow.* The ability of M2 to match the signature $\zeta_{BFI}$, which quantifies the separation between quickflow and baseflow, much better than the other models, supports the hypothesis that geology has a strong control on the partitioning between quickflow and baseflow. M2 is also the model with the average best performance in terms of streamflow metrics.

4. *Hypothesis 4: characteristics that do not show correlations do not influence streamflow variability.* The overall lower performance of M3 compared to M2, in terms of both signatures and streamflow metrics, reassures us that the relatively good results of M2 are not just due to increasing complexity and confirms that adding characteristics that do not show correlations does not improve the representation of spatial variability.

In summary, distributing the inputs in space and accounting for the spatial distribution of snow-related processes are sufficient to get good performance metrics of water balance

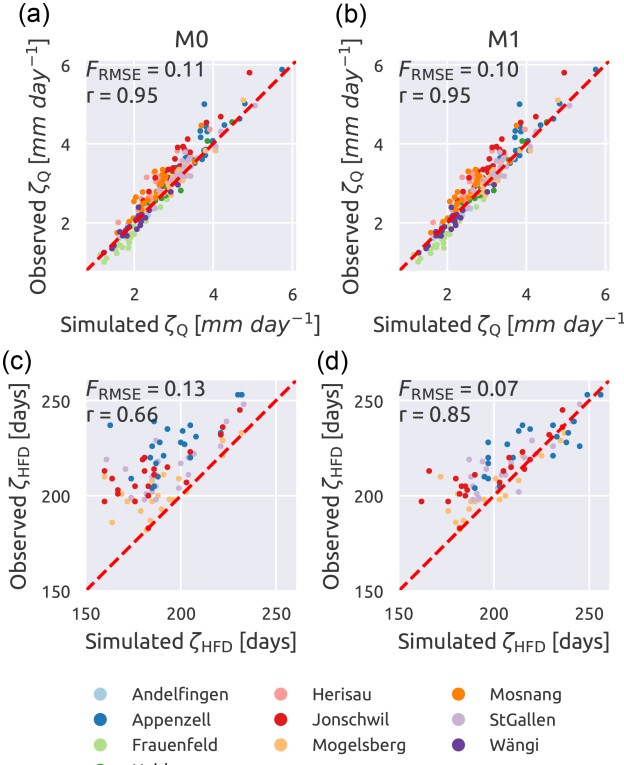

**Figure 8.** Influence of the model structure on the representation of the average streamflow ($\zeta_Q$) and the mean half streamflow day ($\zeta_{HFD}$). Single-HRU model without snow reservoir in (**a, c**) (M0) and single-HRU model with snow reservoir in (**b, d**) (M1). Each dot represents a year and each colour a subcatchment. For $\zeta_{HFD}$, only the four subcatchments with the fraction of snow ($\psi_{FS}$) larger than 10 % are plotted. The red dashed line has a 45° slope and indicates where all points should align in case of a perfect match. Spearman's rank score ($r$) is also reported.

and seasonality, confirming the fact that only the precipitation rate and the partitioning between rainfall and snow are the first controls on these hydrograph characteristics. However, in order to capture other important characteristics of the hydrograph, described by signatures such as $\zeta_{BFI}$, the discretization of the catchment in HRUs is necessary. This discretization has to be carefully made and a preliminary analysis to understand dominant influence factors on signatures can help in this decision. As shown in Fig. 9, if such discretization uses landscape characteristics that are not strongly correlated with the signatures (e.g. land use), the results are worse than if we choose characteristics that show a strong correlation with signatures (e.g. geology). This means that M2 is capable of capturing the signatures not just because it is more complex than M1, but because it incorporates the causality link between the geology and the streamflow signatures into its structure.

## 5 General discussion

Explaining the spatial variability observed in catchment response is a major focus of catchment hydrology and a central theme in classification studies (e.g. McDonnell and Woods, 2004; Wagener et al., 2007). A common approach for interpreting the spatial variability of catchment response is to identify relationships between climatic or catchment characteristics and streamflow signatures. This is typically done through correlation-based analyses (e.g. Lacey and Grayson, 1998; Bloomfield et al., 2009), which however carry the limitations that correlation does not always imply causality and that the presence of multiple correlated variables can obscure process interpretation.

In this study, we combine a correlation analysis for identifying the dominant influence factors on streamflow signatures with hydrological modelling by using the interpretation of the first analysis as an inspiration for generating testable model hypotheses. The combination of correlation analysis on streamflow signatures and hydrological modelling is beneficial because on the one hand, the speculations on dominant processes resulting from the correlation analyses can be verified in the modelling process. Specifically, we developed model experiments to test the influence of precipitation spatial distribution on streamflow average and seasonality and the influence of geology on quickflow vs. baseflow partitioning. On the other hand, model building benefits from the guidance resulting from the preliminary signature analysis. The construction of a distributed model requires several decisions (e.g. Fenicia et al., 2016), including how to "break up" the catchment in a meaningful way, and preliminary signature analysis can motivate some of these decisions. For example, the definition of HRUs based on geology, which was suggested by the signature analysis, resulted in models with better performance than models using HRUs defined on the basis of land use, particularly in the representation of streamflow signatures that reflect the baseflow vs. quickflow partitioning.

Although several modelling decisions were guided by data analysis, it should be noted that alternative decisions would have been similarly consistent with the data. For example, both precipitation and elevation are correlated with average streamflow, and geology, topography, and soil-type characteristics are correlated between each other and with the baseflow index (Sect. 4.1.2 and Fig. 5). The correlation of catchment characteristics (e.g. geology, soil, and topography) can be attributed to the fact that they have evolved together in the shaping of the catchment morphology (e.g. mountainous regions have impervious topography with shallower soil and, for these reasons, are less suitable for human activities, influencing land use). The decisions on which variables are chosen to reflect a causality link are not always obvious from correlation analysis alone, and they require expert judgement, which is necessarily subjective. Although subjectivity is difficult to avoid, it is important to be transparent about

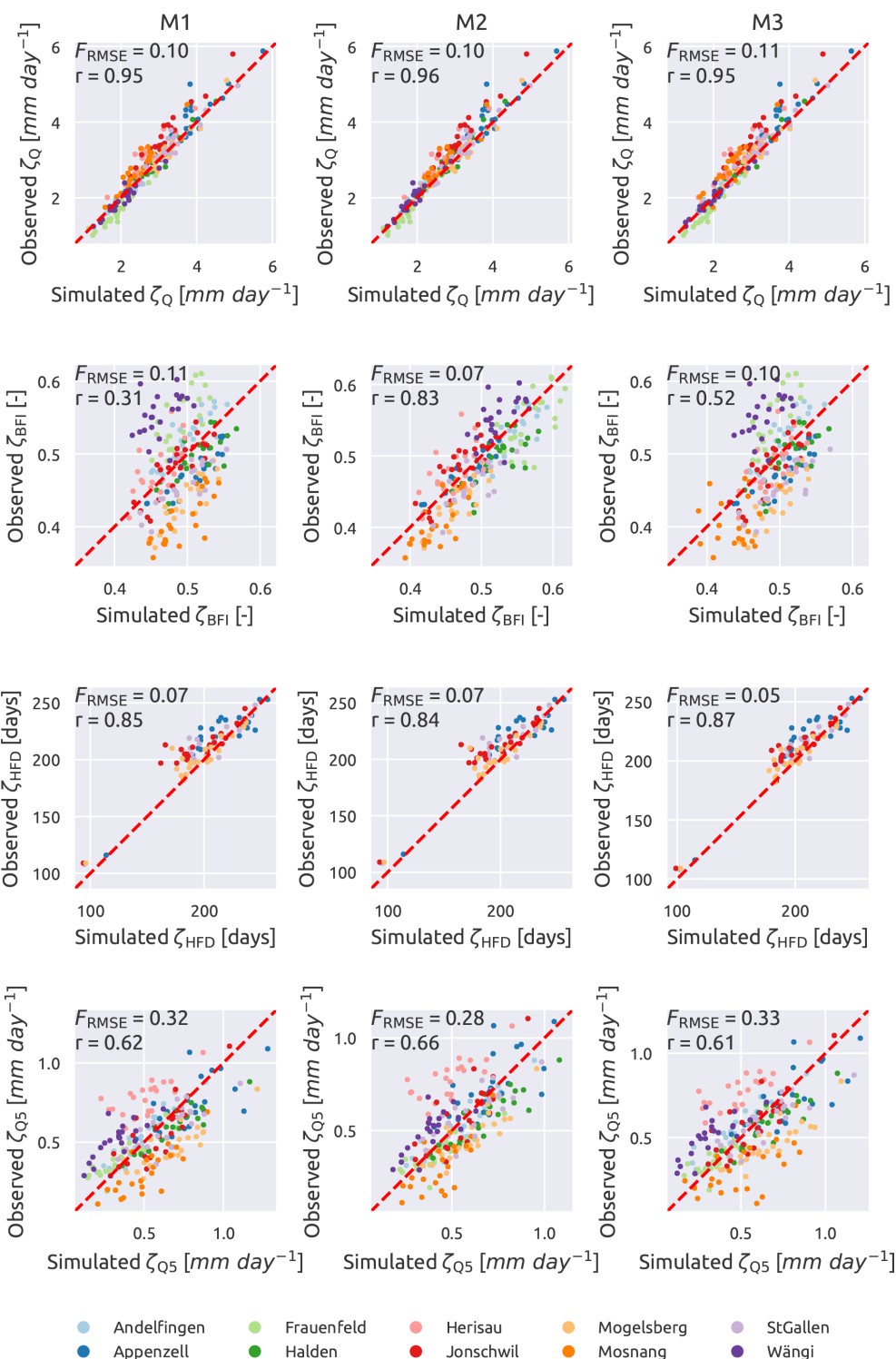

**Figure 9.** Simulated vs. observed streamflow signatures. Single-HRU model on the left (M1), two-HRU model based on geology in the centre (M2), and two-HRU model based on land use on the right (M3). Each dot represents a year and each colour a subcatchment. From up to bottom, mean daily streamflow ($\zeta_Q$), baseflow index ($\zeta_{BFI}$), mean half streamflow date ($\zeta_{HFD}$, only the catchment with $\psi_{FS}$ larger than 10 %), and 5th percentile of the streamflow ($\zeta_{Q5}$). The red dashed line has a 45° slope and indicates where all points should align in case of a perfect match. Spearman's rank score ($r$) is also reported.

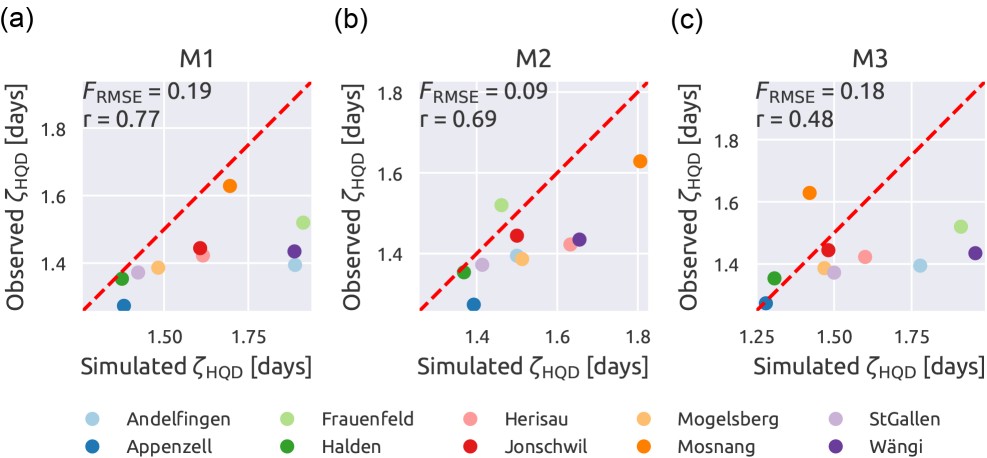

**Figure 10.** Ability of the hydrological models to represent the signature duration of low-flow events ($\zeta_{HQD}$). Single-HRU model **(a)**, two-HRU model based on geology **(b)**, and two-HRU model based on land use **(c)**.

the decision taken and the argumentations on which they are based, how weak or strong they may be, so that they can be reappraised and revised if new evidence is acquired.

Although our results in terms of hypotheses 1–4 described in the previous section appear justifiable based on previous work, they are not a priori obvious. In terms of the first hypothesis, although it is known that precipitation has a strong control on the average streamflow, it is less clear whether the spatial variability in the streamflow average can only be attributed to precipitation: some authors, for example, pointed to the role of regional groundwater flow in affecting the water balance (e.g. Bouaziz et al., 2018); GR4J (Perrin et al., 2003), for example, has a parameter that quantifies catchment gains and losses. Our modelling experiments, in particular through M0, have shown that groundwater processes, which potentially alter the water balance, do not influence the mean streamflow spatial variability of the Thur catchment.

In terms of the snow processes, although it is clear that, when there is snow (as in this case), the model needs to have a snow component, it is less obvious (at least just by looking at hydrographs) how many of the differences in the seasonality of the streamflow response between catchments are due to snow. The objective of the comparison between M0 and M1 is not to show that adding a snow component improves the overall performance, but that the differences in seasonality are captured by the model only when the snow component is integrated.

In terms of the effect of geology, Kuentz et al. (2017) made a classification study over more than 40 000 catchments across all Europe (of which almost 2700 are gauged) and found that geology controls the BFI, topography the flashiness index, and, for most of the cases, land use is the second control of them; Bloomfield et al. (2009) used a linear regression model and linked the lithology of the Thames Basin (UK) with the BFI; Lacey and Grayson (1998) noted that geology controls the BFI in two ways, storing the water

and impacting the soil formations; Fenicia et al. (2016) compared different model structures and catchment discretization methods in the Attert Basin (Luxemburg) and discovered that the best model was the one that incorporates a spatial representation of the meteorological inputs and of the geology. On the other hand, this general tendency should not be generalized to all places. For example, Mazvimavi et al. (2005) found that geology was not important for the BFI, as in their case study the aquifer was deep and disconnected from the river. Bouaziz et al. (2018) found a strong influence of regional groundwater flow in the Meuse catchment which altered the water balance.

The choice of streamflow signatures is based on the large-sample study from Addor et al. (2017), which provides a broad range of signatures typically used in hydrology. Our analyses showed that this selection is rather inclusive, with several strongly correlated signatures (e.g. $\zeta_Q$ and $\zeta_{RR}$). For this reason, we eventually used a much smaller selection of the original set of signatures (12 in the original set vs. 5 in the final set). Although hundreds of signatures have been proposed in the literature (e.g. Olden and Poff, 2003), the apparent inclusivity of the set from Addor et al. (2017) provides confidence that the main properties of streamflow are captured in our study. However, it does not guarantee that this set of signatures is sufficient in representing streamflow time series.

One of the main limitations of this work is the restricted number of catchments involved and the limited spatial extension of the study. For this reason, it is difficult to generalize the results to other climatic regions. The subcatchments all belong to the same region, and the landscape and climatic characteristics, while varying substantially within the basin, are still a small sample of the characteristics found elsewhere. Moreover, although the model evaluation uses validation in space and time, which is a relatively incisive test, the spatial validation is carried out in a nested setup. The ap-

plication of systematic model development strategies to other places and scales, and spatial validation to entirely different regions, is necessary to obtain more generalizable insights.

The small number of subcatchments involved in this study (10) limits the range of viable methods for identifying relationships between landscape and climatic indices and streamflow signatures (Sect. 3.1) to rather simple approaches. In particular, our correlation analysis, although accounting for non-linearity, is limited to monotonic correlations between variables, and it is unable to identify other forms of relationship, including the mutual interaction between various influence factors. The usage of more advanced techniques, including machine learning approaches such as random forest or clustering analyses, is most efficient when larger samples are available and could represent a more suitable choice in these situations.

## 6 Conclusion

In this study, we presented the development process of a distributed model where model hypotheses, instead of being made a priori, are informed by preliminary analysis on determining the dominant climatic and landscape controls on streamflow spatial variability. Besides providing guidance to model development, the proposed approach is useful in the fact that modelling can be used to test specific hypotheses on dominant processes resulting from such preliminary analysis. Our analysis was applied to the Thur catchment, subdivided into 10 subcatchments based on available stream gauging stations. The main findings are summarized in the following points.

1. We found large spatial variability between the subcatchments of the Thur in terms of various streamflow signatures reflecting multiple temporal scales: yearly, seasonal, and event scale. In terms of climatic characteristics, indices reflecting fraction of snow, precipitation totals, and aridity varied considerably among catchments. Other precipitation characteristics such as season, frequency, and duration of dry and wet days did not vary significantly among catchments. In terms of landscape characteristics, there is large variability of topography (e.g. from upstream mountainous to downstream flat areas), geology (with unconsolidated, more permeable, and consolidated, relatively impermeable formations), and soils (with low depths in the mountains and large depths in the floodplains) in all the catchments.

2. Based on correlation analysis and expert judgement, we determined that climatic variables, especially the precipitation average, are the main controls on streamflow average yearly values; the fraction of snow is responsible for streamflow seasonality by delaying the release of winter precipitation to the spring season, and geology

controls the baseflow index, with a higher fraction of unconsolidated material determining higher baseflow.

3. The results of the signature analysis were translated into a set of model hypotheses: a model with uniform parameters and distributed precipitation input (M0), the addition of a snow component (M1), the subdivision of the catchment into geology-based HRUs (M2), and the alternative subdivision of the catchment using land-use-based HRUs (M3).

4. Using model comparison and a validation approach that considers model performance (also in terms of signatures) in space–time validation, we found that it is necessary to account for the heterogeneity of precipitation, snow-related processes, and landscape features such as geology in order to produce hydrographs that have signatures similar to the observed ones. In particular, we confirmed that M0, in spite of a generally poor performance, is sufficient to capture signatures of streamflow average, showing that only distributing the meteorological inputs is sufficient to explain regional differences in average streamflow and that other phenomena potentially altering the water balance (e.g. regional groundwater flows) do not play a significant role. M1 improves signatures of streamflow seasonality, showing that snow is the main cause of the variability of the seasonality among the catchments. M2 enables signatures such as the baseflow index to be reproduced, showing that incorporating the geology of the catchment is important for reproducing regional differences in baseflow. Model modifications that are not in line with the results of the signature analysis, such as subdividing the catchment using land-use-based HRUs (M3), despite leading to the same complexity as M2, cause deterioration in model performance in space–time validation. Overall, these results confirm the hypotheses based on the signature analysis and suggest that the causality relationships, explaining the influence of climate and landscape characteristics on streamflow signatures, can be constructively used for distributed model building.

The relatively good performance obtained in space–time validation suggests that the proposed approach could be used for the prediction of the streamflow in other ungauged locations within the Thur catchment. The method proposed uses data that are commonly available in many gauged catchments (e.g. meteorological data, streamflow measurements, maps of elevation, geology, land use, and soil); therefore, it is easily transferable to other locations.

## Appendix A: Hydrological model details

### Model equations

The equations of the model are listed in this Appendix; the model structure in presented in Fig. 6. Table A1 contains the
5 model parameters with the range of variability used in calibration, Table A2 lists the water-budget equations, and Tables A3 and A4 present the functions and the constitutive functions used.

**Table A1.** Hydrological model parameters with the range of variation used for the definition of the uniform prior distribution. The "component" column indicates the element (reservoir, lag, or network) where the parameter belongs.

| Parameter | Unit | Component | Range of variability |
|---|---|---|---|
| $C_e$ | – | Unsaturated reservoir (UR) | 0.1–3.0 |
| $S_{max}^{UR}$ | mm | Unsaturated reservoir (UR) | 0.1–500.0 |
| $k_{WR}$ | $d^{-1}$ | Snow reservoir (WR) | 0.1–10.0 |
| $t_{rise}^{IL}$ | d | Network lag | 0.5–10.0 |
| $t_{rise}^{OL}$ | d | Network lag | 0.5–10.0 |
| $D$ | – | Structure | 0.0–1.0 |
| $k_{FR}$ | $d^{-1}\,mm^{-2}$ | Fast reservoir (FR) | $10^{-6}$–10.0 |
| $k_{SR}$ | $d^{-1}$ | Slow reservoir (SR) | $10^{-6}$–1.0 |
| $t_{rise}^{lag}$ | d | Structure lag | 1.0–20.0 |

**Table A2.** Water-budget equations (see the model schematic in Fig. 6).

| Component | Equation |
|---|---|
| Snow reservoir (WR) | $\frac{dS_{WR}}{dt} = P_{WR} - Q_{WR}$ |
| Unsaturated reservoir (UR) | $\frac{dS_{UR}}{dt} = P_{UR} - Q_{UR} - E_{UR}$ |
| Lag function | $Q_{UR} = P_{SR} + P_{lag}$ |
| Slow reservoir (SR) | $\frac{dS_{SR}}{dt} = P_{SR} - Q_{SR}$ |
| Fast reservoir (FR) | $\frac{dS_{WR}}{dt} = P_{FR} - Q_{FR}$ |
| Outflow | $Q = Q_{FR} + Q_{SR}$ |

**Table A3.** Constitutive functions of the model. Refer to Table A4 for the definition of the functions $f$.

| Component | Equation |
|---|---|
| Snow reservoir (WR)[a] | $P_{\mathrm{WR}} = \begin{cases} P & \text{if} \quad T \leq 0 \\ 0 & \text{if} \quad T > 0 \end{cases}$ |
| Snow reservoir (WR)[b] | $M_{\max}^{\mathrm{WR}} = \begin{cases} 0 & \text{if} \quad T \leq 0 \\ k_{\mathrm{WR}} T & \text{if} \quad T > 0 \end{cases}$ |
| Snow reservoir (WR) | $Q_{\mathrm{WR}} = M_{\max}^{\mathrm{WR}} f_{\mathrm{e}}(S_{\mathrm{WR}}|2)$ |
| Unsaturated reservoir (UR) | $\overline{S_{\mathrm{UR}}} = \frac{S_{\mathrm{UR}}}{S_{\max}^{\mathrm{UR}}}$ |
| Unsaturated reservoir (UR) | $E_{\mathrm{UR}} = C_{\mathrm{e}}(\mathrm{PET})\,\boxed{\text{TS4}}\,f_{\mathrm{m}}(S_{\mathrm{UR}}|0.01)$ |
| Unsaturated reservoir (UR) | $Q_{\mathrm{UR}} = P_{\mathrm{UR}} f_{\mathrm{p}}(\overline{S_{\mathrm{UR}}}|2)$ |
| Slow reservoir (SR) | $P_{\mathrm{SR}} = D Q_{\mathrm{UR}}$ |
| Slow reservoir (SR) | $Q_{\mathrm{SR}} = k_{\mathrm{SR}} S_{\mathrm{SR}}$ |
| Lag function[c] | $P_{\mathrm{FR}} = (P_{\mathrm{L}} \cdot h_{\mathrm{lag}})(t)$ |
| Lag function | $h_{\mathrm{lag}} = \begin{cases} 2t/\left(t_{\mathrm{rise}}^{\mathrm{lag}}\right)^2 & \text{if} \quad t \leq t_{\mathrm{rise}}^{\mathrm{lag}} \\ 0 & \text{if} \quad t > t_{\mathrm{rise}}^{\mathrm{lag}} \end{cases}$ |
| Fast reservoir (FR) | $Q_{\mathrm{FR}} = k_{\mathrm{FR}} S_{\mathrm{FR}}^3$ |
| Lags in the network[c] | $Q_{\mathrm{out}} = \left(Q_{\mathrm{in}} * h_{\mathrm{lag}}^{\mathrm{net}}\right)(t)$ |
| Lags in the network | $h_{\mathrm{lag}}^{\mathrm{net}} = \begin{cases} 2t/\left(t_{\mathrm{rise}}^{\mathrm{OL/IL}}\right)^2 & \text{if} \quad t \leq t_{\mathrm{rise}}^{\mathrm{OL/IL}} \\ \left(1/t_{\mathrm{rise}}^{\mathrm{OL/IL}}\right)\left(1 - \left(\left(t - t_{\mathrm{rise}}^{\mathrm{OL/IL}}\right)/t_{\mathrm{rise}}^{\mathrm{OL/IL}}\right)\right) & \text{if} \quad t_{\mathrm{rise}}^{\mathrm{OL/IL}} < t \leq 2t_{\mathrm{rise}}^{\mathrm{OL/IL}} \\ 0 & \text{if} \quad t > 2t_{\mathrm{rise}}^{\mathrm{OL/IL}} \end{cases}$ |

[a] This equation is smoothed using a logistic scheme, Eq. (8) in Kavetski and Kuczera (2007), with smoothing parameter $m_{\mathrm{P}} = 1.5\,^{\circ}\mathrm{C}$. [b] This equation is smoothed using a logistic scheme, Eq. (13) in Kavetski and Kuczera (2007), with smoothing parameter $m_{\mathrm{M}} = 1.5\,^{\circ}\mathrm{C}$. [c] The operator $*$ denotes the convolution operator, smoothed according to Kavetski and Kuczera (2007).

**Table A4.** Constitutive functions.

| Function | Name |
|---|---|
| $f_{\mathrm{e}}(x|\theta) = 1 - \exp(-x/\theta)$ | Tessier function. Note that $f_{e}(x|\theta) \rightarrow 1$ as $x \rightarrow \infty$. |
| $f_{\mathrm{p}}(x|\theta) = x^{\theta}$ | Power function |
| $f_{\mathrm{m}}(x|\theta) = \frac{x(1+\theta)}{x+\theta}$ | Monod-type kinetics, adjusted so that $f_{\mathrm{m}}(1|\theta) = 1$ |

*Data availability.* All the interpolated data used for this publication as well as the codes to calculate the metrics (e.g. indices and signatures) are stored in an institutional repository (https://doi.org/10.25678/0001RK, Dal Molin et al., 2020). Due to restrictions, the raw data have to be requested directly by the providers. Meteorological data can be obtained by the Federal Office of Meteorology and Climatology, MeteoSwiss; streamflow data can be obtained from the Federal Office for the Environment (FOEN); the maps used for calculating the catchments characteristics can be obtained by the Federal Office of Topography, swisstopo; all the other codes are available upon request.

*Author contributions.* MDM and FF designed all the experiments. MZ contributed to the preparation of the input data for the study. MDM conducted all the experiments and analysed the results. MDM prepared the paper with the contributions from all the authors.

*Competing interests.* The authors declare that they have no conflict of interest.

*Special issue statement.* This article is part of the special issue "Linking landscape organisation and hydrological functioning: from hypotheses and observations to concepts, models and understanding (HESS/ESSD inter-journal SI)". It is not associated with a conference.

*Acknowledgements.* The authors thank the Federal Office of Meteorology and Climatology, MeteoSwiss, for the meteorological data and the Federal Office for the Environment, FOEN, for the streamflow data. The authors thank Conrad Jackisch (editor), Lieke Melsen, Shervan Gharari, and the two anonymous referees for their feedback and their help in improving this paper. MZ and MS dedicate this work to the memory of Joachim Gurtz, who pioneered the work on distributed hydrological modelling in the Thur catchment (Gurtz et al., 1999).

*Financial support.* This research has been supported by the Schweizerischer Nationalfonds zur Förderung der Wissenschaftlichen Forschung (grant no. 200021_169003).

*Review statement.* This paper was edited by Conrad Jackisch and reviewed by Lieke Melsen, Shervan Gharari, and two anonymous referees.

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

## Remarks from the language copy-editor

## Remarks from the typesetter