# Peer review of "Understanding dominant controls on streamflow spatial variability to set-up a semi-distributed hydrological model: the case study of the Thur catchment."

_Hydrology and Earth System Sciences, 2019_

## Referee Comment (RC1) · Anonymous Referee #1 · 2 Apr 2019

The authors propose to infer the structure of a hydrological model based on landscape and process characteristics (signatures) of the catchment. In the first of a two-stage process different landscape and catchment characteristics are compared to different streamflow signatures to identify the most important controls on runoff formation. In the second step this information is used "as an inspiration for model structure design" (p17, l. 32) as the authors put it.

Inferring structure from function (or vice versa) is at the core of hydrological model building and subject to numerous studies. The topic is hence highly relevant for the hydrological community. The manuscript is well structured and well-written. Accordingly

the manuscript is suitable for a publication on HESS. However, I cannot recommend publishing to current version of the manuscript due to several major points:

1) The purpose of the modelling exercise is not clear. Model requirements for flood forecasting are e.g. totally different from model requirements to simulate climate change. The relevant signatures, temporal and spatial model discretisation, model evaluation metrics and also the degree of model conceptualisation differ accordingly. Please specify more clearly the purpose of you modelling study. Otherwise it is not possible to evaluate the study meaningfully.

2) I consider the selection, evaluation and identification of landscape characteristics as fairly weak due to a number of different reasons:

a. The authors provide no information about why certain characteristics were selected (and why others were not). Catchment characteristics (or signatures) can only provide information on the underlying processes if they have some kind of diagnostic potential or causal relationship. It is clear that these relationships are often unknown and difficult to obtain; nevertheless the selection of appropriate characteristics is vital for the identification of underlying processes and mechanisms. I miss a clear and elaborate description on the selection of catchment descriptors and on their expected diagnostic potential (both in space and time): E.g. why or how can the different land cover ratios or aspects help to derive information on hydrological processes? Are the same characteristics suitable for all catchments (independent of size, altitude, geology)? Please also comment on the importance of the time step e.g. you calculated the flashiness index based on daily streamflow data, although you state that streamflow can change two orders of magnitude in a few hours (p. 3 l. 18). If this is true please explain why you consider a daily-data based flashiness index as a meaningful variable? Please do also explain why you think that "half streamflow period" is a suitable parameter to discriminate to importance of snow. I expect that there are much simpler and more meaningful variables such as temperature and rain, temperature sums or snow data itself to describe the importance of snow. The results in Fig 7-9 also show that streamflow, runoff

coefficient and half streamflow period are pretty identically in all cases. Do you consider them being suitable signatures? Please also provide more signature papers in the introduction as the number of up to date references is small.

b. In your study you included several (fairly easy to derive) landscape characteristics that are obviously highly correlated and describe in great detail how you identify and select appropriate ones based on regression and correlation. In my opinion a rather trivial part which does not add any new knowledge to the literature occupies a lot of space. I hence suggest shortening and streamlining the entire section. If you want to derive structure from function than the first goal must be to derive a (comprehensive) matrix of uncorrelated catchment characteristics that have some kind of diagnostic potential. In my opinion this should be the source of the story and not a result.

3) The approach for informing model structure does not appear very elaborate to me. Since this is the core of "model building for understanding catchment process" I particularly miss a clear and elaborate discussion on how the identified landscape characteristics help in the model building process. More specifically:

a. In chapter 3.1.3. you state that the results of the regression analysis were used to build the hydrological model e.g. the subdivision of the catchment in HRUs (p. 7 l. 32). Later, in 4.1.1 you state that subdivisions were defined by gauge locations (p. 11 l 26). I did not find information on how you derived the number of HRUs and the role of catchment characteristics in this context? Chapter 4.1.1. should be more comprehensive in this regard.

b. The argument that "the regression analyses have indicated that precipitation is a dominant control on average streamflow" (p. 12 l. 4) is trivial. I don't think you need this and particular not as a justification for using spatially distribution rainfall as a model input. From your manuscript it appears to me that the spatial discretization of your model was based on the definition of the subcatchments (which are in turn defined by the location of gauges) and according the definition of fields (definition not clear). In

consequence I don't see that landscape characteristics played an important role in this process. Please clarify?

c. You also mention that "the parameters were motivated by the results of the regression analysis" (p. 8 l. 1). Please omit or explain more detailed. A matrix to illustrate the relationships between model parameters and catchment descriptors would be good. I would for instance be interested in how one could use catchment descriptors to derive (or at least constrain) model storage (kFR or kSR) or network lag (trise,IL trise,OL) parameters. Please comment on that

d. Chapter 4.1.5 is difficult to me due to different reasons: i) your analysis does not VERIFY that "models that account for influencing factors . . . lead to an improved representation". Essentially it only shows that a complicated model (with a larger number of degrees of freedom) outperforms a simpler model (with a smaller amount of degrees of freedom). Please use precise wording. It addresses the question of adequate model complexity. If a lumped representation (M1) is not adequate than also the comparison of M2 to M1 is not adequate. Please explain in more detail why you consider M1 a suitable reference? ii) Please explain why unconsolidated areas receive an individual HRU and why consolidated and alluvial areas can be lumped together (what are your expectation on the underlying processes)? iii) The parameterization of M3 is based on land use, which is not considered to have a causal relationship to the streamflow signatures (Table 2). Please explain why a model which is derived from non-causal properties can be a considered a meaningful reference? Why did you group based on geology and not on elevation, slope or the aridity index which you considered to have a causal relationship? This would maybe be a more appropriate benchmark? iv) Essentially chapter 4.1.5 addresses the questions of optimal degree of model complexity and optimal degree of spatial discretization - which are both very important. However, these aspects are treated together and not separated from each other. Moreover, potential answers to these questions miss a clear link to catchment descriptors. Essentially only differences in geology were considered in the model building. Please clarify to novelties

of your study more clearly.

e. the wohle structure of the model building story is a bit complicated as aspects are described in chapters 3.1.3, 3.3, 4.1.4 und 4.1.5 which makes it difficult to follow. I suggest combining them into a single chapter. Therein start with the theory e.g. snow is important followed by the surrogate you considered it e.g. half stream flow period. Or geology is important due to. . . –> different HRUs.

4) Several conclusions are not appropriate: e.g. "the proposed approach is useful in the fact that modelling can be used to test specific hypotheses on dominant processes resulting from regression analysis" (p. 19 l. 4). This has not be shown. More over aspects related to the event scale are mentioned in the first three bullet points but not subject to the manuscript. In the third bullet point you state: "Higher proportion of consolidated material has an influence on the baseflow vs quickflow portioning, causing lower baseflow and higher peaks" (p 19 l 14). Does the study provide evidence for this statement or does it support this hypothesis? I expect the latter and missed this statement in the chapter 3.1.1. I suggest re-writing of the entire section and to differentiate concisely between hypothesis, results and conclusions.

5) The model performance evaluation (chapter 4.1.4) is complicated but of minor importance in this context. I suggest shorting the evaluation section and to focus on a single, interpretable metric e.g. the Kling-Gupta Efficiency as the NASH has several limitations and the normalized log-likelihood is difficult to interpret. But this is a minor point and a matter of taste.

Technical corrections (figures and tables only) I only provide technical corrections for the figures and tables as I expect that several parts of the manuscript will be subject to major revisions.

Figure 1: A: I suggest to remove the colour code and to provide notations (abbreviation) in or around the map. This would help improve the readability of the stream network and the location of the gauges. If you want to keep the legend please add catchment

abbreviations to it, order it according to Fig 2. and use a meaningful colour code (e.g. mean annual precipitation, elevation or geology), B: Try a discrete legend like in atlases, will improve readability. C: Forest and pasture are hardly distinguishable both on my screen and in a printed version.

Figure 2: Please repeat the variables and their abbreviation in the caption such that the figure can be read independent from the text. Maybe add another row and provide grouping indices based on the results in chapter 3.2.1

Figure 3: Please repeat the variables and their abbreviation in the caption such that the figure can be read independent from the text.

Figure 4: Please repeat the variables and their abbreviation in the caption such that the figure can be read independent from the text. Information on the range of the different variables would be pretty helpful as well. If possible include it otherwise please mention the ranges in the text or add the information to table 1.

Figure 5: I'm not sure if this figure is required since B and C show very little variation. The only important message from A is that there a catchments that are stronger controlled by snow than others. I suggest removing it. If you decide to keep it update the colour code according to the suggestion for Figure 1.

Figure 6: I cannot find a description of the symbols and abbreviations in the Appendix. Please specify at least the meaning of the capital letters in the caption (as in 4.1.1) and provide a more comprehensive description in the appendix.

Figure 7: Order according to Figure 2 or 3. Line type and colour code are redundant.

Figure 8, 9, 10: Nice figures! Suggestions: Combine all three figures in one (each model setup as an individual row). This would improve readability. Streamflow, runoff coefficient and half streamflow period have no or little variation (two out of these could be omitted such that all results would fit in one figure). Remove the correlation coefficients due to their distracting nature (correlation (alone) is pretty meaningless in this

context). Update colour code according to Fig 1 A.

Table 1: Order according to fig 1 A. Index column is not relevant, omit Code or put it to the very right. Rounding is not yet meaningfully and consistent.

Table 2: This table includes variables with spurious correlations (Brett 2004, Kenny 1982). This includes variables that are considered statistically significant and where causality was assumed e.g. the correlation between aridity index AI and the runoff coefficient RC which are both are derived from precipitation. The same applies for P and RC. Since P and Q are highly correlated and AI is based on P I also wonder about the significance of AI and RC, BFI, FI and HDP. Please clarify. Please also explain why you assumed causality among LP and BFI and among LP and FI? Differences among the geological fractions are small as well. Why do you consider causality in some of the individual relationships and in others not?

Table 3: This analysis also includes variables with spurious correlations. Please comment on that.

Table A1: Please provide a brief explanation on parameters and components. Where does the range of variability come from?

Table A2: Explain component

Literature Brett, M. T. (2004). When is a correlation between non-independent variables "spurious"? Oikos, 105(3), 647–656. Kenney, B. C. (1982). Beware of spurious self-correlations! Water Resources Research, 18(4), 1041–1048. https://doi.org/10.1029/WR018i004p01041

---

## Referee Comment (RC2) · Lieke Melsen (Referee) · 3 Apr 2019

Dal Molin et al. investigated, through regression analysis, which indices have explanatory power for streamflow response. Based on the insights gained from the regression analysis, different spatial configurations were implemented in a hydrological model. Although I find the work flow elegant, starting from process-understanding and translating that to the spatial configuration of the model, I have some problems / concerns with the regression set-up.

Major

My main concerns are all related to the regression-part of the study.

1. It is unclear how the indices, on which regression was applied, were selected. There is plenty of literature around on indices and signatures, which could guide indices-selection, but I don't see any justification in the text for the choices made. Check for instance:

Addor et al., A Ranking of Hydrological Signatures Based on Their Predictability in Space, WRR, 2018

Knoben et al., A Quantitative Hydrological Climate Classification Evaluated With Independent Streamflow Data, WRR, 2018

2. Since the choice of the indices is not well justified, I am worried about their mutual correlation. Many indices can describe the same signal. Therefore, please provide the correlation among the indices themselves. This might lead to the insight that you need fewer or different ones.

3. It was rightly mentioned that correlation does not mean causality. It was claimed (not only in the methods, but also in the conclusions) that this study accounts for that by only selecting the indices that have a causal relation, based on expert judgement. I do, however, not recognize the expert judgement in the selection of significant indices, and this actually directly relates to my point 2. Right now, the selection seems to be made based on the mutual correlation of the indices – so the mutual correlation was investigated! – but I don't see any process-reasoning (the expert-judgement) that can justify the selected indices, and that justifies the claim that there is really causality.

4. I disagree with the conclusion of the authors that there is no need to look for nonlinearity in the correlation, based on the results in the table. The authors rightly state that only few correlations that are statistically significant based on Spearman are not significant with Pearson, but how do the authors explain the opposite effect? Quite some correlations are significant with Pearson but not with Spearman, is this a Type HESSD
2 error in the Pearson test? That could have consequences, for example, aridity was significantly correlated with BFI for Pearson, but not Spearman, and based on 'expert judgement' included in the regression.

5. 1 of the 3 points of the guidelines for modelling based on the regression was not based on the regression at all, namely the conclusion that the presence of snow is relevant. Please include a snow-related indicator in the regression to support this conclusion (based on expert judgement we can expect this, of course).

6. It depends a bit on the definition of model building, but the title and the text might give the impression that the model structure itself was adapted with the insights in the regression, while it was basically the model implementation (accounting for HRU's or not) that was adapted.

Minor

Section 3.2.1, the catchments are sort of grouped based on their stream flow response, but this is not used any further in the analysis. Consider to just briefly describe their response, or to use the grouping later to explain results (in that case, also display the groups in the figure).

In the same section, it seems unnecessarily complicated to use combinations of signatures to determine how flashy catchments are; a flow duration curve can generally provide quite some insight on this already (slope of flow duration curve also frequently used signature)

Provide an overview of the indices and their abbreviations, or include their full name more often in text / tables / figures, because now it requires quite some work from the reader to fully understand all sentences and figures (and a lot of going back to the methods).

A large number of figures is dedicated to showing the signature-values, which is not of direct relevance. I would be interested to see a figure that displays the HRU's.

HESSD
For the landscape characteristics, it is not mentioned in the methods-section (3.1.1) that you consider fractional area. Please clarify there, as I was wondering how you would apply regression on nominal values, until I found out in the results that you considered frac. area.

The sentence 'optimizing the parameter of the posterior distribution' (I.11, p13) can give the impression that you minimized e.g. variance (describing distribution), please consider reformulation.

Although overall written well and clear, some language editing seems required, for example "The average value oscillates of about ..", (but I'm not a native either).

Overall, I appreciate the intent of the study and the modelling-part seems well designed (except for my question at point 6 which remained unclear), but I do believe the regression-part requires substantial revision, related to the selection of indices (more embedded in literature and account for snow) and to justify the use of the word 'causality'. Given that the work-flow is largely set-up, I think the authors should be able to incorporate this.

Lieke Melsen

HESSD

---

## Author Comment (AC1) · 22 May 2019

**Reply to review by Anonymous Referee #1**

We thank the reviewer for his/her careful read to the manuscript and insightful suggestions.

After reading carefully the comments of both the reviewers, in order to address some of their most substantial comments, there will be some major changes in the revised version of the manuscript. These changes will affect most significantly the data analysis part of the paper (section 3), and they are summarized hereafter:

- We will extend the selection of hydrological signatures in order to have a more comprehensive picture of the hydrological behavior of the catchments.
- We will select only signatures that are not correlated (reducing substantially their number) for the subsequent analyses.
- We will select a reduced number of catchment and meteorological indices in order to reduce the problem of correlated features.
- All the correlation analysis will be based on the Spearman's rank score since, as pointed out by Dr. Melsen, the usage of Pearson may not be adequate.
- Following this decision, we will remove the regression analysis (table 3) since (1), as stated by the Anonymous Referee #1, " it does not add any new knowledge to the literature occupies a lot of space" and because (2) the usage of linear regression (that looks for linear dependencies in the data) is not coherent with the choice of basing the analysis on the Spearman's rank score (which accounts also for nonlinear correlations).

Below, we answer in detail the various comments, and illustrate our intended approach to address them in the revised version. The original comments of the reviewer are reported in *black and italics*, our replies in blue.

*The authors propose to infer the structure of a hydrological model based on landscape and process characteristics (signatures) of the catchment. In the first of a two-stage process different landscape and catchment characteristics are compared to different streamflow signatures to identify the most important controls on runoff formation. In the second step this information is used "as an inspiration for model structure design" (p17, l. 32) as the authors put it.*

*Inferring structure from function (or vice versa) is at the core of hydrological model building and subject to numerous studies. The topic is hence highly relevant for the hydrological community. The manuscript is well structured and well-written. Accordingly the manuscript is suitable for a publication on HESS. However, I cannot recommend publishing to current version of the manuscript due to several major points:*

1. *The purpose of the modelling exercise is not clear. Model requirements for flood forecasting are e.g. totally different from model requirements to simulate climate change. The relevant*

*signatures, temporal and spatial model discretisation, model evaluation metrics and also the degree of model conceptualisation differ accordingly. Please specify more clearly the purpose of you modelling study. Otherwise it is not possible to evaluate the study meaningfully.*

We will complement the objectives of the studies, already summarized in the introduction (lines 6-10 page 3) with the objectives of the modeling exercise. In the introduction, we already specified the main objectives of the study, which in summary consist of proposing a model building strategy which starts from an analysis of the data, which provide a basis for motivating the various model decisions.

However, as the reviewer noted, the purpose of the model itself has remained unclear. Although one of the main objectives of models is making predictions to address some practical issues, here we do not have such an immediate applied objective. The model exercise is mainly an instrument to help understand and interpret catchment scale processes. In particular, we are interested in identifying the landscape properties and associated processes that dominate catchment response, and that mostly influence the observed spatial variability in streamflow behavior, as characterized by the set of suggested signatures.

The reviewer is right in the fact that aspects such as "signatures, temporal and spatial model discretization, model evaluation metrics and also the degree of model conceptualization differ accordingly". We will clarify that there the objective is explaining the observed spatial diversity of streamflow characteristics, with the minimum possible complexity, while trying to maintain a process based interpretation.

2. *I consider the selection, evaluation and identification of landscape characteristics as fairly weak due to a number of different reasons:*

   a. *The authors provide no information about why certain characteristics were selected (and why others were not). Catchment characteristics (or signatures) can only provide information on the underlying processes if they have some kind of diagnostic potential or causal relationship. It is clear that these relationships are often unknown and difficult to obtain; nevertheless the selection of appropriate characteristics is vital for the identification of underlying processes and mechanisms. I miss a clear and elaborate description on the selection of catchment descriptors and on their expected diagnostic potential (both in space and time): E.g. why or how can the different land cover ratios or aspects help to derive information on hydrological processes? Are the same characteristics suitable for all catchments (independent of size, altitude, geology)? Please also comment on the importance of the time step e.g. you calculated the flashiness index based on daily streamflow data, although you state that streamflow can change two orders of magnitude in a few hours (p. 3 l. 18). If this is true please explain why you consider a daily-data based flashiness index as a meaningful variable? Please do also explain why you think that "half streamflow period" is a suitable parameter to discriminate to importance of snow. I expect that there are much simpler and more meaningful variables such as temperature and rain, temperature sums or snow data itself to describe the importance of snow. The results in Fig 7-9 also show that streamflow, runoff coefficient and half streamflow period are pretty identically in all*

*cases. Do you consider them being suitable signatures? Please also provide more signature papers in the introduction as the number of up to date references is small.*

The reviewer correctly points out that we "miss a clear and elaborate description on the selection of catchment descriptors and on their expected diagnostic potential (both in space and time)". We will complement the selection of catchment characteristics with their expected diagnostic potential. For example, vegetation characteristics are typically assumed to affect evaporation, soil characteristics are typically assumed to influence the partitioning of water between retention and runoff. In general, we tried to select a broad class of characteristics, to be as inclusive as possible. However, it is also true that these characteristics can be represented through a large class of indices, and in order to reduce the size of the problem, some choices had to be made. We will motivate some of our decisions based on how other models have dealt with similar issues, so that some of the choices will appear less ad hoc.

We used a daily data resolution, and this choice clearly affects some of the signatures. As the reviewer points out, the flashiness index is one of such signatures. The values of the flashiness index reduce with increasing time step due to a smoothing effect. In this paper, we did not experiment with varying data resolution, as it was outside our scope. However, we will comment that this choice is expected to affect some of the signatures, and consequently their diagnostic power on some of the associated processes.

We experimented with several signatures to account for the effect of snow on streamflow seasonality, and we ended up selecting the "half streamflow period". The reason is that 1) this signature was used in previous publications to quantify streamflow seasonality, and we did not want to invent our own signature if something was already existing, and 2) this signature captured well the difference between streamflow regimes, because we have seen (figure 5) that all the catchments receive similar precipitation input (in terms of monthly variability) but the snow-affected ones show the peaks during late spring/beginning of summer while the rain dominated ones show their peaks during the winter and the spring.

Figures 7-9 show that all the model configurations represent well the yearly streamflow, the runoff coefficient, and the half streamflow period and this is a result of our study that is also coherent with our assumption that only distributing the inputs (precipitation, PET, and temperature) is sufficient in order to have a model that captures the water balance and the snow dynamics. In the revised paper we will show the analysis of the correlations between the signatures in order to select only the not redundant ones. We will add more references in the introduction to signatures papers.

b. *In your study you included several (fairly easy to derive) landscape characteristics that are obviously highly correlated and describe in great detail how you identify and select appropriate ones based on regression and correlation. In my opinion a rather trivial part which does not add any new knowledge to the literature occupies a lot of space. I hence suggest shortening and streamlining the entire section. If you want to derive structure from function than the first goal must be to derive a (comprehensive) matrix of*

*uncorrelated catchment characteristics that have some kind of diagnostic potential. In my opinion this should be the source of the story and not a result.*

We will try in the revised version to streamline this part. We will remove some of catchment characteristics before carrying out the correlation. This can apply, for example, to variables that are relatively uniform over the catchment, or to catchment characteristics that occupy very small areas of the catchment.

We could also remove variables that are strongly correlated to others, and do not have any perceived influence on the selected signatures.

3. *The approach for informing model structure does not appear very elaborate to me. Since this is the core of "model building for understanding catchment process" I particularly miss a clear and elaborate discussion on how the identified landscape characteristics help in the model building process. More specifically:*

   a. *In chapter 3.1.3. you state that the results of the regression analysis were used to build the hydrological model e.g. the subdivision of the catchment in HRUs (p. 7 l. 32). Later, in 4.1.1 you state that subdivisions were defined by gauge locations (p. 11 l 26). I did not find information on how you derived the number of HRUs and the role of catchment characteristics in this context? Chapter 4.1.1. should be more comprehensive in this regard.*

   We will clarify this aspect in the revised paper. Our intention was to present chapter 4.1.1 as a general overview of the model structure in order to make the following clearer. The information from the regression analysis are used to derive the HRUs is described in chapters 3.3 and 4.1.5. It is important to make clear the difference between the division in subcatchments (areas that have uniform inputs) and HRUs (areas that have the same hydrological response). The former are defined by the presence of gauging stations (and this division is not negotiable) while the latter reflect our understanding of the catchment functioning (and, in this study, of the regression analysis).

   b. *The argument that "the regression analyses have indicated that precipitation is a dominant control on average streamflow" (p. 12 l. 4) is trivial. I don't think you need this and particular not as a justification for using spatially distribution rainfall as a model input. From your manuscript it appears to me that the spatial discretization of your model was based on the definition of the subcatchments (which are in turn defined by the location of gauges) and according the definition of fields (definition not clear). In consequence I don't see that landscape characteristics played an important role in this process. Please clarify?*

   We will clarify that, although it may be a priori clear that precipitation needs to be distributed per subcatchment, it may be not as taken for granted that this is sufficient to capture the water balance of the subcatchments, as many other aspects could in principle play a role (e.g. regional groundwater flow). Here we show that considering distributed precipitation over the subcatchments (defined by the presence of the gauging stations) could by itself be sufficient. Other landscape characteristics play a role in the definition of the HRUs (section 4.1.5, see previous comment).

We will make the definition of subcatchments, HRUs and fields clearer and comment on the role of precipitation.

c. *You also mention that "the parameters were motivated by the results of the regression analysis" (p. 8 l. 1). Please omit or explain more detailed. A matrix to illustrate the relationships between model parameters and catchment descriptors would be good. I would for instance be interested in how one could use catchment descriptors to derive (or at least constrain) model storage (kFR or kSR) or network lag (trise,IL trise,OL) parameters. Please comment on that*

We will clarify this part, which was obviously misleading. All the process of building the model was motivated by the results of the regression analysis (in particular the decisions on the division in HRUs). The parameters are just calibrated using streamflow data (section 4.1.3). No inference of the parameters from catchment characteristics was done.

We will improve the text (section 3.1.3) to make it clearer.

d. *Chapter 4.1.5 is difficult to me due to different reasons: i) your analysis does not VERIFY that "models that account for influencing factors … lead to an improved representation". Essentially it only shows that a complicated model (with a larger number of degrees of freedom) outperforms a simpler model (with a smaller amount of degrees of freedom). Please use precise wording. It addresses the question of adequate model complexity. If a lumped representation (M1) is not adequate than also the comparison of M2 to M1 is not adequate. Please explain in more detail why you consider M1 a suitable reference? ii) Please explain why unconsolidated areas receive an individual HRU and why consolidated and alluvial areas can be lumped together (what are your expectation on the underlying processes)? iii) The parameterization of M3 is based on land use, which is not considered to have a causal relationship to the streamflow signatures (Table 2). Please explain why a model which is derived from non-causal properties can be a considered a meaningful reference? Why did you group based on geology and not on elevation, slope or the aridity index which you considered to have a causal relationship? This would maybe be a more appropriate benchmark? iv) Essentially chapter 4.1.5 addresses the questions of optimal degree of model complexity and optimal degree of spatial discretization - which are both very important. However, these aspects are treated together and not separated from each other. Moreover, potential answers to these questions miss a clear link to catchment descriptors. Essentially only differences in geology were considered in the model building. Please clarify to novelties of your study more clearly.*

We will more clearly explain the reasoning behind the choice of the model configurations done in this study. Essentially the two main model configurations are M1 and M2: the first is the baseline and it is a semidistributed model (in the sense that the inputs are spatially distributed and the routing between subcatchments is explicitly addressed in the model) with only one HRU (meaning that all the catchment responds in the same way to the forcings); the second extends the first providing a subdivision of the subcatchments in two HRUs. M3 is used to show that the subdivision in HRUs has to be

carefully made otherwise a more complex model doesn't imply automatically better results. Answering to the specific points:

i) M2 is indeed more complex than M1 but our thesis is that its better performance is not just due to the fact that it is more complex but to the fact that it incorporates the right catchment characteristics. This is also demonstrated by M3 that is as complex as M2 but it has the same deficiencies of M1. M1 is already a quite complex model since it already considers the spatial distribution of the inputs and incorporates information about the routing between subcatchments. The real baseline would have been a lumped model, with uniform input and no information about the catchment characteristics but it was too simple for the comparison.

ii) There is an error in the text: the two HRUs are unconsolidated and alluvial (HRU1) vs consolidated (HRU2). Alluvial and unconsolidated geology were put together because they show a similar behavior in terms of water dynamics in the sense that they both represent areas with high storage capacity, especially if compared with HRU2 that is quite impermeable.

iii) M3 was designed to demonstrate that M2 outperforms M1 not just because it is more complex but only because it incorporates characteristics that actually have an impact on the response of the catchment. For this reason we used a model with the same complexity of M2 but based on characteristics that don't correlate with the streamflow signatures. Also topography was considered in the modeling experiment (but not reported in the paper), experimenting with a subdivision in HRUs based on the slope, but the model resulted similar to M2 (in terms of spatial discretization) but slightly worse in terms of signatures representation. The meteorological characteristics are known at subcatchment scale and therefore, due to the configuration of the model, they are not suitable for the subdivision in HRUs.

iv) We will consider this comment when we will rewrite paragraph 4.1.5.

e. *the whole structure of the model building story is a bit complicated as aspects are described in chapters 3.1.3, 3.3, 4.1.4 und 4.1.5 which makes it difficult to follow. I suggest combining them into a single chapter. Therein start with the theory e.g. snow is important followed by the surrogate you considered it e.g. half stream flow period. Or geology is important due to… –> different HRUs.*

We will improve the readability of the paper making clearer the process of model building. It was divided in different paragraphs along the paper in order to emphasize the connection between data analysis and modeling choices but we understand that it makes more difficult to follow the story. We will keep it in mind when we will rewrite chapter 4.1.5.

4. *Several conclusions are not appropriate: e.g. "the proposed approach is useful in the fact that modelling can be used to test specific hypotheses on dominant processes resulting from regression analysis" (p. 19 l. 4). This has not be shown. More over aspects related to the event scale are mentioned in the first three bullet points but not subject to the manuscript. In the third bullet point you state: "Higher proportion of consolidated material has an influence on the baseflow vs quickflow portioning, causing lower baseflow and higher peaks" (p 19 l 14). Does the*

*study provide evidence for this statement or does it support this hypothesis? I expect the latter and missed this statement in the chapter 3.1.1. I suggest re-writing of the entire section and to differentiate concisely between hypothesis, results and conclusions.*

We will be more precise in our conclusions. With respect to the first point, we think that the model comparisons have been useful to confirm the interpretations of the regression analysis. Clearly the regression between variables is also a model, but the hydrological model is an integrated model that is meant to explain all dependencies at once, whereas the regression model provides a separate model (regression) for each of the dependencies. Therefore there is an added value in the hydrological model, compared to the regression model.

With respect to the use of "event" in the first three bullet points, we will be more precise and refer directly to the signatures rather than to the time scale.

In terms of the baseflow vs quickflow, we will clarify that we refer to the baseflow index, and to how the model tries to mimic it by varying the partitioning of water between fast and slow reservoirs.

5. *The model performance evaluation (chapter 4.1.4) is complicated but of minor importance in this context. I suggest shorting the evaluation section and to focus on a single, interpretable metric e.g. the Kling-Gupta Efficiency as the NASH has several limitations and the normalized log-likelihood is difficult to interpret. But this is a minor point and a matter of taste.*

We agree that the Nash-Sutcliffe efficiency has several limitations, as any individual index is somehow limited. This is why we have introduced several signatures to evaluate model results. Indeed, we could see that a significant improvement in some of the signatures could result into a negligible improvement in the Nash- Sutcliffe efficiency.

*Technical corrections (figures and tables only) I only provide technical corrections for the figures and tables as I expect that several parts of the manuscript will be subject to major revisions.*

Thank you for the comments for improving the quality of our figures and tables; we will address them bellow.

- *Figure 1: A: I suggest to remove the colour code and to provide notations (abbreviation) in or around the map. This would help improve the readability of the stream network and the location of the gauges. If you want to keep the legend please add catchment abbreviations to it, order it according to Fig 2. and use a meaningful colour code (e.g. mean annual precipitation, elevation or geology), B: Try a discrete legend like in atlases, will improve readability. C: Forest and pasture are hardly distinguishable both on my screen and in a printed version.*

  Figure 1A: We agree that there are some problems with the readability of the river network but they are mainly due to the poor resolution. In the final version we will upload the figures separately with an higher resolution. The presence of the legend doesn't make the figure smaller since the constraint is the height and not the width of the box; we will then keep the legend sorting the names according to the other figures and using abbreviations. The colors used for the single catchments were chosen from a "categorical" color scale in order to be as different as possible. Linking them to some characteristic would mean using a "sequential" color scale, with

little difference between subcatchments, and this would be problematic in the other figures (assuming that we want to be consistent) where we want to clearly see the behavior of the single catchments.
Figure 1B: we will improve the figure according to the suggestion
Figure 1C: we will change the colors to improve the readability and the figures will have better resolution.

- *Figure 2: Please repeat the variables and their abbreviation in the caption such that the figure can be read independent from the text. Maybe add another row and provide grouping indices based on the results in chapter 3.2.1*
  As the names are relatively long, they would not fit on the y axes. Instead, we have opted to place them in the title of the subplots.

- *Figure 3: Please repeat the variables and their abbreviation in the caption such that the figure can be read independent from the text.*
  See reply at earlier point.

- *Figure 4: Please repeat the variables and their abbreviation in the caption such that the figure can be read independent from the text. Information on the range of the different variables would be pretty helpful as well. If possible include it otherwise please mention the ranges in the text or add the information to table 1.*
  In order to make the figure clearer, we will report the meaning of the variables in the caption and group them according to the category that they represent. The range of the variables is always between 0 and 1: all the variables plotted are percentage of the area of the subcatchment occupied by a certain characteristic. The characteristics that don't belong to the category "part of the catchment occupied by …" are reported in table 1.

- *Figure 5: I'm not sure if this figure is required since B and C show very little variation. The only important message from A is that there a catchments that are stronger controlled by snow than others. I suggest removing it. If you decide to keep it update the colour code according to the suggestion for Figure 1.*
  Although the plots B and C show little variability across the catchment, it is still interesting to present the seasonal dynamics. Moreover, we consider that it is useful to show that the monthly variability in streamflow (plot A) is not directly ascribed to variability in precipitation or potential evaporation (plot B or C).

- *Figure 6: I cannot find a description of the symbols and abbreviations in the Appendix. Please specify at least the meaning of the capital letters in the caption (as in 4.1.1) and provide a more comprehensive description in the appendix.*
  We will put a detailed description either in the caption or in the appendix.

- *Figure 7: Order according to Figure 2 or 3. Line type and colour code are redundant.*
  We will reorder plot B according to figure 2. We will substitute the dashed line with a continuous thin line (same for all colors).

- *Figure 8, 9, 10: Nice figures! Suggestions: Combine all three figures in one (each model setup as an individual row). This would improve readability. Streamflow, runoff coefficient and half streamflow period have no or little variation (two out of these could be omitted such that all results would fit in one figure). Remove the correlation coefficients due to their distracting nature*

*(correlation (alone) is pretty meaningless in this context). Update colour code according to Fig 1 A.*

Point taken. We acknowledge that is more meaningful to put the different models together in order to facilitate the comparison. Since we will reduce the number of signatures in the analysis, there will be enough space to host all of them in the same figure.

- *Table 1: Order according to fig 1 A. Index column is not relevant, omit Code or put it to the very right. Rounding is not yet meaningfully and consistent.*
  We will reorder the table according to the figures. The Index column is used in the "upstream catchments" column to define the river network. The "code" column is present to avoid ambiguity with the naming of the gauging stations providing the reader with the code of the gauging station used by the Federal Office for the Environment FOEN. The rounding is meant to maintain the same number of significant digits, and it is consistent with how the FOEN gives the values in the website (no decimal digits if the area is greater than 100 km$^2$, one digit if it is between 10 km$^2$ and 100 km$^2$, two digits if it is lower than 10 km$^2$).

- *Table 2: This table includes variables with spurious correlations (Brett 2004, Kenny 1982). This includes variables that are considered statistically significant and where causality was assumed e.g. the correlation between aridity index AI and the runoff coefficient RC which are both are derived from precipitation. The same applies for P and RC. Since P and Q are highly correlated and AI is based on P I also wonder about the significance of AI and RC, BFI, FI and HDP. Please clarify. Please also explain why you assumed causality among LP and BFI and among LP and FI? Differences among the geological fractions are small as well. Why do you consider causality in some of the individual relationships and in others not?*
  This table will be modified during the review of the paper taking into account this comment and the changes in the text.

- *Table 3: This analysis also includes variables with spurious correlations. Please comment on that.*
  This table will be modified during the review of the paper taking into account this comment and the changes in the text.

- *Table A1: Please provide a brief explanation on parameters and components. Where does the range of variability come from?*
  We will describe the meaning of the columns in the caption and add more description in the appendix.

- *Table A2: Explain component*
  We will describe the meaning of the columns in the caption and add more description in the appendix.

*Literature*

*Brett, M. T. (2004). When is a correlation between non-independent variables "spurious"?*

*Oikos, 105(3), 647–656. Kenney, B. C. (1982). Beware of spurious self-correlations! Water Resources Research, 18(4), 1041–1048. https://doi.org/10.1029/WR018i004p01041*

---

## Author Comment (AC2) · 22 May 2019

**Reply to review by Lieke Melsen**

We thank Dr. Lieke Melsen her careful read to the manuscript and insightful suggestions.

After reading carefully the comments of both the reviewers, in order to address some of their most substantial comments, there will be some major changes in the revised version of the manuscript. These changes will affect most significantly the data analysis part of the paper (section 3), and they are summarized hereafter:

- We will extend the selection of hydrological signatures in order to have a more comprehensive picture of the hydrological behavior of the catchments.
- We will select only signatures that are not correlated (reducing substantially their number) for the subsequent analyses.
- We will select a reduced number of catchment and meteorological indices in order to reduce the problem of correlated features.
- All the correlation analysis will be based on the Spearman's rank score since, as pointed out by Dr. Melsen, the usage of Pearson may not be adequate.
- Following this decision, we will remove the regression analysis (table 3) since (1), as stated by the Anonymous Referee #1, " it does not add any new knowledge to the literature occupies a lot of space" and because (2) the usage of linear regression (that looks for linear dependencies in the data) is not coherent with the choice of basing the analysis on the Spearman's rank score (which accounts also for nonlinear correlations).

Below, we answer in detail the various comments, and illustrate our intended approach to address them in the revised version. The original comments of the reviewer are reported in *black and italics*, our replies in blue.

*Dal Molin et al. investigated, through regression analysis, which indices have explanatory power for streamflow response. Based on the insights gained from the regression analysis, different spatial configurations were implemented in a hydrological model. Although I find the work flow elegant, starting from process-understanding and translating that to the spatial configuration of the model, I have some problems / concerns with the regression set-up.*

*Major*

*My main concerns are all related to the regression-part of the study.*

1. *It is unclear how the indices, on which regression was applied, were selected. There is plenty of literature around on indices and signatures, which could guide indices selection, but I don't see any justification in the text for the choices made. Check for instance:*
   *Addor et al., A Ranking of Hydrological Signatures Based on Their Predictability in Space, WRR, 2018*

*Knoben et al., A Quantitative Hydrological Climate Classification Evaluated With Independent Streamflow Data, WRR, 2018*

We thank the reviewer for the references she provided. We will use them to better contextualize, and potentially expand, our selection.

The signatures used in this work were chosen in order to represent a wide variety of hydrograph characteristics. There are, for instance, two signatures designed to represent the long-term water balance (average streamflow and runoff coefficient), two signatures to capture the "responsiveness" of the hydrograph (baseflow index and flashiness index), and one (the half streamflow period) that is designed to understand seasonality effects, like the ones related to the snow dynamics.

The indices used (here by indices we mean catchment and meteorological indices) were selected in order to have a large group of possible influence factors to start the analysis with. Catchment characteristics capture topography, soil properties, land use, and geology while meteorological characteristics take into account precipitation and potential evapotranspiration.

We will improve the description of the indices, including the motivation behind their selection and integrate or modify the list of indices also according to what has been proposed by the other reviewer.

2. *Since the choice of the indices is not well justified, I am worried about their mutual correlation. Many indices can describe the same signal. Therefore, please provide the correlation among the indices themselves. This might lead to the insight that you need fewer or different ones.*

Mutual correlation, as pointed out by the reviewer, is a potential criticality of this study. Indeed there are multiple indices that describe the same signal (or very similar signals) and this possibility was considered in the design of this study. We decided to keep them all because we didn't want to restrict a priori the space of possible influence factors. However, we will check, in the revised version, for mutual correlation between indices before carrying out the correlation with hydrological signatures.

We will add tables with the correlations between indices themselves in the revised version to make clearer the choices made in section 3.2.2.

3. *It was rightly mentioned that correlation does not mean causality. It was claimed (not only in the methods, but also in the conclusions) that this study accounts for that by only selecting the indices that have a causal relation, based on expert judgement. I do, however, not recognize the expert judgement in the selection of significant indices, and this actually directly relates to my point 2. Right now, the selection seems to be made based on the mutual correlation of the indices – so the mutual correlation was investigated! – but I don't see any process-reasoning (the expert-judgement) that can justify the selected indices, and that justifies the claim that there is really causality.*

The analysis of the correlations was only the starting point of the process of selection of the indices. Starting form that, we then discarded the indices that are either redundant (e.g. average slope vs. fraction of steep areas) or accidental correlations (e.g. using elevation instead of precipitation, line 18 page 7). This step was essential to move from mere correlation to causality and it involved "expert judgment" in order to prune reasonably the list of indices. An example of

the "expert judgment" process is illustrated in paragraph 3.2.2, page 10, lines 1 to 10, where we showed how the indices were selected for the mean streamflow signature.

We will improve our text (especially paragraph 3.2.2) in order to highlight the role of "expert judgment" in the process of indices selection. We will also try to reduce the dimensionality of the problem selecting a sub-set of signatures, indices, and characteristics before doing the correlation analysis.

4.  *I disagree with the conclusion of the authors that there is no need to look for nonlinearity in the correlation, based on the results in the table. The authors rightly state that only few correlations that are statistically significant based on Spearman are not significant with Pearson, but how do the authors explain the opposite effect? Quite some correlations are significant with Pearson but not with Spearman, is this a Type 2 error in the Pearson test? That could have consequences, for example, aridity was significantly correlated with BFI for Pearson, but not Spearman, and based on 'expert judgement' included in the regression.*

    The reviewer is right pointing out the possibility of type 2 error for some correlations, motivated by discording conclusions between Spearman and Pearson correlation. This may be due to the fact that the Pearson correlation was calculated neglecting the assumptions behind it (e.g. normality of the data) that may not be respected in this case.

    We will remove form the revised version the usage of Pearson correlation and use only Spearman's rank since it relaxes some of the assumptions made by Pearson and it also detects nonlinear correlations.

5.  *1 of the 3 points of the guidelines for modelling based on the regression was not based on the regression at all, namely the conclusion that the presence of snow is relevant. Please include a snow-related indicator in the regression to support this conclusion (based on expert judgement we can expect this, of course).*

    The reviewer is right saying that it is not clear that "the conclusion that the presence of snow is relevant" was motivated by the regression analysis. This is due to the fact that we didn't explain earlier in the text that the "half streamflow period" signature and the plots in figure 5 were used to show the presence of seasonality in the streamflow dynamics. In particular, there are some subcatchments that reach their peaks of streamflow in different periods of the year.

    This seasonality is not due to different patterns in precipitation or PET (as shown in figure 5) and correlates well with the elevation (higher subcatchments reach half of their streamflow later in the year). These two points made us think that this seasonality is due to snow dynamics and that, therefore, the model should take them into account; higher catchments are subjected to snow that is then released in the streamflow (as snowmelt) later in the year if compared with rain-dominated subcatchments.

    We should mention earlier in the paper the possibility of snow playing a role in this catchment and that therefore the signature and figure 5 are used to capture this effect.

6.  *It depends a bit on the definition of model building, but the title and the text might give the impression that the model structure itself was adapted with the insights in the regression, while it was basically the model implementation (accounting for HRU's or not) that was adapted.*

    It is true: it depends on the definition of model building. For us it incorporates all the decisions taken in order to have an hydrological model for the Thur catchment. In particular:

- How to spatially divide the inputs
- How to divide the catchments in HRUs
- Structure of the single HRUs

All these points where considered in the construction of the hydrological model and were informed by the regression analysis. The last point (structure of the bucket model) was also considered in the procedure of building the hydrological model but was not discussed in this paper because it was already done in previous studies and for sake of brevity.

We will make more clear what do we mean for model building in the introduction of the paper.

*Minor*

*Section 3.2.1, the catchments are sort of grouped based on their stream flow response, but this is not used any further in the analysis. Consider to just briefly describe their response, or to use the grouping later to explain results (in that case, also display the groups in the figure).*

Section 3.2.1 describes the signatures in the catchments without relating them with the indices. We agree with the reviewer that the subdivision done here is no more used in the paper and it was done only for convenience when describing the signatures. We will remove this division since it is not relevant for the rest of the paper.

*In the same section, it seems unnecessarily complicated to use combinations of signatures to determine how flashy catchments are; a flow duration curve can generally provide quite some insight on this already (slope of flow duration curve also frequently used signature)*

We acknowledge the possibility of using other signatures to describe the behavior of a catchment but we used, among the others, baseflow index and the flashiness index because believe that they are more interpretable and they can be related to dynamics represented by the model; the BFI, for example, can be linked to the separation between quick and slow flow that is a process that is present also in the hydrological model. The selection of the signatures will be revised in the new version of the paper also according to the previous points and to the other reviewer.

*Provide an overview of the indices and their abbreviations, or include their full name more often in text / tables / figures, because now it requires quite some work from the reader to fully understand all sentences and figures (and a lot of going back to the methods).*

We understand that the usage of abbreviations may complicate the reading of the paper but, on the other side, their usage helps avoiding misunderstandings that may happen when calling the same index with different names. The full name of the indices is provided in table 2 and in section 3.1.1 and it will be reported in the caption of the figures when not reported in the figure itself.

*A large number of figures is dedicated to showing the signature-values, which is not of direct relevance. I would be interested to see a figure that displays the HRU's.*

We acknowledge that a figure representing the HRUs used in M2 and M3 is missing but it can be deducted from figure 1 (plot "d" for M2 and "c" for M3) since the HRUs were constructed aggregating

some classes (for example, for M2, one HRU is composed by the orange part and the other by the rest of the catchment). We think that the figures dedicated to showing the signature-values are important for this study to show the high variability present in the catchment response.

*For the landscape characteristics, it is not mentioned in the methods-section (3.1.1) that you consider fractional area. Please clarify there, as I was wondering how you would apply regression on nominal values, until I found out in the results that you considered frac. area.*

We will clarify it in the text.

*The sentence 'optimizing the parameter of the posterior distribution' (l.11, p13) can give the impression that you minimized e.g. variance (describing distribution), please consider reformulation.*

We need to clarify it in the text. The actual meaning is "optimizing the parameters of the hydriological model and of the error model (refer to section 4.1.1 and 4.1.2) in order to find the ones that maximize the posterior distribution"

*Although overall written well and clear, some language editing seems required, for example "The average value oscillates of about ..", (but I'm not a native either).*

We will do our best to improve it.

*Overall, I appreciate the intent of the study and the modelling-part seems well designed (except for my question at point 6 which remained unclear), but I do believe the regression-part requires substantial revision, related to the selection of indices (more embedded in literature and account for snow) and to justify the use of the word 'causality'. Given that the work-flow is largely set-up, I think the authors should be able to incorporate this.*

---

## Author Response (AR1)

**Author's response**

**Changes in the paper**

The paper has been subject to major revision in order to address the comments of the reviewers. The most significant changes concern the signatures analysis, where we introduced an extended selection of signatures, followed by an approach for selecting significant signatures, and a new model experiment (M0), aimed at demonstrating causes for differences in streamflow seasonality. A summary of the changes is presented below, followed by the individual responses to the reviewers.

| SECTION     | DESCRIPTION                                                                                                                        |
|-------------|------------------------------------------------------------------------------------------------------------------------------------|
| 1           | Minor changes to keep consistency with the rest of the paper.                                                                      |
| 3.1 and 3.2 | Completely restructured to address the comments of the reviewers:                                                                  |
|             |  <li>The list of signatures and indices considered was extended.</li>                                                     |
|             |  <li>Correlation analysis based on Spearman instead of Pearson correlation to
account for nonlinearities.</li>        |
|             |  <li>A pre-selection of signatures and indices based on correlation analysis and</li>                                     |
|             | expert judgment was done to avoid spurious correlations in the subsequent analyses.                                                |
|             | • Climatic and catchment indices that drive streamflow variability were identified using correlation analysis and expert judgment. |
| 3.3         | Adapted to the new findings.                                                                                                       |
| 4.1.1       | Completely restructured to address the concerns of the reviewers about clarity.                                                    |
| 4.1.4       | Minor changes to keep consistency with the rest of the paper.                                                                      |
| 4.1.5       | Completely restructured to address the concerns of the reviewers about clarity. New                                                |
|             | model M0 included to test the importance of snow-related processes.                                                                |
| 4.2.1       | Minor changes to include M0.                                                                                                       |
| 4.2.2       | Major changes due to the new group of signatures considered.                                                                       |
| 4.2.3       | Minor changes to keep consistency with the new findings of section 4.                                                              |
| 5           | Minor changes to keep consistency with the rest of the paper.                                                                      |
| 6           | Major changes to address the concerns of the reviewers.                                                                            |

**Changes in text**

**Changes in Figures**

The following figures were eliminated or modified (note that the numbers refer to the old version of the paper)

| FIGURE | DESCRIPTION                                                                |
|--------|----------------------------------------------------------------------------|
| 1      | Panels (a), (b), and (c) modified to address the concerns of the reviewer. |
| 2      | Eliminated.                                                                |
| 3      | Eliminated.                                                                |
| 4      | Eliminated.                                                                |
| 5      | Eliminated.                                                                |
| 7      | Added M0.                                                                  |
| 8      | Eliminated.                                                                |
| 9      | Eliminated.                                                                |
| 10     | Eliminated.                                                                |

The following figures were created (note that the numbers refer to the new version of the paper)

| FIGURE | DESCRIPTION                                                                           |
|--------|---------------------------------------------------------------------------------------|
| 2      | Internal correlation between the streamflow signatures.                               |
| 3      | Internal correlation between the climatic indices.                                    |
| 4      | Internal correlation between the catchment characteristics.                           |
| 5      | Correlation between the selected streamflow signatures and the selected climatic and  |
|        | catchment indices.                                                                    |
| 8      | Influence of the model structure on the representation of the mean half streamflow    |
|        | day.                                                                                  |
| 9      | Ability of the models of representing the signatures.                                 |
| 10     | Ability of the hydrological models of representing the signature duration of low-flow |
|        | events.                                                                               |

**Changes in Tables**

The following tables were eliminated or modified (note that the numbers refer to the old version of the paper)

| Table | DESCRIPTION                                                                   |
|-------|-------------------------------------------------------------------------------|
| 1     | Some columns were eliminated because they were included in table 4 of the new |
|       | version of the paper.                                                         |
| 2     | Eliminated.                                                                   |
| 3     | Eliminated.                                                                   |
| A1    | Minor changes to address the comments of the reviewers.                       |
| A2    | Minor changes to address the comments of the reviewers.                       |
| A3    | Minor changes to address the comments of the reviewers.                       |

The following tables were created (note that the numbers refer to the new version of the paper)

| FIGURE | DESCRIPTION                                 |
|--------|---------------------------------------------|
| 2      | Values of the streamflow signatures.        |
| 3      | Values of the climatic indices.             |
| 4      | Values of the subcatchment characteristics. |

**Reply to review by Anonymous Referee #1**

We thank the reviewer for his/her careful read to the manuscript and insightful suggestions. Below, we answer in detail the various comments, and illustrate how we have addressed them in the revised version. The original comments of the reviewer are reported in *black and italics*, our replies in blue and the changes to the paper in **green and bold**.

Note that, were not indicated, the numbers of figure/tables/lines/pages refer to the old version of the paper.

The authors propose to infer the structure of a hydrological model based on landscape and process characteristics (signatures) of the catchment. In the first of a two-stage process different landscape and catchment characteristics are compared to different streamflow signatures to identify the most important controls on runoff formation. In the second step this information is used "as an inspiration for model structure design" (p17, I. 32) as the authors put it.

Inferring structure from function (or vice versa) is at the core of hydrological model building and subject to numerous studies. The topic is hence highly relevant for the hydrological community. The manuscript is well structured and well-written. Accordingly the manuscript is suitable for a publication on HESS. However, I cannot recommend publishing to current version of the manuscript due to several major points:

- The purpose of the modelling exercise is not clear. Model requirements for flood forecasting are
  e.g. totally different from model requirements to simulate climate change. The relevant
  signatures, temporal and spatial model discretisation, model evaluation metrics and also the
  degree of model conceptualisation differ accordingly. Please specify more clearly the purpose of
  you modelling study. Otherwise it is not possible to evaluate the study meaningfully.
  In the introduction, we already specified (lines 6-10, page 3) the main objectives of the study,
  which in summary consist of proposing a model building strategy which starts from an analysis
  of the data, which provide a basis for motivating the various model decisions.
  As the reviewer noted, the purpose of the model itself has remained unclear. Although one of
  the main objectives of models is making predictions to address some practical issues, here we
  do not have such an immediate applied objective. The model exercise is mainly an instrument to
  help understand and interpret catchment scale processes that dominate catchment
  response, and that mostly influence the observed spatial variability in streamflow behavior, as
  - characterized by the set of suggested signatures.

The reviewer is right in the fact that aspects such as "signatures, temporal and spatial model discretization, model evaluation metrics and also the degree of model conceptualization differ accordingly".

We have addressed this comment specifying (line 5, page 3 in the new version of the paper) that the objective is explaining the observed spatial diversity of streamflow characteristics, with the minimum possible complexity, while trying to maintain a process based interpretation.

- 2. I consider the selection, evaluation and identification of landscape characteristics as fairly weak due to a number of different reasons:
  - a. The authors provide no information about why certain characteristics were selected (and why others were not). Catchment characteristics (or signatures) can only provide information on the underlying processes if they have some kind of diagnostic potential or causal relationship. It is clear that these relationships are often unknown and difficult to obtain; nevertheless the selection of appropriate characteristics is vital for the identification of underlying processes and mechanisms. I miss a clear and elaborate description on the selection of catchment descriptors and on their expected diagnostic potential (both in space and time): E.g. why or how can the different land cover ratios or aspects help to derive information on hydrological processes? Are the same characteristics suitable for all catchments (independent of size, altitude, geology)? Please also comment on the importance of the time step e.g. you calculated the flashiness index based on daily streamflow data, although you state that streamflow can change two orders of magnitude in a few hours (p. 3 l. 18). If this is true please explain why you consider a daily-data based flashiness index as a meaningful variable? Please do also explain why you think that "half streamflow period" is a suitable parameter to discriminate to importance of snow. I expect that there are much simpler and more meaningful variables such as temperature and rain, temperature sums or snow data itself to describe the importance of snow. The results in Fig 7-9 also show that streamflow, runoff coefficient and half streamflow period are pretty identically in all cases. Do you consider them being suitable signatures? Please also provide more signature papers in the introduction as the number of up to date references is small. The reviewer correctly points out that we "miss a clear and elaborate description on the selection of catchment descriptors and on their expected diagnostic potential (both in space and time)". Catchment characteristics can affect the hydrological cycle: vegetation characteristics, for example, are typically assumed to affect evaporation, soil characteristics are typically assumed to influence the partitioning of water between retention and runoff. In general, we tried to select a broad class of characteristics, to be as inclusive as possible. However, it is also true that these characteristics can be represented through a large class of indices, and in order to reduce the size of the problem, some choices had to be made.

We have complemented in section 3.1.1 of the new paper the selection of catchment characteristics with their expected diagnostic potential. We have also motivated some of our decisions based on how other models have dealt with similar issues.

We used a daily data resolution, and this choice clearly affects some of the signatures. As the reviewer points out, the flashiness index is one of such signatures. The values of the flashiness index reduces with increasing time step due to a smoothing effect. In this paper, we did not experiment with varying data resolution, as it was outside our scope. We have commented (line 25-30, page 4 of the new version of the paper) about this choice relating it to findings of previous modeling studies. We experimented with several signatures to account for the effect of snow on streamflow seasonality, and we ended up selecting the "half streamflow period". The reason is that 1) this signature was used in previous publications to quantify streamflow seasonality, and we did not want to invent our own signature if something was already existing, and 2) this signature captured well the difference between streamflow regimes, because we have seen (figure 5) that all the catchments receive similar precipitation input (in terms of monthly variability) but the snow-affected ones show the peaks during late spring/beginning of summer while the rain dominated ones show their peaks during the winter and the spring.

Figures 7-9 show that all the model configurations represent well the yearly streamflow, the runoff coefficient, and the half streamflow period and this is a result of our study that is also coherent with our assumption that only distributing the inputs (precipitation, PET, and temperature) is sufficient in order to have a model that captures the water balance and the snow dynamics.

We have extended the list of signatures considered in this study and used correlation analysis between the signatures to select only the not redundant ones (major changes in section 3 of the new version of the paper).

- b. In your study you included several (fairly easy to derive) landscape characteristics that are obviously highly correlated and describe in great detail how you identify and select appropriate ones based on regression and correlation. In my opinion a rather trivial part which does not add any new knowledge to the literature occupies a lot of space. I hence suggest shortening and streamlining the entire section. If you want to derive structure from function than the first goal must be to derive a (comprehensive) matrix of uncorrelated catchment characteristics that have some kind of diagnostic potential. In my opinion this should be the source of the story and not a result.
  Section 3 of the new version of the paper has been completely restructured to address (also) this comment. The lists of meteorological and climatic indices have been reduced before evaluating the correlation with the streamflow signatures; the regression analysis has been eliminated.
- 3. The approach for informing model structure does not appear very elaborate to me. Since this is the core of "model building for understanding catchment process" I particularly miss a clear and elaborate discussion on how the identified landscape characteristics help in the model building process. More specifically:
  - a. In chapter 3.1.3. you state that the results of the regression analysis were used to build the hydrological model e.g. the subdivision of the catchment in HRUs (p. 7 l. 32). Later, in 4.1.1 you state that subdivisions were defined by gauge locations (p. 11 l 26). I did not find information on how you derived the number of HRUs and the role of catchment characteristics in this context? Chapter 4.1.1. should be more comprehensive in this regard.

Our intention was to present chapter 4.1.1 as a general overview of the model structure in order to make the following clearer. The information from the regression analysis are used to derive the HRUs is described in chapters 3.3 and 4.1.5. It is important to make clear the difference between the division in subcatchments (areas that have uniform inputs) and HRUs (areas that have the same hydrological response). The former are defined by the presence of gauging stations (and this division is not negotiable) while the latter reflect our understanding of the catchment functioning (and, in this study, of the regression analysis).

Sections 3.3, 4.1.1, and 4.1.5 in the new version of the paper have been restructured in order to clarify the difference between subcatchments and HRUs and to make the connection between the findings of the correlation analysis and the modeling experiments clearer.

b. The argument that "the regression analyses have indicated that precipitation is a dominant control on average streamflow" (p. 12 l. 4) is trivial. I don't think you need this and particular not as a justification for using spatially distribution rainfall as a model input. From your manuscript it appears to me that the spatial discretization of your model was based on the definition of the subcatchments (which are in turn defined by the location of gauges) and according the definition of fields (definition not clear). In consequence I don't see that landscape characteristics played an important role in this process. Please clarify?

Although it may be a priori clear that precipitation needs to be distributed per subcatchment, it may be not as taken for granted that this is sufficient to capture the water balance of the subcatchments, as many other aspects could in principle play a role (e.g. regional groundwater flow). Here we show that considering distributed precipitation over the subcatchments (defined by the presence of the gauging stations) could by itself be sufficient. Other landscape characteristics play a role in the definition of the HRUs.

Sections 4.1.1 and 4.1.5 in the new version of the paper have been modified to address this comment; in particular, M0 shows that distributing only the precipitation without accounting for snow related processes is sufficient to capture the average streamflow.

c. You also mention that "the parameters were motivated by the results of the regression analysis" (p. 8 l. 1). Please omit or explain more detailed. A matrix to illustrate the relationships between model parameters and catchment descriptors would be good. I would for instance be interested in how one could use catchment descriptors to derive (or at least constrain) model storage (kFR or kSR) or network lag (trise,IL trise,OL) parameters. Please comment on that

All the process of building the model was motivated by the results of the regression analysis (in particular the decisions on the division in HRUs). The parameters are just calibrated using streamflow data (section 4.1.3). No inference of the parameters from catchment characteristics was done.

We have changed the misleading sentence (lines 17-20, page 9 of the new version of the paper)

d. Chapter 4.1.5 is difficult to me due to different reasons: i) your analysis does not VERIFY that "models that account for influencing factors ... lead to an improved representation".

Essentially it only shows that a complicated model (with a larger number of degrees of freedom) outperforms a simpler model (with a smaller amount of degrees of freedom). Please use precise wording. It addresses the question of adequate model complexity. If a lumped representation (M1) is not adequate than also the comparison of M2 to M1 is not adequate. Please explain in more detail why you consider M1 a suitable reference? ii) Please explain why unconsolidated areas receive an individual HRU and why consolidated and alluvial areas can be lumped together (what are your expectation on the underlying processes)? iii) The parameterization of M3 is based on land use, which is not considered to have a causal relationship to the streamflow signatures (Table 2). Please explain why a model which is derived from non-causal properties can be a considered a meaningful reference? Why did you group based on geology and not on elevation, slope or the aridity index which you considered to have a causal relationship? This would maybe be a more appropriate benchmark? iv) Essentially chapter 4.1.5 addresses the questions of optimal degree of model complexity and optimal degree of spatial discretization - which are both very important. However, these aspects are treated together and not separated from each other. Moreover, potential answers to these questions miss a clear link to catchment descriptors. Essentially only differences in geology were considered in the model building. Please clarify to novelties of your study more clearly.

Essentially the two main model configurations are M1 and M2: the first is the baseline and it is a semidistributed model (in the sense that the inputs are spatially distributed and the routing between subcatchments is explicitly addressed in the model) with only one HRU (meaning that all the catchment responds in the same way to the forcings); the second extends the first providing a subdivision of the subcatchments in two HRUs. M3 is used to show that the subdivision in HRUs has to be carefully made otherwise a more complex model doesn't imply automatically better results. Answering to the specific points:

i) M2 is indeed more complex than M1 but our thesis is that its better performance is not just due to the fact that it is more complex but to the fact that it incorporates the right catchment characteristics. This is also demonstrated by M3 that is as complex as M2 but it has the same deficiencies of M1. M1 is already a quite complex model since it already considers the spatial distribution of the inputs and incorporates information about the routing between subcatchments. The real baseline would have been a lumped model, with uniform input and no information about the catchment characteristics but it was too simple for the comparison.

ii) There is an error in the text: the two HRUs are unconsolidated and alluvial (HRU1) vs consolidated (HRU2). Alluvial and unconsolidated geology were put together because they show a similar behavior in terms of water dynamics in the sense that they both represent areas with high storage capacity, especially if compared with HRU2 that is quite impermeable.

iii) M3 was designed to demonstrate that M2 outperforms M1 not just because it is more complex but only because it incorporates characteristics that actually have an

impact on the response of the catchment. For this reason we used a model with the same complexity of M2 but based on characteristics that don't correlate with the streamflow signatures. Also topography was considered in the modeling experiment (but not reported in the paper), experimenting with a subdivision in HRUs based on the slope, but the model resulted similar to M2 (in terms of spatial discretization) but slightly worse in terms of signatures representation. The meteorological characteristics are known at subcatchment scale and therefore, due to the configuration of the model, they are not suitable for the subdivision in HRUs.

We have completely restructured the section 4.1.5 with the intent of making clearer the differences between the modeling experiments and the reasoning behind them. We have also introduced a new model (M0) to test the effect of snow on the seasonality patterns.

 e. the whole structure of the model building story is a bit complicated as aspects are described in chapters 3.1.3, 3.3, 4.1.4 und 4.1.5 which makes it difficult to follow. I suggest combining them into a single chapter. Therein start with the theory e.g. snow is important followed by the surrogate you considered it e.g. half stream flow period. Or geology is important due to... -> different HRUs.

It was divided in different paragraphs along the paper in order to emphasize the connection between data analysis and modeling choices but we understand that it makes more difficult to follow the story.

Sections 3.1.3, 4.1.1, and 4.1.5 have been restructured in order to improve their readability and to show more clearly the connections between the three sections.

4. Several conclusions are not appropriate: e.g. "the proposed approach is useful in the fact that modelling can be used to test specific hypotheses on dominant processes resulting from regression analysis" (p. 19 I. 4). This has not be shown. More over aspects related to the event scale are mentioned in the first three bullet points but not subject to the manuscript. In the third bullet point you state: "Higher proportion of consolidated material has an influence on the baseflow vs quickflow portioning, causing lower baseflow and higher peaks" (p 19 I 14). Does the study provide evidence for this statement or does it support this hypothesis? I expect the latter and missed this statement in the chapter 3.1.1. I suggest re-writing of the entire section and to differentiate concisely between hypothesis, results and conclusions.

With respect to the first point, we think that the model comparisons have been useful to confirm the interpretations of the regression analysis. Clearly the regression between variables is also a model, but the hydrological model is an integrated model that is meant to explain all dependencies at once, whereas the regression model provides a separate model (regression) for each of the dependencies. Therefore there is an added value in the hydrological model, compared to the regression model.

The conclusions of the paper have been revised avoiding aspects related to the event scale and preferring an analysis of the signatures.

5. The model performance evaluation (chapter 4.1.4) is complicated but of minor importance in this context. I suggest shorting the evaluation section and to focus on a single, interpretable metric

e.g. the Kling-Gupta Efficiency as the NASH has several limitations and the normalized loglikelihood is difficult to interpret. But this is a minor point and a matter of taste. We agree that the Nash-Sutcliffe efficiency has several limitations, as any individual index is somehow limited. This is why we have introduced several signatures to evaluate model results. Indeed, we could see that a significant improvement in some of the signatures could result into a negligible improvement in the Nash-Sutcliffe efficiency.

Technical corrections (figures and tables only) I only provide technical corrections for the figures and tables as I expect that several parts of the manuscript will be subject to major revisions.

Thank you for the comments for improving the quality of our figures and tables; we will address them bellow.

• Figure 1: A: I suggest to remove the colour code and to provide notations (abbreviation) in or around the map. This would help improve the readability of the stream network and the location of the gauges. If you want to keep the legend please add catchment abbreviations to it, order it according to Fig 2. and use a meaningful colour code (e.g. mean annual precipitation, elevation or geology), B: Try a discrete legend like in atlases, will improve readability. C: Forest and pasture are hardly distinguishable both on my screen and in a printed version.

Figure 1A: We agree that there are some problems with the readability of the river network but they are mainly due to the poor resolution. In the final version we will upload the figures separately with an higher resolution. The presence of the legend doesn't make the figure smaller since the constraint is the height and not the width of the panel; The colors used for the single catchments were chosen from a "categorical" color scale in order to be as different as possible. Linking them to some characteristic would mean using a "sequential" color scale, with little difference between subcatchments, and this would be problematic in the other figures (assuming that we want to be consistent) where we want to clearly see the behavior of the single catchments.

Figure 1A : we have changed the figure keeping only the main rivers in order to improve the readability.

Figure 1B: we have improved the figure according to the suggestion

Figure 1C: we have changed the colors (darker green for the forest) to improve the readability.

• Figure 2: Please repeat the variables and their abbreviation in the caption such that the figure can be read independent from the text. Maybe add another row and provide grouping indices based on the results in chapter 3.2.1

As the names are relatively long, they would not fit on the y axes. Instead, we have opted to place them in the title of the subplots.

Figure 2 is not present in the new version of the paper.

 Figure 3: Please repeat the variables and their abbreviation in the caption such that the figure can be read independent from the text.
 See reply at earlier point.

Figure 3 is not present in the new version of the paper.

• Figure 4: Please repeat the variables and their abbreviation in the caption such that the figure can be read independent from the text. Information on the range of the different variables would be pretty helpful as well. If possible include it otherwise please mention the ranges in the text or add the information to table 1.

The range of the variables is always between 0 and 1: all the variables plotted are percentage of the area of the subcatchment occupied by a certain characteristic. The characteristics that don't belong to the category "part of the catchment occupied by ..." are reported in table 1. **Figure 4 is not present in the new version of the paper.**

• Figure 5: I'm not sure if this figure is required since B and C show very little variation. The only important message from A is that there a catchments that are stronger controlled by snow than others. I suggest removing it. If you decide to keep it update the colour code according to the suggestion for Figure 1.

Although the plots B and C show little variability across the catchment, it is still interesting to present the seasonal dynamics. Moreover, we consider that it is useful to show that the monthly variability in streamflow (plot A) is not directly ascribed to variability in precipitation or potential evaporation (plot B or C).

Figure 5 is not present in the new version of the paper.

• Figure 6: I cannot find a description of the symbols and abbreviations in the Appendix. Please specify at least the meaning of the capital letters in the caption (as in 4.1.1) and provide a more comprehensive description in the appendix.

We have put the description of the abbreviations in the caption of the figure

- Figure 7: Order according to Figure 2 or 3. Line type and colour code are redundant.
   We have added M0 to figure 7. The order of the catchment has been kept alphabetical since there is no more need to be consistent with figures 2 and 3.
- Figure 8, 9, 10: Nice figures! Suggestions: Combine all three figures in one (each model setup as an individual row). This would improve readability. Streamflow, runoff coefficient and half streamflow period have no or little variation (two out of these could be omitted such that all results would fit in one figure). Remove the correlation coefficients due to their distracting nature (correlation (alone) is pretty meaningless in this context). Update colour code according to Fig 1 A.

Point taken. We acknowledge that is more meaningful to put the different models together in order to facilitate the comparison.

We have put all the models side by side in the new figures 9 and 10.

• Table 1: Order according to fig 1 A. Index column is not relevant, omit Code or put it to the very right. Rounding is not yet meaningfully and consistent.

The Index column is used in the "upstream catchments" column to define the river network. The "code" column is present to avoid ambiguity with the naming of the gauging stations providing the reader with the code of the gauging station used by the Federal Office for the Environment FOEN.

The new version of Table 1 has a reduced number of columns and its primary goal has changed from describing the catchment characteristics to identifying the gauging stations and to describe the river network.

• Table 2: This table includes variables with spurious correlations (Brett 2004, Kenny 1982). This includes variables that are considered statistically significant and where causality was assumed e.g. the correlation between aridity index AI and the runoff coefficient RC which are both are derived from precipitation. The same applies for P and RC. Since P and Q are highly correlated and AI is based on P I also wonder about the significance of AI and RC, BFI, FI and HDP. Please clarify. Please also explain why you assumed causality among LP and BFI and among LP and FI? Differences among the geological fractions are small as well. Why do you consider causality in some of the individual relationships and in others not?

Table 2 is not present in the new version of the paper.

- Table 3: This analysis also includes variables with spurious correlations. Please comment on that. Table 2 is not present in the new version of the paper.
- Table A1: Please provide a brief explanation on parameters and components. Where does the range of variability come from?
   Table A1 has been modified to address this comment.
- Table A2: Explain component Table A2 has been modified to address this comment.

**Literature**

Brett, M. T. (2004). When is a correlation between non-independent variables "spurious"?

*Oikos, 105(3), 647–656. Kenney, B. C. (1982). Beware of spurious self-correlations! Water Resources Research, 18(4), 1041–1048. https://doi.org/10.1029/WR018i004p01041*

**Reply to review by Lieke Melsen**

We thank Dr. Lieke Melsen for her careful read to the manuscript and insightful suggestions. Below, we answer in detail the various comments, and illustrate how we have addressed them in the revised version. The original comments of the reviewer are reported in *black and italics*, our replies in blue and the changes to the paper in **green and bold**.

Note that, were not indicated, the numbers of figure/tables/lines/pages refer to the old version of the paper.

Dal Molin et al. investigated, through regression analysis, which indices have explanatory power for streamflow response. Based on the insights gained from the regression analysis, different spatial configurations were implemented in a hydrological model. Although I find the work flow elegant, starting from process-understanding and translating that to the spatial configuration of the model, I have some problems / concerns with the regression set-up.

**Major**

My main concerns are all related to the regression-part of the study.

 It is unclear how the indices, on which regression was applied, were selected. There is plenty of literature around on indices and signatures, which could guide indices selection, but I don't see any justification in the text for the choices made. Check for instance: Addor et al., A Ranking of Hydrological Signatures Based on Their Predictability in Space, WRR, 2018

Knoben et al., A Quantitative Hydrological Climate Classification Evaluated With Independent Streamflow Data, WRR, 2018

We thank the reviewer for the references she provided. The signatures used in this work were chosen in order to represent a wide variety of hydrograph characteristics. There are, for instance, two signatures designed to represent the long-term water balance (average streamflow and runoff coefficient), two signatures to capture the "responsiveness" of the hydrograph (baseflow index and flashiness index), and one (the half streamflow period) that is designed to understand seasonality effects, like the ones related to the snow dynamics. The indices used (here by indices we mean catchment and meteorological indices) were selected in order to have a large group of possible influence factors to start the analysis with. Catchment characteristics capture topography, soil properties, land use, and geology while meteorological characteristics take into account precipitation and potential evapotranspiration. **The list of signatures and indices has been extended in the new version of section 3.1.1. taking advantage of previous works.**

 Since the choice of the indices is not well justified, I am worried about their mutual correlation. Many indices can describe the same signal. Therefore, please provide the correlation among the indices themselves. This might lead to the insight that you need fewer or different ones. Mutual correlation, as pointed out by the reviewer, is a potential criticality of this study. Indeed there are multiple indices that describe the same signal (or very similar signals) and this possibility was considered in the design of this study. We decided to keep them all because we didn't want to restrict a priori the space of possible influence factors.

Section 3.2.2 in the new version of the paper deals with mutual correlations with the objective of selecting signatures and indices that are either not correlated or that represent different characteristics.

3. It was rightly mentioned that correlation does not mean causality. It was claimed (not only in the methods, but also in the conclusions) that this study accounts for that by only selecting the indices that have a causal relation, based on expert judgement. I do, however, not recognize the expert judgement in the selection of significant indices, and this actually directly relates to my point 2. Right now, the selection seems to be made based on the mutual correlation of the indices – so the mutual correlation was investigated! – but I don't see any process-reasoning (the expert-judgement) that can justify the selected indices, and that justifies the claim that there is really causality.

The analysis of the correlations was only the starting point of the process of selection of the indices. Starting form that, we then discarded the indices that are either redundant (e.g. average slope vs. fraction of steep areas) or accidental correlations (e.g. using elevation instead of precipitation, line 18 page 7). This step was essential to move from mere correlation to causality and it involved "expert judgment" in order to prune reasonably the list of indices. An example of the "expert judgment" process is illustrated in paragraph 3.2.2, page 10, lines 1 to 10, where we showed how the indices were selected for the mean streamflow signature.

The number of signatures and indices has been reduced using the "internal" correlation analysis and expert judgment (section 3.1.2 with results in 3.2.1 of the new version of the paper). This process has simplified the selection of influencing factors on streamflow signatures (section 3.1.3 with results in 3.1.2 of the new version of the paper). In both cases, we have highlighted the role of "expert judgment" in the process of indices selection.

4. I disagree with the conclusion of the authors that there is no need to look for nonlinearity in the correlation, based on the results in the table. The authors rightly state that only few correlations that are statistically significant based on Spearman are not significant with Pearson, but how do the authors explain the opposite effect? Quite some correlations are significant with Pearson but not with Spearman, is this a Type 2 error in the Pearson test? That could have consequences, for example, aridity was significantly correlated with BFI for Pearson, but not Spearman, and based on 'expert judgement' included in the regression.

The reviewer is right pointing out the possibility of type 2 error for some correlations, motivated by discording conclusions between Spearman and Pearson correlation. This may be due to the fact that the Pearson correlation was calculated neglecting the assumptions behind it (e.g. normality of the data) that may not be respected in this case.

In the new version of the paper all the analyses have been based on Spearman correlation.

5. 1 of the 3 points of the guidelines for modelling based on the regression was not based on the regression at all, namely the conclusion that the presence of snow is relevant. Please include a

snow-related indicator in the regression to support this conclusion (based on expert judgement we can expect this, of course).

The reviewer is right saying that it is not clear that "the conclusion that the presence of snow is relevant" was motivated by the regression analysis. This is due to the fact that we didn't explain earlier in the text that the "half streamflow period" signature and the plots in figure 5 were used to show the presence of seasonality in the streamflow dynamics. In particular, there are some subcatchments that reach their peaks of streamflow in different periods of the year. This seasonality is not due to different patterns in precipitation or PET (as shown in figure 5) and correlates well with the elevation (higher subcatchments reach half of their streamflow later in the year). These two points made us think that this seasonality is due to snow dynamics and that, therefore, the model should take them into account; higher catchments are subjected to snow that is then released in the streamflow (as snowmelt) later in the year if compared with rain-dominated subcatchments.

The explanatory power of the signatures has been highlighted (section 3.1.1 of the new version of the paper) and, in particular, we have made the hypothesis about the importance of the snow more explicit introducing the model MO.

- 6. It depends a bit on the definition of model building, but the title and the text might give the impression that the model structure itself was adapted with the insights in the regression, while it was basically the model implementation (accounting for HRU's or not) that was adapted. It is true: it depends on the definition of model building. For us it incorporates all the decisions taken in order to have an hydrological model for the Thur catchment. In particular:
  - How to spatially divide the inputs
  - How to divide the catchments in HRUs
  - Structure of the single HRUs

All these points where considered in the construction of the hydrological model and were informed by the regression analysis. The last point (structure of the bucket model) was also considered in the procedure of building the hydrological model but was not discussed in this paper because it was already done in previous studies and for sake of brevity. We have made clearer what we mean for model building, especially restructuring sections 4.1.1 and 4.1.5.

**Minor**

Section 3.2.1, the catchments are sort of grouped based on their stream flow response, but this is not used any further in the analysis. Consider to just briefly describe their response, or to use the grouping later to explain results (in that case, also display the groups in the figure).

Section 3.2.1 describes the signatures in the catchments without relating them with the indices. We agree with the reviewer that the subdivision done here is no more used in the paper and it was done only for convenience when describing the signatures.

**Section 3.2.1 has been completely revised in the new version of the paper.**

In the same section, it seems unnecessarily complicated to use combinations of signatures to determine how flashy catchments are; a flow duration curve can generally provide quite some insight on this already (slope of flow duration curve also frequently used signature)

We acknowledge the possibility of using other signatures to describe the behavior of a catchment but we used, among the others, baseflow index and the flashiness index because believe that they are more interpretable and they can be related to dynamics represented by the model; the BFI, for example, can be linked to the separation between quick and slow flow that is a process that is present also in the hydrological model.

**The selection of the signatures has been completely revised in the new version of the paper.**

Provide an overview of the indices and their abbreviations, or include their full name more often in text / tables / figures, because now it requires quite some work from the reader to fully understand all sentences and figures (and a lot of going back to the methods).

We understand that the usage of abbreviations may complicate the reading of the paper but, on the other side, their usage helps avoiding misunderstandings that may happen when calling the same index with different names. The full name of the indices is provided in table 2 and in section 3.1.1 and it will be reported in the caption of the figures when not reported in the figure itself.

We have tried in the new version of the paper to balance the usage of symbols with the usage of the full name.

A large number of figures is dedicated to showing the signature-values, which is not of direct relevance. I would be interested to see a figure that displays the HRU's.

We acknowledge that a figure representing the HRUs used in M2 and M3 is missing but it can be deducted from figure 1 (plot "d" for M2 and "c" for M3) since the HRUs were constructed aggregating some classes (for example, for M2, one HRU is composed by the orange part and the other by the rest of the catchment).

**Several figures have been changed in the new version of the paper.**

For the landscape characteristics, it is not mentioned in the methods-section (3.1.1) that you consider fractional area. Please clarify there, as I was wondering how you would apply regression on nominal values, until I found out in the results that you considered frac. area.

**We have clarified that (section 3.1.1 in the new version of the paper)**

*The sentence 'optimizing the parameter of the posterior distribution' (l.11, p13) can give the impression that you minimized e.g. variance (describing distribution), please consider reformulation.*

The actual meaning is "optimizing the parameters of the hydriological model and of the error model (refer to section 4.1.1 and 4.1.2) in order to find the ones that maximize the posterior distribution"

**We have changed the sentence (line 1, page 14 of the new version of the paper)**

Although overall written well and clear, some language editing seems required, for example "The average value oscillates of about ...", (but I'm not a native either).

**We have done our best to improve it.**

Overall, I appreciate the intent of the study and the modelling-part seems well designed (except for my question at point 6 which remained unclear), but I do believe the regression-part requires substantial revision, related to the selection of indices (more embedded in literature and account for snow) and to justify the use of the word 'causality'. Given that the work-flow is largely set-up, I think the authors should be able to incorporate this.

**Data analysis and model building for understanding catchment processes: the case study of the Thur catchment.**

Marco Dal Molin1,2,3, Mario Schirmer2,3, Massimiliano Zappa4, Fabrizio Fenicia1

 1Department Systems Analysis, Integrated Assessment and Modelling, Eawag, Swiss Federal Institute of Aquatic Science and Technology, 8600 Dübendorf, Switzerland
 2The Centre of Hydrogeology and Geothermics (CHYN), University of Neuchâtel, 2000 Neuchâtel, Switzerland
 3Department of Water Resources and Drinking Water, Eawag, Swiss Federal Institute of Aquatic Science and Technology, 8600 Dübendorf, Switzerland
 4Hydrological Forecast, Swiss Federal Research Institute WSL, 8903 Birmensdorf, Switzerland

10 Correspondence to: Marco Dal Molin (marco.dalmolin@eawag.ch)

**Abstract**

15

The development of semidistributed hydrological models that reflect the dominant processes controlling streamflow spatial variability is a challenging task. This study addresses illustrates this problem by investigating process through the case of the Thur catchment (Switzerland, 1702 km2), an alpine and pre–alpine catchment that, while having a moderate (1702 km2) extension, presents a with large spatial variability in terms of climate, landscape, and streamflow (measured at 10 subcatchments). The methodology for In order to appraise the dominant processes that control catchment response, and build a model that reflects them, the model development consists of follows a two-stages approach. In a first stage, we use

correlation-and regression analysis to identify the main influencing factors on the spatial variability of streamflow signatures. Results of this analysis show that precipitation (rainfall or snow) controlsaverages control signatures of seasonality and water

- 20 balance, snow processes control signatures of seasonality, while landscape characteristics (especially geology) control signatures of hydrograph shape (e.g. characterizing the importance of baseflow-index and flashiness index). In a second stage, we use the results of the previous analysis are used to develop a semidistributed hydrological model that is consistent with the data. Model set of model experiments aimed at determining an appropriate model representation of the Thur catchment. These experiments confirm that only a hydrological model that account for the heterogeneity of
- 25 precipitation-and, snow related processes, and landscape features such as geology-produce, produces hydrographs that have signatures similar to the observed ones. These models provide This model provides consistent results in space-time validation, which is promising for prediction predictions in ungauged conditions basins. The presented methodology for model building can be transferred to other case studies, since the data used in this work (meteorological variables, streamflow, morphology and geology maps) is available in manynumerous regions around the globe.

**1** Introduction**

Hydrographs are affected by meteorological forcing and landscape characteristics (e.g. topography, land use, etc.) and, therefore, they synthetize the hydrological response of a catchment. Because of Due to the spatial variability of landscape (e.g. topography, land use, etc.) and climate characteristics, hydrographs can differ substantially between catchments. Being

5 able to quantify and explain hydrograph spatial variability is important both to improve processes understanding and to make predictions useful for many human activities, such as flood protection, drinking water production, agriculture, energy production, and riverine ecosystems management (e.g., Hurford and Harou, 2014).

Understanding catchment differences and, more specifically, how to transfer hydrological knowledge, methods, and theories from one place to another between places, is a common objective of many research areas in hydrology, including

- comparative hydrology (e.g., Falkenmark and Chapman, 1989), model regionalization (e.g., Parajka et al., 2005), catchment classification (e.g., Wagener et al., 2007), and prediction in ungauged basins (e.g., Hrachowitz et al., 2013). In the case of streamflow, the attempt to explain its spatial variability is typically accomplished either using statistical approaches, which tryare designed to regionalize someselected characteristics of the hydrograph (streamflow signatures), or usingthrough hydrological models that incorporateaccount for relevant spatial information. In particular, statistical approaches such as
- regression analysis (e.g., Berger and Entekhabi, 2001; Bloomfield et al., 2009) and correlation analysis (e.g., Trancoso et al., 2017), or machine learning techniques like clustering (e.g., Sawicz et al., 2011; Toth, 2013; Kuentz et al., 2017) are used to extrapolate the signatures where unknown and to group together catchments that present similar characteristics and to extrapolate the signatures where unknown. Such approaches have been useful to quantify the hydrological variability and identify its principal diversdrivers. However, they are often not designed to discover causality links and can be affected by multicollinearity, that arises when multiple factors are correlated internally and with the target variable (Kroll and Song,
- ---

2013).

By incorporating spatial information about meteorological forcing and landscape characteristics, distributed semidistributed hydrological models have the ability to reproduce mimic the mechanisms that influence hydrograph spatial variability. However, identifying the relevant mechanisms is challenging. One possibility is to be as inclusive as possible in accounting

- 25 for all the catchment properties that are, in principle, important in controlling catchment response. However, this approach leads to models that tend to be data demanding and contain severalmany parameters. For example, Gurtz et al. (1999) considered several landscape characteristics (elevation, land use, etc.) in their application of a semidistributed model to the Thur catchment (Switzerland), which resulted into a model with hundreds of hydrological response units (HRUs) that were defined a-priori based on the complexity of the catchment. The other option is to try to identify the most relevant processes
- and neglect others, by tuning the distributed hydrological model to the available data.in order to control model complexity.
   For example, Fenicia et al. (2016) compared various model hypotheses to determine an appropriate discretization of the catchment in HRUs and appropriate structures for different HRUs. However, in their work, the space of plausible hypotheses could be constrained by a good experimental understanding of the area, which is not always available. Antonetti et al. (2016)

used a map of dominant runoff processes following Scherrer and Naef (2003) for defining HRUs. However, these approaches require a good experimental understanding of the area, which is not always available.

Convincing model calibration-validation strategies are essential to provide confidence that the model ability to fit observations is a reflection of model realism and not a consequence of calibrating an overparameterized model (e.g.,

- 5 Andréassian et al., 2009). A common approach for calibration of semidistributed models is the so called 'sequential' approach, where subcatchments are calibrated sequentially from upstream to downstream (e.g., Verbunt et al., 2006; Feyen et al., 2008; Lerat et al., 2012; De Lavenne et al., 2016). Although this approach may provide good fits and therefore it has its practical utility where data is available, it does not provide understanding into the causes of streamflow spatial variability and results into models that are not spatially transferable. Moreover, such models are prone to contain many parameters, as
- 10 each subcatchment would be represented by its own set of parameters set. Alternative calibration-validation approaches that enable model validation not only in time but also in space are conceptually preferable, particularly when the modeling is used for process understanding or prediction in ungauged locations (e.g., Wagener et al., 2004; Fenicia et al., 2016).
  - This study combines the strengths of catchment regionalization approaches and distributed semidistributed hydrological models by first using regionalization studies regression analysis to understand the main causes of variability of streamflow
- 15 signatures, and then using this analysis to inform the structure of a distributed hydrological model. The model objective is to explain the observed spatial diversity of streamflow characteristics with the minimum possible complexity, while maintaining a process based interpretation. In particular, the objectives of the study are to: (1) explore the spatial variability present in the Swiss Thur catchment regarding landscape characteristics, meteorological forcing and streamflow signatures; (2) find which characteristics identify the main drivers that explain the variability of the hydrological response; (3) based on
- 20 this analysis, build a semidistributed hydrological set of model that considers only features that actually contribute experiments aimed to the spatial variability test the relative importance of dominant processes and their effect on the hydrograph; (4) validate appraise model assumptions against competing alternatives using a stringent validation strategy. The paper is organized as follows: Section 2 presents the study area and gives information about data collection and availability; Section 3 and Sect. 4 are both divided in methods and results and present, respectively, the correlation and
- 25
- regression analysis and the modeling part of this paper; Section 5 puts the results of this work in prospective comparing them with other studies; Section 6, finally, summarizes the main conclusions.

**2 Study area**

This study is carried out in the Thur catchment (Fig. 1), located in north-east of Switzerland, south-west of the Lake Constance. With a total length of 127 km and a catchment area of 1702 km2, the Thur is the longest Swiss river without any

natural or artificial reservoir along its course. Due to this characteristic, it The Thur river is a very dynamic-river, where the 30 with streamflow values that can change of by two orders of magnitude inwithin 
[revised manuscript text omitted]

- 20

15

10

$$\zeta_{\mu\nu} = \frac{\sum_{t=2}^{N_{\tau}} |q_{t} - q_{t-1}|}{\sum_{t=2}^{N_{\tau}} q_{t}};$$
(2)

and used to describe the "responsiveness" of a catchment.

• streamflow elasticity ( $\zeta_{EL}$ ) defined as

$$\zeta_{\rm EL} = \mathrm{med}\left(\left(\frac{\Delta \bar{q}}{\bar{q}}\right) / \left(\frac{\Delta \bar{p}}{\bar{p}}\right)\right)$$
(1)

where  $\Delta \overline{q}$  and  $\Delta \overline{p}$  represent the streamflow and precipitation jumps between two consecutive years and med is the median function;

- slope of the flow duration curve ( $\zeta_{FDC}$ ) defined as the slope between the log-transformed 33rd and 66th streamflow percentiles:
- baseflow index  $\zeta_{BFI} = \frac{\overline{q^{(b)}}}{\overline{q}}$ , where  $q^{(b)}$  represents the baseflow and was calculated using a low-pass filter as illustrated in Ladson et al. (2013) with the equation

$$q_{t}^{(f)} = \min \left( 0, \vartheta_{b} q_{t-1}^{(f)} + \frac{1 + \vartheta_{b}}{2} (q_{t} - q_{t-1}) \right)$$
(2)
$$q_{t}^{(b)} = q_{t} - q_{t}^{(f)}$$
(3)

- with  $q_t^{(f)}$  representing the quick flow. The settings of the filter were taken according to the findings of Ladson et al. (2013) and, in particular, three filter passes were applied (forward, backward, and forward), the parameter  $\vartheta_b$  was chosen to be equal to 0.925, and a reflection of 30 time steps at the beginning and at the end of the time series was used;
- mean half streamflow period\_date (ζHSPζHFD) (Court, 1962), defined as the number of days needed in order to have a cumulated streamflow that reaches the 50 % of the total annual streamflow; the value obtained is then normalized by the total number of the days in the year. This index is designed to capture the seasonality of streamflow, since it helps differentiating between catchments with high streamflow during the winter and catchments with high streamflow during the spring.

Climatology was represented through the following indices:

- $5^{\text{th}}$  and  $95^{\text{th}}$  percentiles of the streamflow ( $\zeta_{Q5}$  and  $\zeta_{Q95}$  respectively);
- frequency ( $\zeta_{HQF}$ ) and mean duration ( $\zeta_{HQD}$ ) of high-flow events: they are defined as the days when the streamflow is bigger than nine times the median daily streamflow;
- frequency ( $\zeta_{LQF}$ ) and mean duration ( $\zeta_{LQD}$ ) of low-flow events: they are defined as the days when the streamflow is smaller than 0.2 times the mean daily streamflow;

The frequency of days with zero streamflow, present in Addor et al. (2017), was not considered in this study because there are no ephemeral subcatchments in the study area.

- 20 This group of streamflow signatures is capable of capturing various characteristics of the hydrograph:  $\zeta_{Q}$  measures the overall water flows,  $\zeta_{RR}$  represents the proportion of precipitation that becomes streamflow,  $\zeta_{EL}$  measures the sensitivity of the streamflow to precipitation variations, with a value greater than 1 indicating an elastic subcatchment (i.e. sensitive to change of precipitation) (Sawicz et al., 2011),  $\zeta_{FDC}$  measures the variability of the hydrograph with a steeper flow duration curve indicating a more variable streamflow,  $\zeta_{BFI}$  measures the magnitude of the baseflow component of the hydrograph, and
- 25 can be considered as a proxy for the relative amount of groundwater flow in the hydrograph,  $\zeta_{HFD}$  measures the streamflow seasonality,  $\zeta_{Q5}$ ,  $\zeta_{LQF}$ , and  $\zeta_{LQD}$  measure low-flow dynamics,  $\zeta_{Q95}$ ,  $\zeta_{HQF}$ , and  $\zeta_{HQD}$  measure high-flow dynamics. Climatology was represented through the following indices (see Addor et al. (2017), Table 2):
  - average precipitation  $\psi_{\overline{\mu}}\psi_{P} = \overline{p}$ ;
  - average PET  $\psi_{PET} = \overline{e_{pot}};$

30 • average PET  $\psi_{\text{PET}} = \overline{e_{\text{pot}}}$ , where  $e_{\text{pot}}$  is the potential evapotranspiration time series;

• aridity index  $\psi_{AI} = \frac{\overline{e_{pot}}}{\overline{p}} \psi_{AI} = \frac{\overline{e_{pot}}}{\overline{p}}$

10

5

- These indices were designed to capture different features fraction of snow ( $\psi_{FS}$ ), defined as the time series: yearly streamflow, volumetric fraction of precipitation falling as snow (i.e. on days colder than 0 °C);
- frequency  $(\psi_{HPF})$  and PET can be called "magnitude" indices since mean duration  $(\psi_{HPD})$  of high precipitation events: they are a measure of defined as days when the water flows; precipitation is bigger than five times the remaining indices give information about mean daily precipitation;
- season ( $\psi_{HPS}$ ) with most high precipitation events (defined as above);
- frequency ( $\psi_{LPF}$ ) and mean duration ( $\psi_{LPD}$ ) of dry days: they defined as days when the "shape" of precipitation is lower than 1 mm day-1;
- season ( $\psi_{LPS}$ ) with most dry days (defined as above).
- 10 The seasonality of precipitation used in Addor et al. (2017) was not considered in this study as it relied on fitting a sinusoidal function to the time seriesprecipitation values, which in our case did not produce reliable results. Nevertheless, these climatological indices are able to comprehensively represent the climatic conditions of the suubcatchment, with  $\psi_{\rm P}$  representing average water input,  $\psi_{\rm PET}$  representing average evaporative demand,  $\psi_{\rm AI}$  measuring the dryness of the climate,  $\psi_{\rm FS}$  measuring the relative importance of snow,  $\psi_{\rm HPF}$ ,  $\psi_{\rm HPD}$ , and  $\psi_{\rm HPS}$  measuring the importance of intense precipitation
- 15 events, and  $\psi_{\text{LPF}}$ ,  $\psi_{\text{LPD}}$ , and  $\psi_{\text{LPS}}$  measuring the importance of dry days.

The landscape characteristics, illustrated in Sect. 2, need to be synthetized in a numeric value were divided in four categories: topography, land use, soil, and geology. In order to quantify the characteristics of each category, a set of indices  $(\xi)$  before being used in the correlation and regression analysis. The maps were processed using GIS techniques and, for each subcatchment, numerical features were extracted. Allwas defined. It is important to notice that all the areas calculated in this analysis were normalized by their the respective subcatchment area  $(\xi_{\pi}\xi_{\Lambda})$  in order to get comparable values between

20

25

30

5

subcatchments of different size.

In particular, Topography was represented with the following indices, calculated based on the digital elevation model (DEM) was used to calculate the following topographic information:):

- average elevation  $(\xi_{TETE})$ ;
- average slope  $(\xi_{TSmTSm})$ ;
  - fraction of the subcatchment with steep areas  $(\xi_{TSS}\xi_{TSS})$ , with slope larger than 10°;
  - aspect, i.e. areasfraction of the subcatchment facing north  $(\xi_{TAR}\xi_{TAn})$ , south  $(\xi_{TAS})\xi_{TAS}$ , or east and west  $(\xi_{TACWTACW})$ .

[revised manuscript text omitted]
 parametersthat: all these choices were, in this study, were motivated by the results of the regressioncorrelation analysis, i.e. only catchment characteristics that were found capable of explaining the hydrological response were used.

**3.2 Results and interpretation**

5

10

This section illustrates the results of the correlation analysis aimed to identify influencing factors that control the spatial variability of streamflow signatures; Section 3.2.1 showspresents the spatial results of the selection of meaningful statistics; Section 3.2.2 identifies climate and landscape indices controlling streamflow signatures and presents consequences for model development.

**3.2.1 Selection of meaningful streamflow signatures, climatic indices, and catchment indices**

The streamflow signatures defined in Sect. 3.1.1 were calculated for each subcatchment and the values are shown in Table 2 together with the coefficient of variation. All the signatures have a coefficient of variability of the indices, the correlation and regression analysis and bigger than the threshold value of 5%, with the most variable signature being  $\zeta_{LQF}$  (71%) and the least variable  $\zeta_{HOD}$  (6%). Therefore, none of these signatures was discarded.

Figure 2 shows the correlations between the streamflow signatures: the lower triangle contains the Spearman's rank correlation and the upper triangle the p-value associated with the correlations. Based on correlations and on its interpretation is presented in Sect. 3.2.2., a subset of  $\zeta$  can be defined as follows:

**3.2.1 Spatial and temporal variability of catchment indices**

In Fig. 2, each boxplot shows the variability (between years) of the observed streamflow signatures. This analysis suggests that, based on the signatures ζQ, ζRC, ζBFI, and ζFI, the subcatchments can be qualitatively divided in three separate groups:
 subcatchments in the north west hilly part (Frauenfeld and Wängi) characterized by on average lower values of ζQ (less than about 700 mm yr-1), ζRC (less than 0.60), and ζFI (about 0.30), and higher values of ζRFI (
[revised manuscript text omitted]

(56)

$$p(\boldsymbol{q}_{obs}|\boldsymbol{\theta}_{h},\boldsymbol{\theta}_{E},\boldsymbol{x}) = \prod z'(\boldsymbol{q}_{obs}|\boldsymbol{\theta}_{E})f_{N}(\boldsymbol{E}|0;\sigma^{2})$$
(6)
where  $p(\boldsymbol{q}_{obs}|\boldsymbol{\theta}_{h},\boldsymbol{\theta}_{E},\boldsymbol{x}) = \prod_{t=1}^{T} z'(\boldsymbol{q}_{obs,t}|\boldsymbol{\theta}_{E})f_{N}(E_{t}|0;\sigma^{2})$
(6)

where T represents the length of the time series,  $f_N$  is the Gaussian probability density function (PDF) and  $z'(q_{obs}|\theta_E)$  is the derivative of  $z(q_{obs}, \theta_E)$  with respect to q evaluated at the observed data  $q_{obs}$ . Specifying Eq. (67) for the case where  $z(q_{obs}; \theta_E)$  is defined by Eq. (45), the expression of the likelihood function becomes:

| 25 | $p(\boldsymbol{q}_{obs} \boldsymbol{\theta}_{h},\boldsymbol{\theta}_{E},\boldsymbol{x}) = \prod \boldsymbol{q}_{obs}^{(\lambda-1)} f_{\mathcal{H}}(\boldsymbol{E} 0;\sigma^{2})$         | <del>(7)</del> |
|----|------------------------------------------------------------------------------------------------------------------------------------------------------------------------------------------|----------------|
|    | $p(\boldsymbol{q}_{\text{obs}} \boldsymbol{\theta}_{\text{h}},\boldsymbol{\theta}_{\text{E}},\boldsymbol{x}) = \prod_{t=1}^{T} q_{\text{obs},t}^{(\lambda-1)} f_{N}(E_{t} 0;\sigma^{2})$ | (8)            |
|    | Equation (8) represents the likelihood function that is then used, together with an uniform prior distribution, to                                                                       | calibrate the  |
|    | parameters of the model as described in Sect. 4.1.3.                                                                                                                                     |                |

**4.1.3 Calibration**

30 Parameter calibration was performed by optimizingwith the parametersobjective of the maximizing their posterior distribution density. According to Bayes equation, the posterior distribution of model parameters is expressed as the product
between the prior distribution and the likelihood function; since an uniform prior was used for the parameters, this is equivalent to maximizing the likelihood function in the defined parameter space; the optimization procedure was <del>done</del> <del>usingperformed with</del> a multi-start quasi-Newton method (Kavetski et al., 2007) with 20 independent searchers. We empirically established that with models of our complexity (about 10 parameters), 20 independent searches provide good

**5 confidence that a global optimum is found.**

The evaluation of the model ability to reproduce streamflow was carried out in space-time validation-(see also Fenicia et al., 2016). For this purpose, the time domain was divided in two periods of 12 years each (from 01 September 1981 to 01 September 1993, and from 01 September 1993 to 01 September 2005) and the subcatchments were split ininto two groups (A and B), according to a spatial alternation (subcatchment in group A flows into a subcatchment in group B that flows into one

- in group A and so on); the subcatchments belonging to group A are Andelfingen, Herisau, Jonschwil, St. Gallen, Wängi and the ones in group B are Appenzell, Frauenfeld, Halden, Mogelsberg, Mosnang. This method implies a division of the space–time domain in four partsquadrants, such that the model can be calibrated in one quadrant and validated in the other three. For space–time validation, the model was calibrated using each group of subcatchment and each period, and validated using the other group of subcatchment and period. That is, the model calibrated using group A and period 1 was validated using
- group B and period 2, and so on for the other 3 combinations of subcatchments and groups. The model output in the 4 space– time validation periods was then combined, to calculate model performance using various indicators (see Sect. 4.1.4).
   Results are presented for space time validation, which represents the most challenging test of model performance.

**4.1.4 Performance assessment**

25

30

Model performance was assessed using the following metrics:

- Time series metrics, which evaluate the ability of reproducing streamflow time series. The metrics used for this assessment are the following:
  - Normalized log-likelihood (LL), that is, the logarithm of Eq. (78) normalized by the number of time steps present in the time series. This metrics corresponds to the objective function used for model optimization. It can be observed that, since λ is fixed at 0.5 in the Box-Cox transformation, model calibration is equivalent to maximising the Nash-Sutcliffe efficiency (NS) calculated with the square root of the streamflow. LL is not bounded but a higher value means a better match between two time series since, in this case, the absolute value of the residual is smaller and, thus, their PDF higher.
  - Nash–Sutcliffe efficiency

$$NS(q_{obs}, q_{sim}) = 1 - \frac{\sum_{t=1}^{T} (q_{sim}^{t} - q_{obs}^{t})^{2}}{\sum_{t=1}^{T} (q_{obs}^{t} - \overline{q_{obs}})^{2}} - \frac{(8(q_{obs}, q_{sim}) = 1 - \frac{\sum_{t=1}^{T} (q_{sim,t}^{t} - q_{obs,t})^{2}}{\sum_{t=1}^{T} (q_{obs,t}^{t} - \overline{q_{obs}})^{2}} - \frac{(9)}{\sum_{t=1}^{T} (q_{obs,t}^{t} - \overline{q_{obs,t}})^{2}} - \frac{(9)}{\sum_{t=1}^{T} (q_{obs,t}^{t} - \overline{q_{obs,t}})^{$$

Which is often used in hydrological applications, and it provides a sense of general quality of the simulations. NS is bounded between  $-\infty$  and 1, with 1 meaning a perfect match.

2. Signature metrics, which determine the ability of reproducing the selected streamflow signatures ( $\zeta$ ) presented which, as illustrated in the regression analysis part (Sect.Section 3.2.1.1), that is, of , are average daily streamflow ( $\zeta_{Q}$ ), runoff coefficient ( $\zeta_{RC}\zeta_Q$ ), baseflow index ( $\zeta_{BFT}$ ), flashiness index ( $\zeta_{FT}$ ), and  $\zeta_{BFI}$ ) mean half streamflow period date ( $\zeta_{HFD}$ ), 5th percentile of the streamflow ( $\zeta_{Q5}$ ), and duration of high-flow events ( $\zeta_{HSFP}\zeta_{HQD}$ ). The accordance between simulated and observed signatures was assessed both visually and using the PearsonSpearman's rank correlation.

This set of metrics, together with the fact that they are calculated in space time validation (Sect. 4.1.3), provides a comprehensive assessment of model performance.

The use of multiple metrics for assessing model performance enables a comprehensive assessment of various characteristics of the simulations. Time series metrics were designed to appraise the general quality of the model fit. Signatures, instead, were designed to highlight selected characteristics of the data at the expense of others.

**4.1.5 Model experiments for testing the results of the correlation analysis**

5

10

20

25

- 15 Using the model structure described in Sect. 4.1.1<del>.1</del>, several, four model variants areconfigurations were compared. The main motivations for such comparisons are as follows:
  - verify that models that account for the influencing factors identified through the regression analysis indeed lead to an improved representation of streamflow spatial variability;
  - provide a mechanistic interpretation of how influencing factors affect streamflow, which cannot be achieved by regression analysis;
  - get some insights on the relationship between model complexity and performance.

The model variants and their specific rationale are described below:

- In order to verify the effect of spatial distribution of landscape properties, we constructed a reference model with a single HRU, called M1, (i.e. no spatial distribution of landscape properties); in this case only the input variability (the catchment is still divided in subcatchments) is considered.
- In order to verify that geology controls streamflow variability (see Sect. 3.3, point 3), particularly by influencing baseflow conditions, a two HRUs model, called M2, was implemented, dividingvarying the three geology classes in unconsolidated, for number and the first HRU, and consolidated and alluvial, for the second HRU (see Sect. 2 and Fig. 1d).
- 30 In order to verify that eventual improvements in performance brought by M2 compared to M1 are not just due to increase in complexity, we implemented a two-definition of the HRUs, and changing the structure of the HRUs model, called M3, using the land use to discretize the domain. The land use classes were arbitrarily defined so that the first HRU contains forest and

erops and the second occupies the rest of the catchment. (Fig 6). The objective of the experiments was to test the hypotheses 1-4 in Sect. 3.3.

The first hypothesis (precipitation controls the water balance) is tested with the model M0, with uniform parameters on the catchment (i.e. a single HRU) and distributed precipitation input. This model does not consider snow processes. We expect

5 that this model will be able reproduce differences in streamflow averages between subcatchments. The second hypothesis (snow controls seasonality) is tested with the model M1. Relatively to M0, M1 accounts for snow processes, represented by simple degree day snow module (see Kavetski and Kuczera, 2007), with inputs (temperature) distributed per subcatchment.

The third hypothesis (geology controls baseflow) is tested with the model M2. Relatively to M1, M2 considers two HRUs,

defined based on geology type. One HRU contains the areas with consolidated geology while the other HRU contains the 10 rest of the catchment (unconsolidated and alluvial geology together).

The fourth hypothesis (other catchment characteristics should not be considered if the idea is to keep the model as simple as possible), is exemplified by the model M3. M3 is analogous to M2 except that HRUs are based on catchment characteristics that did not show correlation with the streamflow signatures. Among those characteristics, we have selected land use, and considered an HRU based forest and crops and the second one that occupies the rest of the catchment.

- The total number of the calibrated parameters depends on the number of HRUs and on the structure used to represent them: it was nine in the first experiment (Table A1)8 for M0, 9 in M1, and 13 in the other two M2 and M3, where five5 parameters were linked between different HRUs; (Table A1); those parameters are:  $\frac{C_e}{C_e} C_e$  that governs the evapotranspiration,  $\frac{t_{e}}{t_{rise}} t_{rise}^{OL}$ and  $\frac{t_{FLS}}{r_{FLS}} t_{rise}^{IL}$  that control the routing in the river network,  $\frac{k_{WR}}{k_{WR}} k_{WR}$  that regulates the outflow of the snow reservoir, and  $\frac{S_{max}^{UR}}{S_{max}}S_{max}^{UR}$  that determines the behaviour of the unsaturated reservoir.
- 20

15

**4.2 Results and interpretation**

This section presents the results of the modelling experiments. Section 4.2.1 illustrates model results in terms of hydrograph metrics. Section 4.2.2 presents model results in terms of signatures. An interpretation of the results, including a comparison with the conclusions of the regression correlation analysis, is given in Sect. 4.2.3.

**25 4.2.1 Model performance in terms of hydrograph metrics**

Figure 7a shows the values of the likelihood function (corresponding to the calibration objective function) for the threefour models in calibration and validation. It can be observed that M0 is, by far, the worst model, having a low value of likelihood. Moving to the other three models, it can be seen that, during calibration, M1, which has the lowest number of calibration parameters, has the lowest likelihood value of the three, indicating lowest performance, whereas M2 and M3 have similar higher likelihood values. This behaviour continuespersists in time validation, with M2 and M3 that outperform M1. In space

and space-time validation, however, M3 has the lowest likelihood value of the three, whereas M1 and M2 limit their decrease in performance, havingranking, respectively, the second and the first value of optimal likelihood value.

The likelihood function represents an aggregate metric of model performance; in order to get a sense of appreciation of model fit on individual subcatchments, Fig. 7b reports the values of Nash Sutcliffe efficiency in space time validation for
each of the subcatchments. On average, M2 has the best performance of all models (NS = 0.79), followed by M1 (NS = 0.78) and), M3 (NS = 0.77), and M0 (NS = 0.68). M3 hasand M0 have the highest variability of performance, with NS values between 0.58 and 0.86 and between 0.59 and 0.81. M1 and M2 have similar spread of NS values, ranging from 0.69 to 0.85 for M1 and from 0.73 to 0.87 for M2. Therefore, M1 and M2 have a more stable performance across subcatchments than

M3. M3 obtains a significantly worse performance than the other 2 models on Mosnang, where it reaches a NS value of 0.58 (M1 and M2 have values of 0.69 and 0.73 respectively).

It can also be observed that M2 is generally better than M1, with NS values that are higher or approximately equal except for the subcatchments Andelfingen and Halden, where the NS is slightly worse (however still higher than 0.80). M3 is clearly better than M1 on Andelfingen, Frauenfeld and Wängi, and clearly worse on Herisau and Mosnang. In particular, in Mosnang (the smallest basin), M3 reaches the worst performance of all models on all subcatchments.

15 Regarding M0, it is interesting to observe that it has the worst performance (among all the subcatchments) in Appenzell, which is the subcatchment that is most affected by snow ( $\psi_{FS} = 0.21$ ), while it reaches a performance similar to M1 in Frauenfeld and Wängi, which are two subcatchments with almost no snow.

**4.2.2 Model performance in terms of signature metrics**

Figure 8 compares the observedability of M0 and simulated signatures for M1. Figure 9 to capture the signatures

[revised manuscript text omitted]

---

## Author Response (AR2)

**Author's response**

**Changes in the paper**

The paper has been subject to major revision in order to address the comments of the reviewers. The most significant changes concern the structure of the paper, where we have separated the methods from the results, and the introduction, the discussion, and the conclusion sections where we tried to address the concerns of the reviewers. A summary of the changes is presented below, followed by the individual responses to the editor and to the reviewers.

**Changes in text**

| SECTION | DESCRIPTION                                                                             |
|---------|-----------------------------------------------------------------------------------------|
| Title   | The title has been changed following the indications of Anonymous Referee #3.           |
| 1       | The introduction has been changed adding more information about the objective of the    |
|         | study.                                                                                  |
| 3       | Section 3 now contains all the methodology, both of the correlation analysis and of the |
|         | modeling study. Minor changes have been made to the single paragraphs.                  |
| 4       | Section 4 now contains all the results, both of the correlation analysis and of the     |
|         | modeling study. Minor changes have been made to the single paragraph.                   |
| 5       | The discussion has been enlarged, clarifying the choices made and pointing out their    |
|         | limitations.                                                                            |
| 6       | Major changes to address the concerns of the reviewers.                                 |

**Changes in Figures**

The following figures were modified

| FIGURE | DESCRIPTION                                              |
|--------|----------------------------------------------------------|
| 1      | Different colors for the land use map.                   |
| 6      | Changed to address the concerns of Anonymous Referee #3. |
| 7      | Changed to address the concerns of Anonymous Referee #3. |

**Changes in Tables**

Table 2 was added listing the signatures and the indices used in the study.

**Reply to the editor Dr. Conrad Jackisch**

**Dear Marco Dal Molin and co-authors,**

Thank you again for your contribution to our special issue and the work you invested into your manuscript's revision. After reading your manuscript again and considering the two independent reviewer reports, I follow their suggestion to open a second round of major revisions. Please be aware that both reviewers scored scientific quality and significance as "good" and presentation quality as "fair". Given your interesting material in your manuscript, I am sure this can benefit from considering the very thoughtful comments.

The two reviewers point out two lines of revisions. While Referee #4 has suggestions for fundamental clarifications of the study's aims and scopes. I read these not in a sense questioning your overall study but as valuable references a revision should orientate on. Hence I would expect that answering his questions on the fundamental level as key to structure the revisions on. Referee #3 addresses fundamental methodological aspects, which might on second look not be too far from the reflections of Referee #4. Considering the scope of our special issue as third pole, this lines up quite nicely in my view ("Linking landscape organisation and hydrological functioning: from hypotheses and observations to concepts, models and understanding").

If you see any trouble in addressing the comments during your revisions, please do not hesitate to contacting me for further clarification.

Thank you very much for your efforts and work you put into this manuscript.

All the best.

Conrad

We thank the Editor Dr. Conrad Jackisch for his thoughtful suggestions. We believe we have done another major review of the manuscript in order to address the comments of both reviewers. We noted that unfortunately the paper received a new set of reviewers, who came up with several new points, sometimes in contrast with the points raised in the first round of reviews. We did our best to address the comments of the current set of reviewers, without penalizing the changes already made to comply with the suggestions of the previous reviewers. Nonetheless, the paper underwent major changes, as can be noted in the differences' file, including a major restructuring, as suggested by Anonymous Referee #3.

We believe that our paper contributes to understanding the link between catchment properties and hydrological functioning, and therefore is well in line with the topic of the special issue. We are

confident that the changes made, including the change in title and the new argumentations in the introduction, make this link even more visible.

Kind regards,

Marco Dal Molin (on behalf of the coauthors).

**Reply to review by Anonymous Referee #3**

We thank the reviewer for his/her careful read to the manuscript and insightful suggestions.

As it can be noticed in the differences' file, the paper has undergone a major restructuring, in the spirit of capturing most of the suggestions of the reviewers. However, as the reviewers in this round of reviews are different from the reviewers from the previous round of reviews, we had to be careful that the suggestions of the current reviewers are not in contrast with the modifications already made to comply to the suggestions of the previous reviewers. Cases where a conflict occurs are mentioned in our replies.

Below, we answer in detail the various comments, and illustrate how we have addressed them in the revised version. The original comments of the reviewer are reported in *black and italics*, our replies in blue.

All the references to specific pages and lines of the paper are based on the version without track changes. Since the numbering of the sections has changed in the reviewed paper, we will call "first revision of the paper" the version that you have reviewed and "new revision" the version that we are submitting together with this reply.

**Main comments**

Dal Molin and colleagues submitted their revised manuscript entitled "Data analysis and model building for understanding catchment processes: the case study of the Thur catchment" to Hydrology and Earth System Sciences (HESS) Special Issue: Linking landscape organisation and hydrological functioning: from hypotheses and observations to concepts, models and understanding. The manuscript was substantially improved after major modifications following first iteration with reviewers and editors and I enjoyed reading it. However, I failed to identify a major scientific contribution in terms of processes understanding supported by hydrological interpretations, which makes me feel that authors are targeting the proposition of the regional modelling framework rather than the potential hydrological insights of the modelling exercise. In that sense, there are some issues that need to be addressed in order to achieve a replicable regional modelling framework, which are discussed in details below.

1. Section 3 still worries me a bit. First, the climate indices presented in Addor et al (2017) were selected to be representative in large-scale studies (e.g., CAMELS) where a large climatic gradient is the main control of catchment's streamflow spatial variability. This is not necessarily a valid assumption for regional studies where climate variability (across space) is much smaller and variables are highly correlated (as per Fig 3).

The reviewer is right saying that in a limited region, such as the one presented in this study, the climatic conditions may not vary as strongly as in studies targeting large climatic gradients (e.g. CAMELS); however assuming no climatic variability in the Thur catchment would be unjustified: for example, the mean annual precipitation varies significantly between the subcatchments (e.g. 5.15 mm/d in Appenzell vs. 3.36 mm/d in Frauenfeld); moreover the variability of the amount of precipitation falling as snow is large (e.g. 21% in Appenzell vs. 4% in Frauenfeld) which induces significant differences in streamflow seasonality, as we have shown in figure 5 presented in the

first submission of the paper, then removed to comply to the reviewers suggestions, and reported below.

The figure shows that, while precipitation and potential evapotranspiration follow the same annual pattern in all the catchments, the streamflow follows two different patterns, dividing the catchments in two different groups:

- Snow affected catchments (e.g. Appenzell) with high streamflow during the late spring and summer;
- Catchments with less snow (e.g. Frauenfeld) with highest streamflow between October and March.

Since this behavior was also captured by the "mean half streamflow date" signature, the figure was omitted by the second submission of the paper, as suggested by Anonymous Referee #1.

**We will address this point below**

Third, the final "expert judgment" adds subjectiveness and undermines replicability of modelling framework. The other way around would be more intuitive – i.e., run the "expert judgment" prior in order to select relevant metrics and establish a process-based conceptual (see third point below) model taking accounting relevant specificities of study area (see point 4 below) and then the metrics assessment part.

In principle there are many climate and landscape characteristics that influence catchment response. The question is which one are the most relevant for the application, in particular at the spatial scale of the study and for the variables that one wants to predict. The reason for running the expert judgment after the correlation analysis is to be able to derive some of these key model decisions from it and not to decide them a-priori which would be difficult, if not impossible.

2. Highly correlative nature of section 3: in the metrics assessment part (section 3), essentially, criteria for metric selection should go beyond correlation and represent similarity, dissimilarity, complementarity and/or importance of metrics and uncertainty. Bray-Curtis ordination or PCA could be helpful to understand data structure and complementarity and random forests could be used to calculate importance of metrics (see Kennard at al (2010) River Res Applic and Trancoso et al (2016) JoH for analytical examples). That would strength the analytical component of section 3.

The possibility to use more advanced methods for metrics selection has been considered in the process of our study; the reason why we eventually selected a simple method is that the sample size of this study is relatively small. We are in fact limited to only 10 catchments. Studies that use complex regression techniques like random forests use a much larger sample of catchments; for example, the work proposed by Trancoso et al. (2016) deals with 355 catchments. Using such techniques risks to result in models that overfit the data, especially considering the fact that we would need to split the catchments in a calibration and a validation group.

In the previous revision of the paper, we took the suggestion of the first set of reviewers and used Spearman correlation instead of Pearson, in order to account also for nonlinear correlation.

We have been more specific about the reasons behind this choice in the "limitations" part of the discussion.

In the attempt to comply with the suggestion of the reviewer, we tested LASSO regression using catchment characteristics and climatic indices (indicated as  $x_i$ ) to predict every single hydrological signature (indicated as y).

$$y = \sum \lambda_i x_i + \alpha \sum |\lambda_i|$$

The idea behind this method is that it should perform (thanks to the normalization term  $\alpha \sum |\lambda_i|$ ) feature selection, setting to zero all the  $\lambda_i$  that are associated with indices that are not important in calculating the output y.

While being, in theory, a technique suitable also for a small unbalanced (unbalanced in the sense that there are more indices than catchments) data set, in the case of the Thur catchment the method performed poorly mainly because of the high correlation among the indices. In particular, the method prefers the most correlated variables, but we may be interested in a slightly less correlated variable if this reflects a more plausible cause effect relationship.

One possible solution would be applying a pre-selection of the indices based on expert judgment (as done in section 3.2.1 of the first revision of the paper) before doing LASSO regression but this would fall back to the fallacies in the methodology criticized by the reviewer. If a pre-selection is done, this method would produce results comparable with the one given by the correlation analysis (done in section 3.2.2 of the first revision of the paper), with the disadvantage that the information that we get from the LASSO regression would be only a list of selected indices, while the correlations express also the strength of the relationship between indices and signatures.

For these reasons, we have preferred to keep using the (non-linear) correlation analysis, aided by expert judgment, to select the meaningful catchment characteristics and climatic indices that influence streamflow signatures: it is true that this approach may be subjective but it guaranties meaningful insights for building the hydrological model that are not affected by spurious correlations.

Uncertainty in the streamflow signatures and in the climatic indices was not considered because the time window used for their calculation (24 years) is long enough to assume limited bias and high precision (as shown by Kennard et al. (2010)).

*3.* A conceptual model would be helpful on section 3 to understand how selected metrics represent catchment processes.

Point taken. Figure 6 has been modified showing how the catchment has been modeled.

4. Metrics from continental scale studies are a good starting point but should not be the final call – there are also other relevant metrics that could be tested such as phase-offset between the seasonal cycle in precipitation and potential evaporation – see Donohue et al (2010) JoH for details.

The number of signatures and indices proposed in literature to represent streamflow and climate is enormous (e.g. 120 metrics considered by Kennard et al. (2010)); therefore we had to limit our selection and we decided to use the one proposed by Addor et al. (2017) since we think they cover a wide range of characteristics of the time series that they synthetize.

In the first submission of the paper, the climatic indices and the streamflow were selected to represent particular features of the time series (e.g. the flashiness index was used to measure the variability of the hydrograph). This choice was criticized in the review of Dr. Lieke Melsen that suggested to refer to other studies for the selection of signatures/indices. Therefore, the original version of the paper was modified to account for her suggestion.

We have also tried to calculate the phase-offset between the seasonal cycle in precipitation and potential evapotranspiration as suggested by this reviewer but, since precipitation and PET have the same seasonal cycle in all the catchments (as shown in figure 5 of the first submission of the paper, reported above), the phase-offset would be the same for all the catchments and therefore it would be excluded by the correlation analysis since it does not show variability (first bullet point in section 3.1.2 of the new revision). For this reason we would not include it in the paper.

5. Manuscript structure is a bit unusual and not easy to navigate. It looks more like a thesis/report than a paper. Would be better to group all the methods and results together instead of presenting them separately on sections 3 and 4.

We thank the reviewer for this suggestion. Now the paper has been restructured according to the standard practice of presenting methods followed by results. We hope that this major restructuring has improved the readability of the paper.

**Other comments**

Manuscript title is focused on methods rather than contribution. Currently title is a bit vague and not attractive as most hydrological modelling papers do data analysis and model building to understand catchment processes. Therefore, there is nothing new in the title and many potential readers might skip it if the title is kept the same (It is likely I would be one of them). I strongly recommend changing it focusing on the main contribution.

Point taken. We have changed the title into: "Understanding dominant controls on streamflow spatial variability to set-up a semi-distributed hydrological model: the case study of the Thur catchment", which focuses more clearly on the paper objectives.

L1-2: "The development of semidistributed hydrological models that reflect the dominant processes controlling streamflow spatial variability is a challenging task" – Irrelevant, every science has challenges, otherwise would not be science.

We have removed the sentence from the new version of the paper, and added two paragraphs in the introduction to better clarify the scope of our work.

*The term "semidistributed" sometimes appears as semi-distributed. Better standardise.*

Thank you for pointing that out; we have standardized this term in the paper, using semi-distributed everythere.

*Figure 1c: Forest and pasture are not easy to distinguish. Suggest use different colour for pasture.* We have changed the colors used in the figure. Now the figure should be more readable.

Figure 7 uses line plots to show variability of model performance across study catchments. Choice of plot type is a bit misleading as line plots are usually used to show continuity across the x-axis, such as time-series plots. Suggest use only dots / jitters instead to avoid misinterpretation.

We have removed the lines from figure 7

Most figures refer to metrics acronyms and a lot of back and forth is needed to find their definition and keep on track with reading. If authors do not want to redefine acronyms on figure captions, suggest present all the metrics in a table and cite table on figure captions.

We have added table 2 that contains all the symbols used in the paper to represent signatures and indices

P28 L2 – "varied considerably between catchments" – I think it should be among instead of between as several catchments are assessed.

We have changed the sentence accordingly.

P28 L7-8: "based on correlation analysis and expert judgment, we determined that climatic variables, especially the precipitation average, are the main controls the on streamflow average yearly values" – that's well known and trivial. I would expect more elaborated hydrological insights. Main issue is that section 3 is not robust enough to offer in-depth interpretations.

Although it is known that precipitation has a strong control on average streamflow, this is not granted in some cases where, for example, regional groundwater flow alters the water balance of the catchments. For this reason we believe that this point is still important for describing the hydrological processes happening in the catchment.

We have clarified these aspects in Section 4.1.3 point 1, Section 4.2.1 first paragraph, and Conclusions, point 4 of the new revision.

We are thankful for the reviews of Anonymous Referee #3 and we are looking forward to his/her assessment of our revised paper.

**References**

Addor, N., Newman, A. J., Mizukami, N., and Clark, M. P.: The CAMELS data set: catchment attributes and meteorology for large-sample studies, Hydrol Earth Syst Sc, 21, 5293-5313, 10.5194/hess-21-5293-2017, 2017.

Donohue, R. J., Roderick, M. L., & McVicar, T. R. (2010). Can dynamic vegetation information improve the accuracy of Budyko's hydrological model?. Journal of Hydrology, 390(1-2), 23-34.

Kennard, M. J., Mackay, S. J., Pusey, B. J., Olden, J. D., & Marsh, N. (2010). Quantifying uncertainty in estimation of hydrologic metrics for ecohydrological studies. River Research and Applications, 26(2), 137-156.

Trancoso, R., Larsen, J. R., McAlpine, C., McVicar, T. R., & Phinn, S. (2016). Linking the Budyko framework and the Dunne diagram. Journal of Hydrology, 535, 581-597.

**Reply to review by Dr. Shervan Gharari**

We thank Dr. Shervan Gharari for his careful read to the manuscript and insightful suggestions. As it can be noticed in the differences' file, the paper has undergone a major restructuring, in the spirit of capturing most of the suggestions of the reviewers. However, as the reviewers in this round of reviews are different from the reviewers from the previous round of reviewers, we had to be careful that the suggestions of the current reviewers are not in contrast with the modifications already made to comply to the suggestions of the previous reviewers. Cases where a conflict occurs are mentioned in our replies. Below, we answer in detail the various comments, and illustrate how we have addressed them in the revised version. The original comments of the reviewer are reported in *black and italics*, our replies in blue.

All the references to specific pages and lines of the paper are based on the version without track changes. Since the numbering of the sections has changed in the reviewed paper, we will call "first revision of the paper" the version that you have reviewed and "new revision" the version that we are submitting together with this reply.

**Review of "Data analysis and model building for understanding catchment processes: the case study of the Thur catchment"**

The paper tries to rationally build/infer an appropriate model suture based on data for Thur catchment. The authors have tried their best to answer to the reviewers' comments. Reading the manuscripts, reviewers' comments on the work and authors' replies, I have a feeling that most of the reviewers' concerns including myself is coming from the fact that the manuscript lack some fundamental direction which in turn might be the result of lack of proper research question. I personally don't have any issue with the choice of signatures and correlation analysis. At the end of the day this is an engineering decision that any modeller will make and there is little to back them even if correlation exists (present or absence of causation).

**The first question the authors should answer is the real purpose of this study.**

We have modified abstract and introductions to clarify the purpose of the study. In particular, the first paragraph of the introduction presents the general purposes of conceptual semi-distributed hydrological models in hydrology and some unresolved questions, which now better substantiate objectives of the studies, indicated in lines 37 of page 2 to 2 of page 3. The title has also been changed to more clearly reflect the study objectives, as suggested also by Anonymous Referee #3.

The spatial variability can have a wide range of interpretation. For example spatial variability to streamflow, or spatial variability to account for slope and aspect and etc. The authors should clearly make this case what spatial variability they are talking about (variability is case dependent).

To avoid misunderstandings, we have clarified that we are interested in explaining the hydrograph spatial variability. This is now more clearly apparent from the title ("Understanding first order controls on streamflow spatial variability..."), the abstract (e.g. "In order to appraise the dominant controls on streamflow spatial variability, and build a model that reflects them..."), and objectives (e.g. "The objective of this study is to develop a semi-distributed hydrological model with the appropriate level of functional complexity to reproduce streamflow spatial variability in the Thur catchment."). If the

catchment response is spatially variable there must be some spatially variable controls, and therefore we also analyze the spatial variability of meteorological inputs and catchments characteristics.

In this study, it is all about the streamflow as the models are calibrated against the streamflow. Streamflow is often easy to predict (calibrate). So the author should show the clear gain by moving toward spatially distributed input and spatial data such as slope and aspect, vegetation, geology, etc. In its current format the manuscript is lacking this direction. In the beginning, the manuscript promises to account for various processes but it is kind of missed in the manuscript or boiled down to very basic or common knowledge interpretation of the processes for streamflow simulation.

We hope our restructuring of the manuscript, where the methods are all presented in the same place, makes the reasoning of the paper clearer. In particular, we have revised sections 3.1.4 and 4.1.3 of the new revision to clarify these aspects.

In summary, the starting point for the modeling study is a semi-distributed model with uniform characteristics (single HRU) and distributed (per subcatchment) climatic inputs (MO) as the effect of distributing precipitation would be obvious from the signatures analysis; we then show the gain in moving towards accounting for the presence of spatially distributed snow, geology, and land use (vegetation).

As the reviewer notes, there are multiple characteristics that could be included in the model experiments. In order to reduce the number of model comparisons, we made use of the results of the signatures analysis. For example, signatures analysis showed that vegetation was not a major influencing factor and, in our model experiments, we confirmed that including vegetation does not improve model performance. Similarly, it could be expected that accounting for e.g. aspect, which was not a major influence factor according to the signatures analysis, would not improve model performance.

Sth else that I don't understand is the choice of model, for example from M0 to M1, if temperature is always above the threshold there will be no phase change for the precipitation. Then why even bother having model M0? The choice of the model is decided by the data itself (for example a land surface models have always snow component but if simulated for warm region they never simulate any snow).

It is clear that when there is snow (as in this case) the model needs to have a snow component. It is less obvious (at least just by looking at hydrographs) how much of the differences in seasonality of the streamflow response between catchments are due to snow. Due to the large lag time between snowfall and hydrograph response it would be difficult to quantify this aspect without model experiments and the main result of the comparison between M0 and M1 is that the attribution of difference in seasonality (represented by the mean half streamflow date) is due to the spatial variability of snow processes.

Moreover, the fact that the precipitation is first order control is also a bit obvious. If a multiplier is used to scale precipitation up and down it will be the most sensitive parameter of the model which in turn shows that the simulation is heavily affected by precipitation (or the driving force). Following that, I don't see much translation of the observed processes into the model and I don't see the added value of the added heterogeneity in the spatial models simulation etc. This can be further improved by the authors. It is clear that precipitation is a first order control on streamflow. Less clear, at least before carrying out any analysis, is if the spatial variability in streamflow average is only due to precipitation: several authors, for example, pointed out the role of regional groundwater flow and incorporated this possibility in the models; GR4J, for example, has a parameter that quantifies catchment gains or losses. This shows that a-priori there are several processes that can affect the water balance; our analysis is intended to understand which modeling decisions are relevant in this case study. We have clarified these aspects in Section 4.3 point 1, Section 4.2.1 first paragraph, and Conclusions, point 4 of the new revision.

Following this point, the choice of the models and the modeling looks a bit sloppy; in the sense that the continuum of model, spatial data is not very well visible. I think this can be further improved by the authors in the revised manuscript (maybe adding more model or stepwise introduction of spatial variability).

Because spatially distributed models are time consuming to develop, even within a multi-model framework, and expensive to run, we focused the comparisons on a few interesting cases which were decided following the signatures analysis. We have clarified the models line-up in Section 4.2.1 of the new revision, where we have specified the expectations that the various models are supposed to meet.

I also suggest the author to have a more structural in to the paper by organizing the signatures that they use for model evaluation. These signatures can be grouped into four main categories (1) the signatures that are coming from the spatial heterogeneity of the topography, geology, soil, land cover etc. (2) the signature that are coming from the response that the model is built to replicate such as flow duration curve, flashiness, etc, (3) the signatures that are coming from the system including the precipitation etc (4) hybrid such as runoff ratio. Each of these signatures have their own effect on the modeling result as some are used for calibration and some are not. I would suggest the author to segregate them more carefully in the test and analyses.

We tried to be explicit about the different nature of these metrics: we called metrics derived from the landscape properties "landscape indices" and indicated them with the letter  $\xi$ ; the metrics derived from the climate were named "climatic indices" and indicated them with the letter  $\psi$ ; the metrics derived from the streamflow, finally, were named "streamflow signatures" and indicated them with the letter  $\zeta$ . Only the runoff ratio and the streamflow elasticity are "hybrid signatures" and we have decided to put them in the category of the "streamflow signatures" as done by Addor et al. (2017).

Is it surprising M0 can do well for the annual average? In my experience, a single reservoir with evapotranspiration function can get the annual mean streamflow perfectly well, while it cannot get the correlation and variability.

The reviewer is right pointing out that a simple model calibrated on an individual gauge can get the average streamflow correctly; however this is a distributed model which is simultaneously calibrated on multiple gauges without catchment specific correction factors for precipitation, evaporation or streamflow. In addition, the model is evaluated in space-time validation, meaning that the model has not been calibrated in the specific gauge where it is evaluated. The ability of this model to simultaneously capture the annual averages at multiple gauges is, therefore, not a-priori obvious.

I am interested to know how the authors dealt with the nested gauges. The information/correlation in nested gauges can be replicated. Howe the correlation plays in for these nested basin. Any comment on that.

The model deals with nested catchments by routing the water from upstream catchments to the downstream outlets through transfer functions. By validating the model in space and time we are not reusing the same data. Clearly, this is a spatial validation in a nested setup, which is presumably easier to fulfill than a spatial validation in entirely different basins. This limitation has been added in Section 5 of the new revision, second last paragraph.

Why did the authors have use NS and likelihood at the same time? What would it add...? We used the likelihood because it was the objective function for model calibration. Since the model was calibrated simultaneously in multiple stations, the likelihood is an aggregated metric. The NS was, on the

other hand, calculated for each catchment individually.

I think both in the modeling set up and also discussion a significant elements regarding the scale is missing. For example, have refereed to some work, Kuentz et al., 2017, that did a large sample hydrology. Is the manuscript really is about large sample hydrology and if the study area is following large sample hydrology or is it about how the Thur catchment is functioning and how it is modelled. We have clarified that the paper is not about large sample hydrology but about distributed modeling (e.g. see first paragraph of the introduction). In order to formulate model decision we have used elements of catchment classification studies.

I would also suggest the author to look into the signature before and after bias correction or accounting for orographic effects. As mentioned earlier changing in forcing can drastically change the model output therefore it should also be noted how different the forcing becomes when is downscaled. Maybe I missed but how did the author include slope and aspect in their model?

Climatic inputs are influenced by orographic effects; as specified in Section 2 of the paper, the elevation has been considered in the interpolation of the data from the meteorological stations. We did not do sensitivity analyses on the input variables as it was outside of the scope of this paper.

I believe the manuscript can be an interesting contribution but in its current format it is far from being in perfect shape. The story needs to follow smoothly and the merit of this work should be better presented. Shervan Gharari

We are thankful for the reviews of Dr. Shervan Gharari and we are looking forward to his assessment of our revised paper.

[revised manuscript text omitted]
 avaluated from the subsequent                                                                                                                                         |
|    | some more that show very finite of no variation at an and, increase, may could be arready excluded from the subsequent correlation analysis: they are: $\eta_{\mu} = (1.\%) \eta_{\mu} = (0.\%) \eta_{\mu} = (4.\%) \eta_{\mu} = (3.\%)$ and $\eta_{\mu} = (0.\%)$ . |
|    | Fig. 2 shows the correlation between the remaining indices. It can be observed they all have strong internal correlation                                                                                                                                             |
| 5  | (r > 0.71) For this reason it was decided to rate only the and the loss that have lower correlation. The first represents an                                                                                                                                         |
| 5  | $(1 > 0.11)$ . For this reason it was decided to retain only $\varphi_{\mu}$ and $\varphi_{\mu s}$ , as they have rower correlation. The first represents an important term of the water hudget, the latter continues enough dynamics.                               |
|    | Table 4 shows the values of the catchment characteristics considered in this study. All of them have a coefficient of variation                                                                                                                                      |
|    | larger than the minimum threshold of 5%. Therefore none of them was excluded based on this criterion. The second                                                                                                                                                     |
|    | eritorion for the pre-exclusion of the estelyments characteristics, consisting in removing E occupying loss than 5% of the                                                                                                                                           |
| 10 | subcatchments, led to the suppression of $\xi_{rec}$ (which occupies 4% of the subcatchment).                                                                                                                                                                        |
|    | Figure 4 shows the correlations between eatchment characteristics: in many cases the high correlation is due to the fact that                                                                                                                                        |
|    | many indices are complementary (e.g. different types of geology). The following $\xi$ were selected (one index per class):                                                                                                                                           |
|    | • • • • • • • • • • • • • • • • • • •                                                                                                                                                                                                                         |
|    | $\xi_{\rm A}$ = $\xi_{\rm mp}$ and $\xi_{\rm mr}$ in representation of the topography                                                                                                                                                                                |
| 15 | $\xi_{12}$ and $\xi_{143}$ in representation of the topography,                                                                                                                                                                                                      |
| 15 | - - Σ L 2 for the fail decreatoristics.                                                                                                                                                                                             |
|    | - ζ sp for the goology                                                                                                                                                                                                                                    |
|    | $\varsigma_{\rm GC}$ to the periody.                                                                                                                                                                                                                                 |
|    | In summary, the onginar set of catemnent indices was reduced to a set of 5 indices.                                                                                                                                                                                  |
|    | 3.1.2 1.1.1 3.2.2 Selection of controlling factors on streamflow signatures                                                                                                                                                                                   |
| 20 | Fig. 5 reports the results of the Spearman correlation between alimatic indices plus established thereateristics on streamflow                                                                                                                                       |
| 20 | signatures. The upper papel contains the Spearman's rank coefficients and the lower papel shows a values associated with                                                                                                                                             |
|    | them.                                                                                                                                                                                                                                                                |
|    | The following results can be noted:                                                                                                                                                                                                                                  |
|    | The three statistics average precipitation $(4k_{\rm e})$ fraction of snow $(4k_{\rm e})$ and average elevation $(\xi_{\rm e})$ correlate                                                                                                                            |
| 25 | strongly with average streamflow (7.) and seesonality (7) ( $r > 0.64$ and p-value < 0.05). This correlation can                                                                                                                                                     |
| 20 | be interpreted as follows: subsetabments with high elevation $(\xi_{-})$ tond to have higher presidential $(\psi_{-})$ due to                                                                                                                                        |
|    | be interpreted as follows: subcateminents with high circulation $(\underline{\varphi}_{\underline{p}})$ that is have higher precipitation $(\underline{\varphi}_{\underline{p}})$ due to correspond to have more snow $(\underline{\mu})$ due to lower               |
|    | to instruction or which influences the accounting $\langle \zeta_{\mathbf{q}} \rangle$ . They also tend to have more show $\langle \varphi_{\mathbf{rs}} \rangle$ due to lower                                                                                       |
|    | temperatures, which influences the seasonanty $(\zeta_{\text{HPD}})$ .                                                                                                                                                                                               |
| 20 | • There are then some catchment characteristics that have no correlation ( $r < 0.45$ ) with the streamflow signatures                                                                                                                                               |
| 30 | (cateniment area $(\xi_{\pm})$ and land use $(\xi_{\pm})$ ) or limited correlation (aspect $(\xi_{\pm AS})$ and deep solit $(\xi_{\pm B})$ ), with $\tau < 0.64$ ).                                                                                                  |
|    | • The consolidated geology ( $\xi_{GC}$ ) presents a strong correlation ( $r = -0.87$ ) only with the baseflow index ( $\zeta_{BFI}$ ) that it                                                                                                                       |
|    | is not captured by the other indices.                                                                                                                                                                                                                                |
|    | • The streamflow signatures of low and high flows ( $\zeta_{qqs}$ and $\zeta_{Hqp}$ ) cannot be explained by any index, with little                                                                                                                                  |
|    | correlation only with $\psi_{\mathbf{P}}$ and $\xi_{\mathbf{TE}}$ ( $\tau < 0.60$ ) that is not sufficient to reach a p-value lower than 0.05.                                                                                                                       |
| 35 | These results are the premise for designing meaningful model experiments.                                                                                                                                                                                            |
|    | 3.1.31.1.1 3.3 designed model experiments aimed to confirm the hypothesized climatic and landscape controls                                                                                                                                                   |
|    | on streamflow Hypotheses for model building                                                                                                                                                                                                                          |
|    | Our hypothesis is that only a model that accounts for the influencing factors that affect the streamflow signatures will be able                                                                                                                                     |
|    | to reproduce spatial streamflow variability. In this section, we synthetize the outcomes of previous analyses in the form of                                                                                                                                         |
|    |                                                                                                                                                                                                                                                                      |

40 testable hypotheses for model building.

- The precipitation is the first driver of the differences in the water balance of the subcatchments. The effect of topographic variability manifests itself primarily as an influence on precipitation (amount and type). Accounting for variability of precipitation therefore implicitly reflects such effect of topography on the hydrograph, since the inputs were interpolated taking into account the effect of the elevation (Sect. 2).
- 2.1. Snow related processes (e.g. amount of snow, timing of snowmelt) control differences in streamflow seasonality between subcatchments.
  - 3. Geology exerts an important control on the partitioning between quick flow and baseflow.
  - 4.1. The other catchment characteristics (e.g. soil, vegetation) show little or no correlations with the streamflow signatures and therefore they should not be considered if the idea is to keep the model as simple as possible.

**10 These hypotheses will be tested through specific model comparisons, described in Sect. 4.1.5.**

• 4

5

15

25

The overall objective of the model experiments is to prove that only models that incorporate the correct dependencies are able to correctly predict regional streamflow variability. In order to test this assumption, the model experiments will include cases where the assumed dependencies are not incorporated. Omitting an assumed dependency leads to structurally simpler model, which may raise the doubt that potential differences in model performance might be due to differences in model complexity. For this reason, the model experiments will include cases where alternative dependencies are incorporated, which do not reduce model complexity. In order to keep the study and presentation tractable, the model experiments will be

**limited to a few cases, illustrated in Sect. 4.2.1 which we judge relevant for this specific application.**

**3.2 General structure of the semi-distributed hydrological model and model evaluation approach**

**20 3.21.1 Modelling**

**41-4.1 Methods**

This section describes the approach for building and testing a semi-distributed hydrological model designed to represent the observed streamflow and particularly the observed spatial variability of streamflow signatures. The general model structure is explained in Sect. 4.13.2.1, the error model and the calibration procedure are described in Sect. 4.13.2.2 and 4.13.2.3, the metrics utilized to assess the performance are shown in Sect. 4.1.4, and the model experiments done are illustrated in Sect. 4.1.53.2.4.

**4.1.13.2.1 4.1.1 General structure of the hydrological model**

We describe here the general model structure. Specific choices for: the variousdefinition of specific model experiments-are, which depends on the results of the signatures analysis done in the first step, will be described in SectionSect. 4.2.1.5.

- 30 The model uses a two-layers decomposition of the catchment:
  - Subcatchments are defined by the presence of the gauging stations; this subdivision wasis due to the necessity of having locations in the model where the streamflow wasis both observed and simulated and, therefore, it wasis possible to ca

---

## Author Response (AR3)

**Author's response**

**Changes in the paper**

The paper has been subject to revision in order to address the comments of the reviewers. The most significant changes concern an improvement of the modelling results section, with the intent of highlighting the relationship between hypotheses and modelling experiments, and the introduction of the normalized root mean square error to evaluate the performance of the models in representing the signatures. A summary of the changes is presented below, followed by the individual responses to the reviewers.

| SECTION | DESCRIPTION                                                                         |
|---------|-------------------------------------------------------------------------------------|
| 3.2.4   | Introduction of a new metric to evaluate the ability of the models to represent the |
|         | signatures                                                                          |
| 4       | Change in the numbering of the subsections to highlight the relationship between    |
|         | hypotheses and modelling experiments                                                |
| 4.3.2   | Added information about the normalized root mean square error                       |
| 4.4     | Changes in the text to highlight the relationship between hypotheses and modelling  |
|         | experiments                                                                         |
| 5       | Changes in the text to incorporate the suggestions of Anonymous Referee #3          |

**Changes in Figures**

The following figures were modified

| FIGURE   | DESCRIPTION                                                                     |
|----------|---------------------------------------------------------------------------------|
| 8, 9, 10 | Added the normalized root mean square error                                     |
| 9        | Kept only the snow-affected catchment in the panels representing the streamflow |
|          | seasonality signature                                                           |

**Reply to the editor Dr. Conrad Jackisch**

Dear Marco Dal Molin and co-authors.

Thank you again for your contribution to our special issue and the good work you invested into your manuscript during the revisions. I agree with the reviewers that the manuscript has highly improved and should be ready for publication very soon.

Please see the suggestions of the reviewers for final revisions. I find it very interesting that both reviewers and myself see some interesting points which have formed during the discussion and revisions. Could you please check, if they could be incorporated in your manuscript? The reviewers make very nice suggestions towards this. As referee #2 also pointed to some questions about the model evaluation, I think this should be clarified further. With regards to the overall revision suggestions of referee #2, I will leave it with you, how you chose to deal with it.

I am looking very much forward to receiving your reply. If you have any trouble accessing the reports of the referees, please contact me on shot notice.

All the best. Merry Christmas (or equivalent seasonal greetings), Conrad

We thank the Editor Dr. Conrad Jackisch for its thoughtful suggestions. We have done our best to incorporate the suggestions of the reviewers in the manuscript.

In particular, Anonymous Referee #3 suggested to incorporate some discussion points emerged during the previous round of reviews, which we have included in the revised version. Dr. Lieke Melsen, proposed to use a "bias metric" in the signature analysis, which has been included, leading to a more complete analysis of the results. Regarding her suggestions to change the structure of the paper, we have decided to maintain the classical methods-results structure (as suggested in the second round of revisions), but we have modified the modelling results section to highlight the connection between the hypotheses and the modelling experiments.

Kind regards,

Marco Dal Molin (on behalf of the coauthors).

**Reply to review by Anonymous Referee #3**

We thank the Anonymous Referee #3 for his/her careful read to the manuscript and insightful suggestions. Below, we answer in detail the various comments, and illustrate how we have addressed them in the revised version. The original comments of the reviewer are reported in *black and italics*, our replies in blue.

All the references to specific pages and lines of the paper are based on the reviewed version without track changes.

Authors have substantially improved the manuscript (MS) with regards to structure and readability. My comments have been mostly addressed and I believe the MS is in better shape for publication now. The discussion is not optimum yet and my suggestion is that it should undergo minor revision before publication.

Authors have developed some interesting explanations while answering reviewers' comments, which I think could be worthwhile to explore when revising the discussion. The points are highlighted below:

1) Sample size influence on approach choice - that would be useful to let people know and point out other approaches suitable for larger catchment samples:

"The possibility to use more advanced methods for metrics selection has been considered in the process of our study; the reason why we eventually selected a simple method is that the sample size of this study is relatively small. We are in fact limited to only 10 catchments. Studies that use complex regression techniques like random forests use a much larger sample of catchments; for example, the work proposed by Trancoso et al. (2016) deals with 355 catchments. Using such techniques risks to result in models that overfit the data, especially considering the fact that we would need to split the catchments in a calibration and a validation group. "

This point is already present in the discussion section, page 18 lines 8-14.

"The small number of subcatchments involved in this study (10) limits the range of viable methods for identifying relationships between landscape and climatic indices and streamflow signatures (Sect. 3.1) to rather simple approaches. [...] The usage of more advanced techniques, including machine learning approaches such as random forest or clustering analyses, are most efficient when larger samples are available and could represent a more suitable choice in these situations."

**2) Other potential metrics that can suit best other regions"**

"The number of signatures and indices proposed in literature to represent streamflow and climate is enormous (e.g. 120 metrics considered by Kennard et al. (2010)); therefore we had to limit our selection and we decided to use the one proposed by Addor et al. (2017) since we think they cover a wide range of characteristics of the time series that they synthetize."

**We have introduced this point in the discussion section, page 17 lines 40-43**

**3) These are interesting hydrological interpretations:**

"Although it is known that precipitation has a strong control on average streamflow, this is not granted in some cases where, for example, regional groundwater flow alters the water balance of the catchments. "

**We have introduced this point in the discussion section, page 17 lines 16-21**

"It is clear that when there is snow (as in this case) the model needs to have a snow component. It is less obvious (at least just by looking at hydrographs) how much of the differences in seasonality of the streamflow response between catchments are due to snow. Due to the large lag time between snowfall and hydrograph response it would be difficult to quantify this aspect without model experiments and the main result of the comparison between M0 and M1 is that the attribution of difference in seasonality (represented by the mean half streamflow date) is due to the spatial variability of snow processes." We have introduced this point in the discussion section, page 17 lines 22-26

"It is clear that precipitation is a first order control on streamflow. Less clear, at least before carrying out any analysis, is if the spatial variability in streamflow average is only due to precipitation: several authors, for example, pointed out the role of regional groundwater flow and incorporated this possibility in the models; GR4J, for example, has a parameter that quantifies catchment gains or losses. This shows that a-priori there are several processes that can affect the water balance; our analysis is intended to understand which modeling decisions are relevant in this case study" We have introduced this point in the discussion section, page 17 lines 22-26

**4) Good to bring on board in the discussion as well:**

"To avoid misunderstandings, we have clarified that we are interested in explaining the hydrograph spatial variability. "

The interest of the paper in explaining hydrograph spatial variability has been stated in the abstract ("In order to appraise the dominant controls on streamflow spatial variability") and in the introduction ("The objective of this study is to develop a semi-distributed hydrological model with the appropriate level of functional complexity to reproduce streamflow spatial variability in the Thur catchment", "Our specific objectives are to: (1) explore the spatial variability present in the Swiss Thur catchment regarding landscape characteristics, meteorological forcing and streamflow signatures; (2) identify the main climate and landscape controls that explain the variability of the hydrological response").

**5) This could be reinforced and explored together with point 1 above:**

"We have clarified that the paper is not about large sample hydrology but about distributed modeling (e.g. see first paragraph of the introduction). In order to formulate model decision we have used elements of catchment classification studies. The usage of element from catchment classification studies and the focus of the paper on distributed modelling have already emerged in section 3.1.1 (methods)

"Addor et al. (2017) recently compiled a comprehensive list of streamflow signatures and climatic indices for characterizing catchment behaviour (see Table 3 in Addor et al., 2017). Here, we adopted their selection: while being originally introduced for a study about large sample hydrology, we believe that the indices proposed are also able to capture several different aspects of the time series and are therefore suitable also for this regional study"

**Reply to review by Dr. Lieke Melsen**

We thank Dr. Lieke Melsen for her careful read to the manuscript and insightful suggestions. Below, we answer in detail the various comments, and illustrate how we have addressed them in the revised version. The original comments of the reviewer are reported in *black and italics*, our replies in blue. All the references to specific pages and lines of the paper are based on the reviewed version without track changes.

Dal Molin et al did a great job in incorporating feedback and improving the manuscript. The selection of signatures and indices is now more transparent and tested for mutual correlations, the hypotheses are now more directly linked to the correlation results, and the model results show interesting relations to the predefined hypotheses.

I have only one major concern left, and that is the interpretation of model results solely based on correlation (Figure 8, 9, 10). Correlation clearly does not account for bias (as becomes specifically apparent in Figure 10) and therefore does not guarantee "good" model results. In the text (Section 4.2) the correlation is now discussed as indicator to demonstrate which model performs best for which signature. A suggestion could be to include bias and variability (the KGE-terms) also in the same figures (or use the NSE?). Unless the authors have good reasons to do it the way they did, but then please clarify.

This is a good point and we thank the reviewer for pointing it out. While correlation is the only numerical metric considered to evaluate the results in Fig. 8, 9, and 10, simulations are also evaluated qualitatively looking at the alignment of the points to the diagonal.

In the new revision, we capture the bias calculating the normalized root mean square error between modelled and observed signatures.

**Minor point;**

I would expect the lower right panel (M1, HFD) of figure 8 to be the same as the third row first column panel (M1, HFD) of figure 9, but they are different. Is this because in Figure 8 only snow catchments are included, and in Figure 9 all? If so, please put more clearly and also explain why for figure 9 no snow-catchment selection was made.

Yes, in the old version of the manuscript Figure 8 presents only snow-affected catchments while Figure 9 shows all the catchments. We acknowledge that this has been a bad choice and, therefore, we now show, in the new revision, only snow-affected catchments regarding HFD.

**Textual;**

> "an uniform" -> "a uniform" (several times)

> "not represents well extreme values " -> "not represents extreme values well"

> "which therefore represents an independent evaluation metrics" -> "which therefore represents independent evaluation metrics"

**Thank you for the corrections. We implement them in the text.**

**Suggestions;**

I have a few suggestions that might improve the conveyance of the conclusions. This is not based on the content, and therefore beyond the scope of my role as reviewer. As such, it is up to the authors to decide whether they agree and if it is worth the effort.

I think the manuscript can benefit from leaving the traditional 'methods-results'-structure. For instance, the results on the mutual correlation of the metrics and indices could be presented right after the introduction of these metrics and indices. After that, the second part of the paper can focus solely on the modelling.

The general structure of the paper has been a major subject of discussion during the earlier review stage. The original paper was in the structure suggested by the reviewer, but it was criticized the Anonymous Referee #3 for having two methods-results sections. We therefore restructured it to the current format. We therefore kept the current structure, but made some changes in the results section to address the following points.

Also, I think it would be insightful if the Results-section follows the structure of the four hypotheses. Hypothesis 1 is on precipitation as main driver. Then the first sub-section should be on testing this hypothesis, and so on. Now, the results-section is a little bit of everything, and the hypotheses are only clearly discussed again in 4.2.4. Which is a pity given that I think you chose an elegant way of approaching semi-distributed modelling.

We thank the reviewer for the suggestion. We have now changed the structure of the results section to highlight the pattern from hypotheses formulation to testing and verification. In particular, section 4.2 now is about the formulation of the hypotheses (old section 4.1.3) and the design of the modelling experiments (old 4.2.1); section 4.3 presents the modelling results in terms of performance metrics (old 4.2.2) and signatures representation (old 4.2.3); section 4.4 interprets the modelling results, relating them to the hypotheses (old 4.2.4, with mayor changes to highlight the hypotheses-modelling pattern).

In the same way, I think the manuscript could benefit from one clear conceptual figure that demonstrates the approach and the conclusions. For me, the main value of this paper is in demonstrating a way to thoughtfully develop a semi-distributed model. This procedure can be schematized and as such better convey the message to future-semi-distributed-model-developers. The same is true for the hypotheses and the different model structures; a simple figure showing the different model structures, the hypothesis, and whether the hypothesis was confirmed or rejected could summarize all insights from this study. If number of figures becomes too high, the signature/indices-correlation figures can be considered for supplementary material.

We thank the reviewer for her suggestion; we think that, with the changes made to the paper, the procedure and the results are clearer and, therefore, such conceptual figure is not needed.

Overall, I think, besides the few small points I raised that might be clarified by the authors, the manuscript is in good shape for publication.

[revised manuscript text omitted]

---

## Author Response (AR4)

*Dear Marco Dal Molin and co-workers,*

*thank you again for your contribution to our special issue and the improvements of you manuscript. I think it really has reached a very nice state, which I am happy to accept.*

*One rather technical issue which should have been caught much earlier remains: What is about the availability of your data and scripts? You should be aware of the data policy of HESS which actually requires to make these available alongside the manuscript (https://www.hydrology-and-earth-system-sciences.net/about/data_policy.html). We have even stressed this issue further by having a joint special issue in both HESS and ESSD. Could you please include the (required) Data and Code Availability Statement in your manuscript and carefully check how you can make things reproducibly available as supplement or through any other path (e.g. Pangaea or simply GitHub)?*

*Thank you again. All the best.*

*Conrad*

Dear Dr. Conrad Jackisch,

Thank you for granting us additional time, it is our fault not to have realized this issue, and we agree it is important to have the data shared.

We have put all the data and codes used in the paper in a repository managed by our institute (Eawag). We have a DOI associated to the repository, which we have reported in the data statement of the paper, but it will be active only after the publication of the paper.

We hope that this satisfies the data policy of HESS.

Kind regards,

Marco Dal Molin (on behalf of the coauthors).

[revised manuscript text omitted]

| Unsaturated reservoir (UR) | $$\overline{S_{UR}} = \frac{S_{UR}}{S_{max}^{UR}}$$ |
| Unsaturated reservoir (UR) | $$E_{UR} = C_e(PET) f_m(S_{UR}|0.01)$$ |
| Unsaturated reservoir (UR) | $$Q_{UR} = P_{UR} f_p(\overline{S_{UR}}|2)$$ |
| Slow reservoir (SR) | $$P_{SR} = D Q_{UR}$$ |
| Slow reservoir (SR) | $$Q_{SR} = k_{SR} S_{SR}$$ |
| Lag function[c] | $$P_{FR} = (P_L * h_{lag})(t)$$ |
| Lag function | $$h_{lag} = \begin{cases} 2t / \left(t_{rise}^{lag}\right)^2 \text{ if } t \leq t_{rise}^{lag} \\ 0 \text{ if } t > t_{rise}^{lag} \end{cases}$$ |
| Fast reservoir (FR) | $$Q_{FR} = k_{FR} S_{FR}^3$$ |
| Lags in the network[c] | $$Q_{out} = (Q_{in} * h_{lag}^{net})(t)$$ |
| Lags in the network | $$h_{lag}^{net} = \begin{cases} 2t / \left(t_{rise}^{OL/IL}\right)^2 \text{ if } t \leq t_{rise}^{OL/IL} \\ \left(1/t_{rise}^{OL/IL}\right) \left(1 - \left((t - t_{rise}^{OL/IL})/t_{rise}^{OL/IL}\right)\right) \text{ if } t_{rise}^{OL/IL} < t \leq 2t_{rise}^{OL/IL} \\ 0 \text{ if } t > 2t_{rise}^{OL/IL} \end{cases}$$ |

[a] This equation is smoothed using logistic scheme, Eq. (8) in Kavetski and Kuczera (2007), with smoothing parameter $m_P = 1.5°C$

[b] This equation is smoothed using logistic scheme, Eq. (13) in Kavetski and Kuczera (2007), with smoothing parameter $m_M = 1.5°C$

[c] The operator $*$ denotes the convolution operator, smoothed according to Kavetski and Kuczera (2007)

**Table A4: Constitutive functions**

| Function | Name |
|---|---|
| $f_e(x\mid\theta) = 1 - exp(-x/\theta)$ | Tessier function. Note that $f_e(x\mid\theta) \to 1$ as $x \to \infty$ |
| $f_p(x\mid\theta) = x^\theta$ | Power function |
| $f_m(x\mid\theta) = \dfrac{x(1+\theta)}{x+\theta}$ | Monod–type kinetics, adjusted so that $f_m(1\mid\theta) = 1$ |

$f_e(x\mid\theta) = 1 - exp(-x/\theta)$

$f_p(x\mid\theta) = x^\theta$

$f_m(x\mid\theta) = \dfrac{x(1+\theta)}{x+\theta}$